# Talin and kindlin use integrin tail allostery and direct binding to activate integrins

Jonas Aretz [1], Masood Aziz[2,3], Nico Strohmeyer [4], Michael Sattler[2,3] & Reinhard Fässler [1]✉

Integrin affinity regulation, also termed integrin activation, is essential for metazoan life. Although talin and kindlin binding to the β-integrin cytoplasmic tail is indispensable for integrin activation, it is unknown how they achieve this function. By combining NMR, biochemistry and cell biology techniques, we found that talin and kindlin binding to the β-tail can induce a conformational change that increases talin affinity and decreases kindlin affinity toward it. We also discovered that this asymmetric affinity regulation is accompanied by a direct interaction between talin and kindlin, which promotes simultaneous binding of talin and kindlin to β-tails. Disrupting allosteric communication between the β-tail-binding sites of talin and kindlin or their direct interaction in cells severely compromised integrin functions. These data show how talin and kindlin cooperate to generate a small but critical population of ternary talin–β-integrin–kindlin complexes with high talin–integrin affinity and high dynamics.

Integrins adhere cells to the extracellular matrix, probe biochemical and biophysical properties of the extracellular matrix and convert the information into cellular responses such as spreading, migration, proliferation, survival and differentiation.

Integrins are α–β heterodimers consisting of a large ectodomain, a single-span transmembrane (TM) helix and a short C-terminal cytoplasmic tail (CT). A hallmark of integrins is the ability to reversibly switch between conformations with low and high affinity for ligand. The affinity switch requires the binding of the adaptor proteins talin and kindlin to the β-integrin CT (β-CT). In mammals, there are two talin (TLN1 and TLN2) and three kindlin (KIND1, KIND2 and KIND3) isoforms. Talin and kindlin colocalize in integrin-containing focal adhesions and cooperate to enable integrin–ligand binding and cell adhesion[1–3]. The mechanism underlying this cooperation remains a major unresolved question in adhesion biology.

Talin and kindlin are FERM (protein 4.1, ezrin, radixin, moesin) domain proteins consisting of F1, F2 and F3 subdomains and an additional N-terminal talin- and kindlin-specific ubiquitin-like F0 subdomain[4,5]. The F2 subdomain of kindlin harbors a pleckstrin homology (PH) domain, which binds negatively charged lipids such as phosphatidylinositol 4,5-bisphosphate (PtdIns(4,5)P$_2$ or PIP2) and phosphatidylinositol (3,4,5)-trisphosphate (PtdIns(3,4,5)P$_3$ or PIP3)[6–8]. The talin FERM domain, called the talin head domain (THD), binds charged membrane lipids such as PIP2 (ref. 9) and is connected via a flexible linker peptide to an elongated, mechanosensitive F-actin-binding rod domain[10]. The F3 subdomains of talin and kindlin fold into a phosphotyrosine-binding domain and bind distinct and juxtaposed regions in β-CTs. NMR and crystallographic studies show that talin F3 binds the membrane-proximal NPxY motif and the α-helical region of the β$_3$-CT[11,12], and kindlin F3 binds the membrane-distal NxxY motif[4,13].

A ternary talin–β-integrin–kindlin complex has been observed for different kindlin and β-integrin isoforms by NMR spectroscopy, analytical ultracentrifugation or super-resolution microscopy[14–16] and is considered crucial for talin–kindlin cooperativity. However, this hypothesis has not been proven so far. Theoretically, one would expect that talin and kindlin binding to the β-CT cooperate by amplifying each other's activity[10]. In the case of the β$_3$-CT, surface plasmon resonance experiments point to independent binding of TLN1 and KIND2 to the β$_3$-CT[14]. On the contrary, pulldown experiments reported competition

[1]Department of Molecular Medicine, Max Planck Institute of Biochemistry, Martinsried, Germany. [2]Department of Bioscience, Technical University of Munich, TUM School of Natural Sciences, Garching, Germany. [3]Helmholtz Munich, Institute of Structural Biology, Neuherberg, Germany. [4]Department of Biosystems Science and Engineering, Eidgenössische Technische Hochschule Zürich, Basel, Switzerland. ✉e-mail: faessler@biochem.mpg.de

of recombinant KIND2 and KIND3 by TLN1 (ref. 17), and molecular dynamics simulations suggested that talin and kindlin differentially influence each other during $\beta_3$-CT binding[18].

Here, we investigated the mechanistic basis of talin and kindlin association with $\beta$-CT. We found that ternary interactions of THD and KIND2 with $\beta_1$-CT or $\beta_3$-CT can induce an allosteric change of the $\beta$-CT, which increases THD and decreases KIND2 affinity. We also observed a direct talin–kindlin interaction, which likely enables rebinding and maintaining the populations of ternary talin–$\beta$-integrin–kindlin complexes at a critical threshold. The complex cooperativity between talin and kindlin results in an intrinsic cycle of assembly and disassembly of ternary talin–$\beta$-integrin–kindlin complexes that is solely governed by molecular communication between talin and kindlin.

## Results

### $\beta_1$-tail, TLN1 and KIND2 form a ternary complex at equilibrium

To determine whether the $\beta_{1A}$ CT splice isoform ($\beta_1$-CT) assembles, similar to $\beta_3$-CT[14], a ternary complex with TLN1 and KIND2 in vitro, we used NMR spectroscopy to characterize the interaction of isotope-labeled $\beta_1$-CT with the unlabeled F3 domain of TLN1 (TLN1-F3) and KIND2 lacking the flexible loop in F1 and the PH domain ($\Delta$KIND2; Fig. 1a and Extended Data Fig. 1a) to reduce molecular weight and enhance solubility and NMR spectral quality. NMR [$^1$H,$^{15}$N and $^1$H,$^{13}$C]methyl correlation spectra (Fig. 1b-d and Extended Data Fig. 1b,c) are affected by the presence of a binding partner, and changes appear either as chemical shift perturbation (CSP), defined as shift of an NMR signal (Extended Data Fig. 1b), or line broadening leading to decreasing signal intensity (Fig. 1b and Extended Data Fig. 1e). Interestingly, $\Delta$KIND2 binding affected the $\beta_1$-CT$^{G778}$ residue, which is located in the talin-binding site, whereas TLN1-F3 binding did not affect $\beta_1$-CT$^{G797}$ in the kindlin-binding site, pointing to a different mutual influence of kindlin and talin during $\beta_1$-CT binding. Of note, the line width of an NMR signal is related to the molecular weight of the protein tumbling in solution. Thus, the line broadening observed during complex formation will reflect the increase in the molecular weight of the complex. In addition, the stability of the complex, that is, the binding off-rate and local conformational dynamics due to interaction with the binding partners, can provide an additional contribution to the line width. Residues in $\beta_1$-CT undergoing the greatest line broadening shown by NMR signal are those in the talin- or kindlin-binding sites. The extent of line broadening likely reflects the molecular weight of the complex, although additional contributions from binding kinetics or conformational dynamics cannot be excluded.

Superposition of $^1$H–$^{15}$N correlation spectra of $\beta_1$-CT in the absence and the presence of TLN1-F3 or $\Delta$KIND2 shows reduced peak intensities assigned to a region from A773 to A786 including the membrane-proximal NPxY motif for TLN1-F3 and from Y783 to the C-terminal end of the $\beta_1$-CT including the membrane-distal NPxY motif for $\Delta$KIND2 (Fig. 1b,c and Extended Data Fig. 1e). This finding agrees well with the reported talin- and kindlin-binding sites[4,11,12]. Furthermore, $^1$H–$^{15}$N (Fig. 1c and Extended Data Fig. 1e) and $^1$H–$^{13}$C (Fig. 1d and Extended Data Fig. 1c) correlation spectra show minor CSPs and small changes in signal intensities assigned to the membrane-proximal, $\alpha$-helical region from H758 to K770, suggesting that, in contrast to $\beta_3$-CT, this conserved region is not an important binding site for TLN1-F3 in $\beta_1$-CT[12,19]. Interestingly, spectral peak intensities assigned to the $\beta_1$-CT$^{Y783–A786}$ region are reduced upon addition of TLN1-F3 as well as $\Delta$KIND2, suggesting that talin- and kindlin-binding sites overlap in $\beta_1$-CT (Fig. 1c).

To test whether talin and kindlin form a ternary complex with $\beta_1$-CT, we titrated both $\Delta$KIND2 and TLN1-F3 to isotope-labeled $\beta_1$-CT. While the chemical shifts in the $\alpha$-helical region (H758–K770) remained almost unchanged (Extended Data Fig. 1b), the average signal intensities throughout the entire $\beta_1$-CT decreased in the presence of increasing $\Delta$KIND2 and TLN1-F3 concentrations compared to spectra of binary TLN1-F3–$\beta_1$-CT or $\Delta$KIND2–$\beta_1$-CT complexes (Fig. 1c and Extended

Data Fig. 1e). This global signal intensity decrease in the $\beta_1$-CT is consistent with a molecular weight increase upon formation of a ternary complex. In addition, the $^1$H–$^{13}$C correlation spectra of $\beta_1$-CT in the presence of both TLN1-F3 and $\Delta$KIND2 (Fig. 1d and Extended Data Fig. 1c) reveal that methyl signals of several amino acids between the membrane-proximal and -distal NPxY motifs (A786–V791) are line broadened beyond detection, which is neither observed for all the other peaks nor is it in binary $\beta_1$-CT–TLN1-F3 or $\beta_1$-CT–$\Delta$KIND2 complexes. This observation indicates changes in the binding regime (on and off kinetics) during formation of the ternary TLN1-F3–$\beta_1$-CT–$\Delta$KIND2 complex and possibly a conformational change of $\beta_1$-CT upon simultaneous binding of TLN1-F3 and $\Delta$KIND2. Because simultaneous as well as sequential addition of TLN1-F3 and $\Delta$KIND2 to labeled $\beta_1$-CT produces identical $^1$H–$^{13}$C correlation spectra, we conclude that the ternary complex forms at thermodynamic equilibrium (Extended Data Fig. 1d).

### Talin and kindlin affinities for $\beta$-tails

To investigate whether talin and kindlin influence each other upon $\beta_1$-CT or $\beta_3$-CT binding, we determined their dissociation constants ($K_d$ values) for $\beta$-CTs at thermodynamic equilibrium. In the simplest case of ternary talin–$\beta$-CT–kindlin complex formation, the assumption is that either talin or kindlin binds to $\beta$-CT first and then the remaining free adaptor binds to the occupied $\beta$-CT, which can be illustrated with an 'energy square' (equation (1)):

$$
\begin{array}{ccc}
\text{Talin + kindlin + } \beta\text{-CT} & \xrightleftharpoons{\quad K_{a,\text{talin}} \quad} & \text{kindlin + talin–}\beta\text{-CT} \\
K_{d,\text{kindlin}} \Big\Updownarrow & & \Big\Updownarrow K^*_{a,\text{kindlin}} \\
\text{Talin + kindlin–}\beta\text{-CT} & \xrightleftharpoons[\quad K^*_{d,\text{talin}} \quad]{} & \text{talin–}\beta\text{-CT–kindlin.}
\end{array}
\qquad (1)
$$

Measurement of the dissociation constant of talin or kindlin for the $\beta$-CT that is either free ($K_d$) or occupied with kindlin or talin ($K_d^*$) allows us to prove or disprove the ternary-complex model. As the total energy does not change in a closed system, microscopic reversibility demands that the product of dissociation and association constants ($K_a = 1/K_d$) around a reaction cycle of an energy square must equal 1 (equation (2)):

$$
\begin{aligned}
& K_{a,\text{talin}} \times K^*_{a,\text{kindlin}} \times K^*_{d,\text{talin}} \times K_{d,\text{kindlin}} \\
& = \frac{1}{K_{d,\text{talin}}} \times \frac{1}{K^*_{d,\text{kindlin}}} \times K^*_{d,\text{talin}} \times K_{d,\text{kindlin}} = 1
\end{aligned}
\qquad (2)
$$

Furthermore, a comparison of the dissociation constants of talin and kindlin for free or occupied $\beta$-CT allows us to differentiate independent ($K_d = K_d^*$), competitive ($K_d < K_d^*$) or mutually reinforced binding of talin and kindlin to $\beta$-CT ($K_d > K_d^*$) and assign a potential function to the ternary talin–$\beta$-integrin–kindlin complex.

We determined the talin- and kindlin-binding mode to $\beta_1$-CT by microscale thermophoresis (MST)[20], which quantifies changes in fluorescence induced by a temperature-related intensity change as well as thermophoresis of a fluorescently labeled probe. The extent of temperature-related intensity change due to ligand binding and thermophoresis due to size, charge and solvation entropy differences were used to quantify binding affinities in titration experiments. To minimize heat effects, we measured the fluorescence changes only 1.5 s before and after turning on the infrared laser (Extended Data Fig. 1f). MST-based $K_d$ measurements of recombinant THD1, THD2 and KIND2 (Fig. 1a and Extended Data Fig. 1a) for ATTO 488-labeled (488)-$\beta_1$-CTs were performed in the absence and the presence of KIND2, THD1 or THD2 at near-saturation binding concentrations (Extended Data Fig. 1f–h). The maximal solubilities of KIND2 and THD1, which were about 500 $\mu$M (40 mg ml$^{-1}$) and 2 mM (100 mg ml$^{-1}$), respectively,

allowed only near-saturation binding experiments and the determination of apparent affinities for β-CTs.

MST experiments revealed that near-saturation binding levels of KIND2 did not interfere with THD1 binding to 488–$\beta_1$-CT, resulting in $K_{d,THD1} = K_{d,app,THD1}$* (equations (1) and (2)). Unexpectedly, THD1 at near-saturation binding concentrations decreased KIND2 binding to $\beta_1$-CT (Extended Data Fig. 1g,h), resulting in $K_{d,KIND2} < K_{d,app,KIND2}$*. However, measurements with the 488–$\beta_1$-CT produced bell-shaped binding curves in the presence of KIND2 (Extended Data Fig. 1h), which are quite frequently observed in MST traces caused by unknown physical phenomena[21]. To obtain more accurate MST measurements, we measured binding to 488–$\beta_3$-CTs, which produced sigmoid, one-site-binding curves when plotted against increasing concentrations of THD1, THD2 and KIND2, respectively, with dissociation constants of $108 \pm 12$ μM for THD1, $39 \pm 3$ μM for THD2 and $9 \pm 2$ μM for KIND2 (Fig. 1e,f, Extended Data Fig. 1i and Extended Data Table 1). In agreement with $\beta_1$-CT competition measurements (Extended Data Fig. 1g,h), the apparent dissociation constants of THD1 or THD2 for $\beta_3$-CT were unaffected by the presence of KIND2 ($K_{d,THD} = K_{d,app,THD}$*; Fig. 1e and Extended Data Fig. 1i), whereas the apparent dissociation constants of KIND2 increased to $54 \pm 27$ μM, $120 \pm 1$ μM, $27 \pm 5$ μM and $47 \pm 14$ μM in the presence of THD1, TLN1-F3, THD2 and TLN2-F3, respectively ($K_{d,KIND2} < K_{d,app,KIND2}$*; Fig. 1f and Extended Data Fig. 1j). Strikingly, insertion of MST-derived dissociation constants of THD1, KIND2 and $\beta_3$-CT in the energy square did not equal 1 but produced a value of $0.16 \pm 0.11$:

$$\frac{1}{108 \pm 12\,\text{μM}} \times \frac{1}{54 \pm 27\,\text{μM}} \times (102 \pm 40\,\text{μM})$$

$$\times (9 \pm 2\,\text{μM}) = 0.16 \pm 0.1$$

The unidirectional competitive behavior of THD1 and KIND2 for $\beta_3$-CT (and $\beta_1$-CT) at chemical equilibrium contradicts a simple ternary-complex model (Fig. 1g and equation (1)) and suggests that binding of talin and kindlin to β-CT is much more complex.

To confirm unidirectional competition between talin and kindlin with an orthogonal assay that includes lipid-binding sites for THD1 and KIND2, we incorporated recombinant, biotinylated $\beta_1$ and $\beta_3$ TM- and CT-containing polypeptides ($\beta_1$-TM–CT, $\beta_3$-TM–CT; Fig. 1a) into 10% phosphatidylinositol phosphate- and 90% phosphocholine-containing nanodiscs (Extended Data Fig. 1a). The reconstituted nanodiscs were immobilized on streptavidin beads and analyzed in a flow cytometry-based reporter-displacement assay (FC-RDA) that allowed us to determine the concentration of unlabeled THD1 or KIND2 required to decrease binding of fluorescently labeled THD1 and KIND2 to 50%, respectively ($IC_{50}$; Fig. 1h,i). The $IC_{50}$ values of unlabeled THD1 competing with fluorescently labeled THD1 and of unlabeled KIND2 competing

with fluorescently labeled KIND2 report affinities. The $IC_{50}$ values of unlabeled THD1 competing with fluorescently labeled KIND2 and of unlabeled KIND2 competing with fluorescently labeled THD1 report the capability to displace the other adaptor, which is related to the apparent dissociation constant $K_{d,app}$*, measured by MST. FC-RDA measurements revealed that affinities of THD1 as well as KIND2 for $\beta_1$-TM–CTs and $\beta_3$-TM–CTs embedded in PIP2- or PIP3-containing nanodiscs were in the range of around 80–100 nM and, for $\alpha_5$-TM lacking the cytoplasmic domain (tailless $\alpha_5$-TM) embedded in PIP2- or PIP3-containing nanodiscs, were in the range of around 300–500 nM. As the tailless $\alpha_5$-TM interacts with neither talin nor kindlin, these findings indicate that charged lipids contribute the largest binding energy for talin and kindlin, whereas β-tails make a minor contribution (Extended Data Table 1), which is in line with reports for THD1 and $\beta_3$-CT[22]. The ability of THD1 to compete with labeled KIND2 and of KIND2 with labeled THD1 with similar $IC_{50}$ values from tailless $\alpha_5$-TM embedded in PIP2- or PIP3-containing nanodiscs indicates that talin and kindlin compete with similar efficiency for the same lipid-binding sites. In line with the data obtained by MST (Fig. 1g), KIND2 was unable to effectively outcompete THD1 binding to $\beta_1$-TM–CTs as well as $\beta_3$-TM–CTs embedded in PIP3- or PIP2-containing nanodiscs, whereas THD1 readily displaced KIND2 (Fig. 1i and Extended Data Fig. 1k). MST data also showed that THD1 failed to displace KIND2 from talin-binding-impaired $\beta_1$-CT$^{Y783A}$ and $\beta_3$-CT$^{Y772A}$ (Extended Data Fig. 1l,m), which altogether indicates that THD1 displaced KIND2 in a β-CT-binding-dependent manner.

## KIND2 directly binds TLN1 to stabilize the ternary complex

To achieve microscopic reversibility, the decrease in KIND2 affinity for $\beta_3$-CT in the presence of THD1 (Fig. 1g) must be accompanied by the same decrease in THD1 affinity in the presence of KIND2. Because THD1 affinity, however, does not decrease in the presence of KIND2, we hypothesized that allostery in the β-CT and/or between talin and KIND2 caused by a direct talin–KIND2 interaction counteracts the decrease in THD1 affinity and may lead to unidirectional binding preference. To investigate the talin–kindlin interaction, we probed for an interaction of TLN1-F3 with KIND2 in the absence and presence of $\beta_1$-CT using NMR. First, we examined whether TLN1-F3 and KIND2 bind in the absence of β-CTs by adding KIND2 in a 0.75-fold ratio to uniformly $^{15}$N-labeled TLN1-F3. Analysis of $^1$H–$^{15}$N heteronuclear single quantum coherence (HSQC) NMR spectra revealed CSPs in the $\alpha_1$-helix of TLN1-F3 located between the integrin-binding site and the flexible linker segment that joins THD and the rod domain in full-length talin (Fig. 2a–c and Extended Data Fig. 2a). NMR spectral changes in the $\alpha_1$-helix of TLN1-F3 differ from those observed in NMR spectra of binary complexes between $\beta_1$-CT and uniformly $^{15}$N-labeled TLN1-F3 (Extended Data Fig. 2b–d), indicating that the KIND2 and β-CT-binding sites localize

**Fig. 1 | Ternary interactions between talin, KIND2 and β-CT. a**, Proteins used in this study. Amino acid numbering of isoforms is shown in brackets. ΔKIND2 lacks the flexible loop (star; Δ168–217) and the PH domain (Δ337–512). **b**, Overlay of one-dimensional traces of amide signals of residues G778 and G797 in talin- and kindlin-binding regions of $\beta_1$-CT in $^1$H–$^{15}$N HSQC NMR spectra of $^{15}$N-labeled $\beta_1$-CT before (black) and after addition of increasing stoichiometries of ΔKIND2 (top, green) or TLN1-F3 (middle, blue) or both TLN1-F3 and ΔKIND2 (bottom, purple). AU, arbitrary units. **c**, Intensity ratio of peaks of $\beta_1$-CT in the presence and the absence of ΔKIND2 (top, green), TLN1-F3 (middle, blue) or both (bottom, magenta). Talin- (blue) and kindlin- (green) binding sites are indicated above plots (MP, membrane proximal; MD, membrane distal). The isolated peaks in **b** are indicated by (1) amino acid numbering, shown below plots, (2) dashed lines and (3) reported mean intensity values for each titration in their respective color code. **d**, Magnified view of [$^1$H,$^{13}$C]methyl correlations observed for 100 μM $^{13}$C,$^{15}$N-labeled $\beta_1$-CT before (black) and after addition of ΔKIND2 (green) or TLN1-F3 (blue) or TLN1-F3 and ΔKIND2 (purple). Methyl groups of A764 (α-helical), A773 (membrane-proximal NPxY) and A786 (membrane-distal NPxY) are shown. Arrows indicate chemical shifts induced by protein

addition. **e**, MST measurements of THD1 affinity for 488–$\beta_3$-CT in the presence (orange) or the absence (blue) of 30–60 μM KIND2 (3–6-fold excess of $K_d$). **f**, MST measurements of KIND2 affinity for 488–$\beta_3$-CT in the presence (purple) or the absence (green) of 250–450 μM (2–4-fold excess of $K_d$) THD1 or 100–400 μM (2–10-fold excess of $K_d$) THD2 (purple). **g**, Binding affinities of THD1 and/or KIND2 for $\beta_3$-CT are inconsistent with the simple ternary-complex model. **h**, Exemplary dose–response curves from FC-RDA showing competition of ATTO 565-labeled THD1 (565–THD1) with unlabeled THD1 (blue) or unlabeled KIND2 (magenta) and competition of Alexa 647-labeled KIND2 (647–KIND2) with unlabeled KIND2 (green) or unlabeled THD1 (orange) for $\beta_1$-TM–CT embedded in PIP2-containing nanodiscs. Mean fluorescence intensity (MFI) was normalized to positive and negative values. Dashed lines indicate the $IC_{50}$ value of 565–THD1 competing with unlabeled THD1 (blue) or unlabeled KIND2 (purple). **i**, FC-RDA-generated $IC_{50}$ values of unlabeled THD1 competing with 565–THD1 (blue) and 647–KIND2 (orange) and unlabeled KIND2 competing with 647–KIND2 (green) and 565–THD1 (purple) for $\beta_1$-TM–CT, $\beta_1$-TM–CT$^{Y795A}$, $\beta_1$-TM–CT$^{Y783A}$ and $\alpha_5$-TM embedded in PIP3-containing nanodiscs and $\beta_3$-TM–CT embedded in PIP2- or PIP3-containing nanodiscs. Bars display mean ± s.d. with a line for the median.

to adjacent but distinct regions of the TLN1-F3 domain. This finding was confirmed by adding KIND2 to uniformly $^{15}$N-labeled TLN1-F3 in the presence of $\beta_1$-CTs at saturating concentrations, which induced CSPs in the $\alpha_1$-helix as well as in the integrin-binding site of TLN1-F3 (Extended Data Fig. 2e–g).

To narrow down the KIND2-binding site, we substituted each amino acid in the TLN1-F3 $\alpha_1$-helix individually with alanine or glutamic acid to identify THD1 mutants that affect affinities for $\beta_3$-CT in the presence of near-saturation binding concentrations of KIND2. MST measurements revealed that, among all THD1 mutants tested

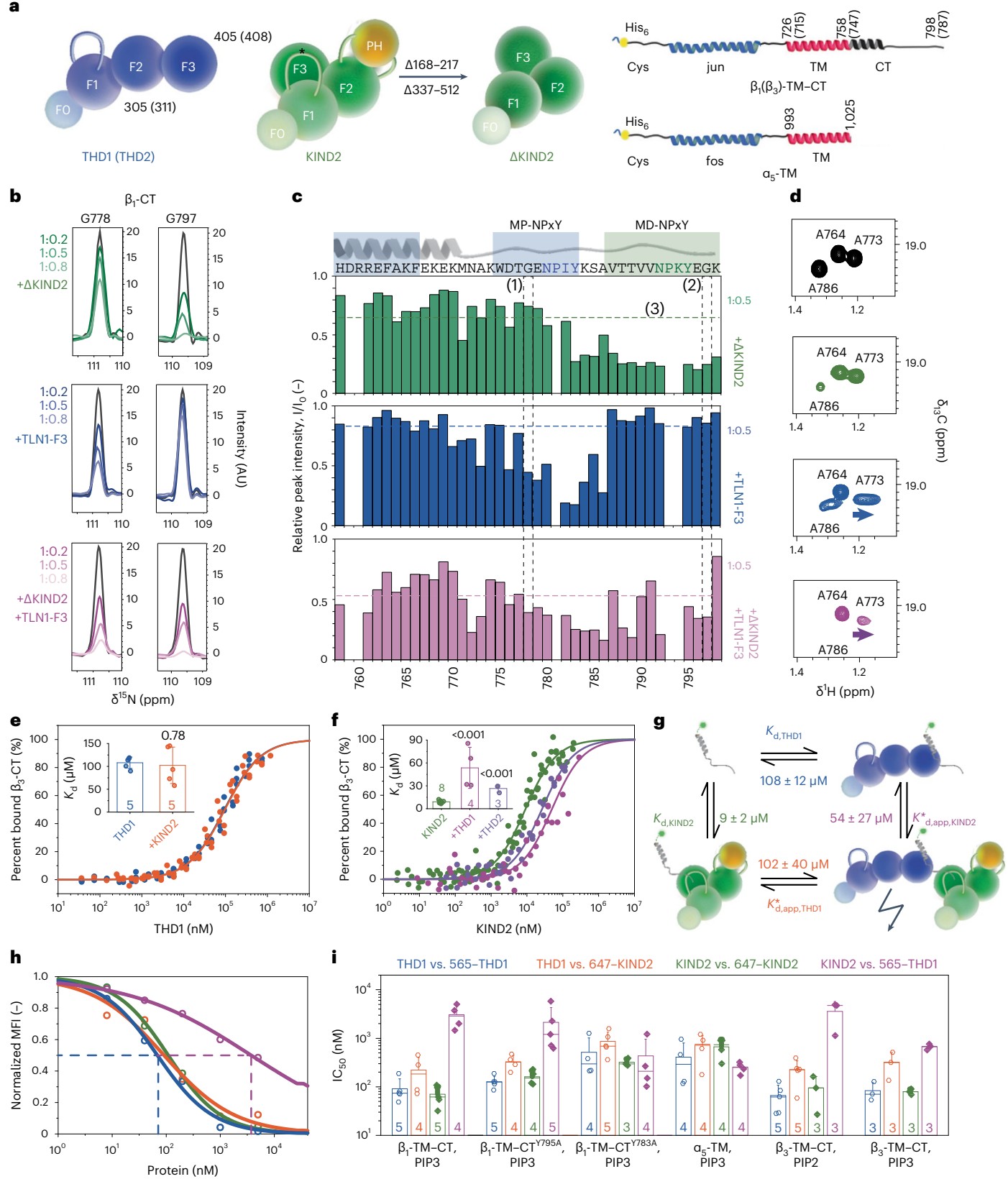

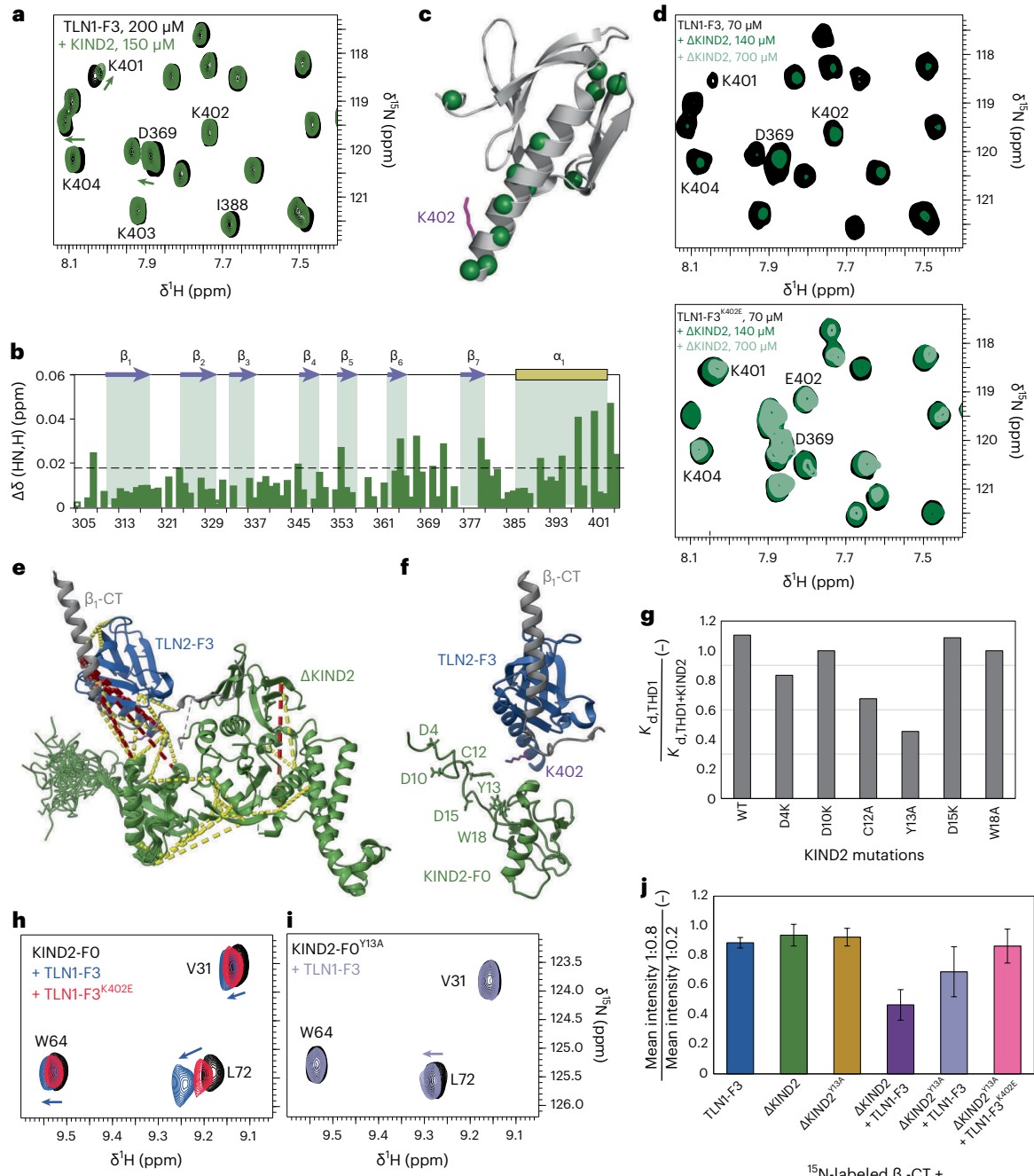

**Fig. 2 | Talin and kindlin directly interact. a,b,** Overlay of $^1$H–$^{15}$N HSQC NMR spectra (**a**) and CSP plot (**b**) of $^{15}$N-labeled TLN1-F3 in the absence (black) and the presence (green) of KIND2. **c,** Significant CSP changes (>0.018 ppm) are indicated by the dashed line ($2\sigma$ interval, $P < 0.05$) and mapped on the TLN2-F3 crystal structure (PDB 3G9W). Amino acid numbering is below the plots; TLN1$^{K402}$ is shown in purple. **d,** Overlay of $^1$H–$^{15}$N HSQC NMR spectra of 70 μM $^{15}$N-labeled wild-type TLN1-F3 (top) and TLN1-F3$^{K402E}$ (bottom) in the absence (black) and the presence of 140 μM (dark green) and 700 μM (light green) ΔKIND2. Most peaks in TLN1-F3 spectra disappear (top) and remain in TLN1-F3$^{K402E}$ upon ΔKIND2 titration (bottom). Note that contour line thickness is adjusted in both images to improve visibility. **e,** Model of the TLN-F3–$\beta_1$-CT–ΔKIND2 complex based on chemical cross-links identified by mass spectrometry (Extended Data Fig. 2l) and reported structures of TLN2-F2F3 (blue)–$\beta_{1D}$-CT (gray; PDB 39GW), ΔKIND2

(green)–$\beta_{1A}$-CT (gray; PDB 5XQ0) and KIND2-F0 (green; PDB 2LGX). Cross-links between relevant distance of 5–30 Å are colored in yellow; further cross-links are in red. **f,** Model indicating residues in KIND2 that were analyzed further. **g,** Affinity reduction of THD1 for 488–$\beta_3$-CT in the absence ($K_{d,THD1}$) and the presence of 30–40 μM KIND2 ($K_{d,THD1+KIND2}$) with indicated substitutions in KIND2 ($n = 2$), measured by MST. **h,i,** Magnified view of an overlay of $^1$H–$^{15}$N HSQC NMR spectra of 100 μM $^{15}$N-labeled KIND2-F0 in the absence (black) and the presence of 100 μM TLN1-F3 (blue) or 100 μM TLN1-F3$^{K402E}$ (red) (**h**) and $^1$H–$^{15}$N HSQC NMR spectra of 70 μM $^{15}$N-labeled KIND2-F0$^{Y13A}$ in the absence (black) and the presence (purple) of 70 μM wild-type TLN1-F3 (**i**). **j,** Ratio of average mean peak intensities (±s.d.) assigned to residues H758–K770 in the α-helical region of $^{15}$N-labeled $\beta_1$-CT at 1:0.8 and 1:0.2 ratios of TLN1-F3 and/or ΔKIND2 (raw data in Fig. 1c and Extended Data Fig. 2l).

(Extended Data Fig. 2h), only K401, K402 or K403, double substitutions of KK402/403 or deletions of amino acids 401–405 in THD1 lowered the affinity for $\beta_3$-CTs in the presence of KIND2. A comparison of NMR

spectra of $^{15}$N-labeled TLN1-F3 and TLN1-F3$^{K402E}$ in the presence of ΔKIND2 confirmed the involvement of TLN1-F3$^{K402E}$ for KIND2 binding. Almost all NMR signals in the wild-type TLN1-F3 spectra experience

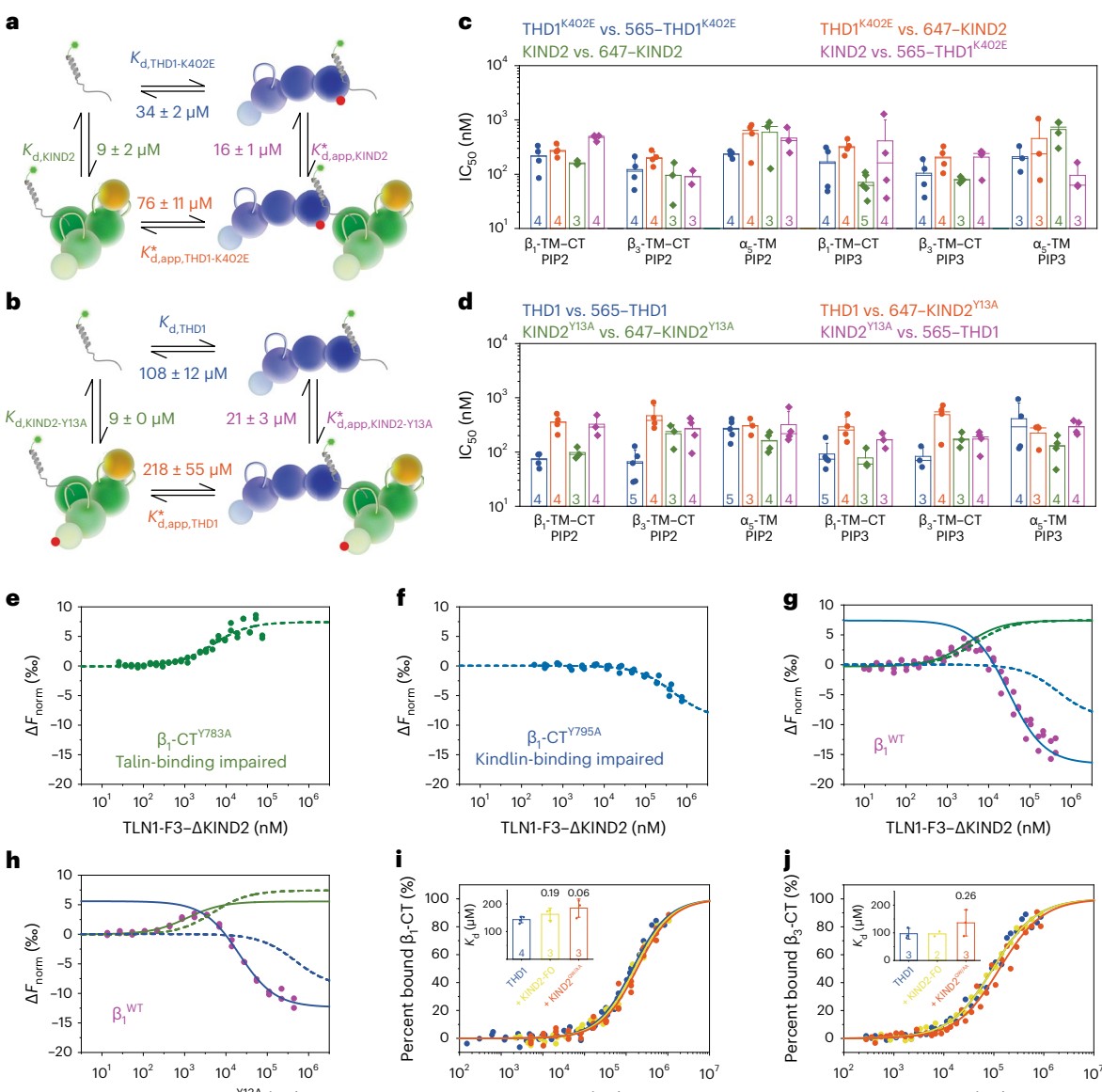

**Fig. 3 | Talin–integrin affinity increases upon kindlin binding to the $\beta_1$-CT.**
**a,b**, Schemes of MST-derived affinities of ternary-complex formation of THD1[K402E] and KIND2 (**a**) and THD1 and KIND2[Y13A] (**b**) with the $\beta_3$-CT at thermodynamic equilibrium. **c,d**, FC-RDA-derived IC$_{50}$ values of unlabeled THD1[K402E] and KIND2 competing for binding of 565–THD1[K402E] (blue and purple) and 647–KIND2 (orange and green) (**c**) or unlabeled THD1 and KIND2[Y13A] competing for binding of wild-type 565–THD1 (blue and purple) or 647–KIND2[Y13A] (orange and green) (**d**) to $\beta_1$-TM-CT, $\beta_3$-TM–CT and $\alpha_5$-TM embedded in PIP2- or PIP3-containing nanodiscs. **e–h**, MST affinity measurements of the TLN1-F3–ΔKIND2 fusion protein for 488–$\beta_1$-CT[Y783A] (**e**), $\beta_1$-CT[Y795A] (**f**) and $\beta_1$-CT (**g**) and of the TLN1-F3–ΔKIND2[Y13A] fusion protein for 488–$\beta_1$-CT (**h**). Data in **e,f** were fitted globally using a one-site-binding

model resulting in $K_d$ = 4.8 ± 1.1 μM for the talin-binding-deficient $\beta_1$-CT[Y783A] (kindlin contribution, green line) and $K_d$ = 470 ± 140 μM for the kindlin-binding-deficient $\beta_1$-CT[Y795A] (talin contribution, blue line) and are shown for comparison as dashed lines in **g,h**. Data in **g,h** were fitted with two one-site-binding fits, setting the maximum response to the response fitted in **e**. Concentrations up to 5 μM were fitted with $K_d$ = 2.9 ± 0.4 μM (**g,h**) (kindlin contribution, solid green line), whereas concentrations above 5 μM were fitted with $K_d$ = 25 ± 3 μM (**g**) and $K_d$ = 16 ± 1 μM (**h**) (talin contribution, solid blue line). **i,j**, MST affinity measurements of THD1 for 488–$\beta_1$-CT (**i**) and 488–$\beta_3$-CT (**j**) in the absence (blue) and the presence of 500–1,100 μM KIND2-F0 (yellow) or 300–400 μM $\beta$-CT-binding-impaired KIND2[Q614A,W615A] (QW/AA; orange). Data represent mean ± s.d.

severe line broadening beyond detection in the presence of ΔKIND2 at a molar excess of 2–10-fold. Under these conditions, the substantial molecular weight increase of the TLN1-F3–ΔKIND2 complex (from 11.6 kDa to 65.1 kDa) and potential dynamics in the binding interface led to substantial line broadening beyond detection. However, NMR signals in the TLN1-F3[K402E] spectra remain largely unaffected upon ΔKIND2 titration, consistent with the substantially reduced binding of TLN1-F3[K402E] to ΔKIND2 (Fig. 2d).

To define the corresponding talin-binding site on KIND2, we fused the C terminus of TLN1-F3 via a flexible glycine–serine (GS) linker to the N terminus of $\beta_1$-CT to produce recombinant TLN1-F3–$\beta_1$-CT or TLN1-F3[K402E]–$\beta_1$-CT fusion proteins with increased talin–integrin

binding events. The fusion proteins were cross-linked with disuccinimidyl suberate in the presence of excess KIND2, and cross-links were identified by mass spectrometry. We found numerous cross-links between TLN1-F3 and the F0 domain of KIND2 (KIND2-F0) and very few between TLN1-F3[K402E] and KIND2-F0 (Extended Data Fig. 2i). Next, we used the cross-links to build a model of a ternary TLN2-F3–$\beta_1$-CT–ΔKIND2 complex derived from published crystal structures (Fig. 2e) and noticed that the flexible N-terminal loop of KIND2 may be close to the $\alpha_1$-helix of TLN1-F3 (Fig. 2f). As four lysine residues in the TLN1-F3 $\alpha_1$-helix comprise the binding site for KIND2-F0 (Extended Data Fig. 2h), we expected either negatively charged or aromatic residues on the KIND2-F0 countersurface as the contact site (Fig. 2f).

Substitution of the positively charged and aromatic residues in the flexible N terminus of the F0 domain of recombinant KIND2 for lysine and alanine, respectively, revealed that only KIND2$^{Y13A}$ decreased THD1 affinity for 488-β$_3$-CT in MST measurements (Fig. 2g). To confirm the interaction between TLN1$^{K402}$ and KIND2$^{Y13}$, we titrated TLN1-F3 or TLN1-F3$^{K402E}$ with the $^{15}$N-labeled KIND2-F0 domain, recorded $^1$H–$^{15}$N-HSQC NMR spectra and observed that TLN1-F3 induces stronger CSPs in the KIND2-F0 domain than TLN1-F3$^{K402E}$ (Fig. 2h and Extended Data Fig. 2j–m). Furthermore, almost all CSPs induced by TLN1-F3 titration with $^{15}$N-labeled KIND2-F0 drastically decreased when TLN1-F3 was titrated with $^{15}$N-labeled KIND2-F0$^{Y13A}$ (Fig. 2i and Extended Data Fig. 2k), confirming that KIND2-F0$^{Y13A}$ lost the ability to interact with TLN1-F3.

To validate whether disrupting the talin–kindlin interaction influences binding to $^{15}$N-labeled β$_1$-CT, we analyzed amide signal line widths in $^1$H–$^{15}$N correlation spectra. The signals assigned to the α-helical region of β$_1$-CT (from H758 to K770) showed little line broadening (intensity reduction of less than 10%) upon addition of TLN1-F3, ΔKIND2 or ΔKIND2$^{Y13A}$ at a molar stoichiometry of 1:0.8 compared to 1:0.2 (Fig. 2j), confirming that neither protein interacts with the α-helical region of the β$_1$-CT (Fig. 1c and Extended Data Fig. 2n). By sharp contrast, signal intensities in the membrane-proximal α-helix decreased by 50% upon addition of TLN1-F3 together with ΔKIND2, by about 30% upon addition of TLN1-F3 together with ΔKIND2$^{Y13A}$ and by about 10% upon addition of TLN1-F3$^{K402E}$ together with ΔKIND2$^{Y13A}$ (Fig. 2j), indicating that disrupting the talin–kindlin interaction decreases but does not inhibit ternary interactions with β$_1$-CT.

## Talin and kindlin compete for β-CT without direct binding

The ability of KIND2$^{Y13A}$ to decrease ternary TLN1–β-CT–KIND2 complex formation assigns an important function to the talin–kindlin interaction that was tested with recombinant proteins in different experiments. First, we excluded structural changes in recombinant talin and kindlin that harbor the K402E or Y13A substitutions (Extended Data Fig. 3a–j). Subsequently, we measured their dissociation constants using MST, which revealed that THD1$^{K402E}$ was less potent ($K_{d,app,KIND2}$* = 16 ± 1 μM; Fig. 3a and Extended Data Fig. 3k) than THD1 in decreasing KIND2 affinity for β$_3$-CT ($K_{d,app,KIND2}$* = 54 ± 27 μM; Fig. 1f,g) and that THD1 was less potent in decreasing KIND2$^{Y13A}$ affinity than KIND2 affinity for β$_3$-CT ($K_{d,app,KIND2-Y13A}$* = 21 ± 3 μM; Fig. 3b and Extended Data Fig. 3l), indicating that the talin–kindlin interaction potentiates the decrease in kindlin affinity for the β$_3$-CT. We also found that the KIND2$^{Y13A}$-substituted protein gained the ability to decrease the affinity of THD1 for β$_3$-CT ($K_{d,THD1}$ = 105 ± 20 μM; $K_{d,app,THD1}$* = 218 ± 55 μM; Fig. 3b and Extended Data Fig. 3n), indicating that the talin–kindlin interaction increases TLN1 affinity for the β$_3$-CT, which, probably due to the low population of ternary talin–β-CT–kindlin complexes, escapes detection in MST measurements with wild-type THD1 and KIND2. Interestingly, insertions of the dissociation constants measured for THD1$^{K402E}$, KIND2 and the β$_3$-CT in the energy square produced a value of approximately 1, which agrees with an allosteric ternary-complex model (Fig. 3a):

$$\frac{1}{34 \pm 2\,\mu M} \times \frac{1}{16 \pm 1\,\mu M} \times (76 \pm 11\,\mu M) \times (9 \pm 2\,\mu M) = 1.3 \pm 0.4$$

as does THD1, KIND2$^{Y13A}$ and the β$_3$-CT (Fig. 3b):

$$\frac{1}{108 \pm 12\,\mu M} \times \frac{1}{21 \pm 3\,\mu M} \times (218 \pm 55\,\mu M)$$
$$\times (9 \pm 0\,\mu M) = 0.9 \pm 0.3$$

The ability of KIND2 to compete with THD1$^{K402E}$ and the ability of KIND2$^{Y13A}$ to compete with THD1 for β-CT binding were confirmed by FC-RDA with β$_1$-TM–CT and β$_3$-TM–CT containing PIP2- or

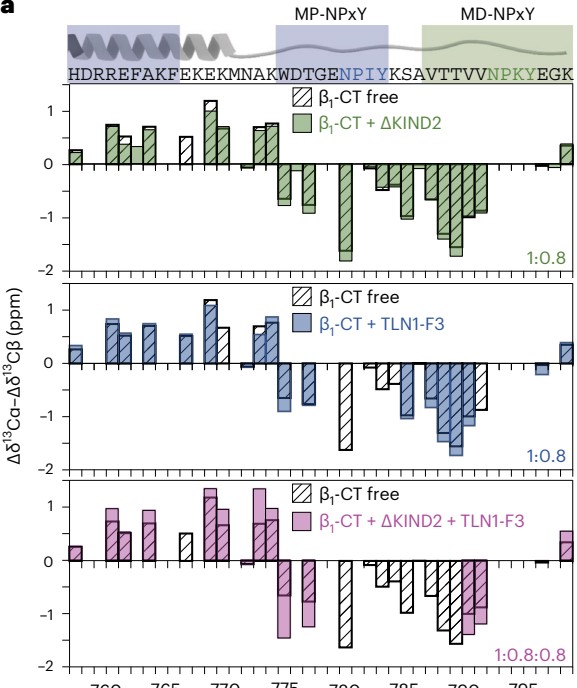

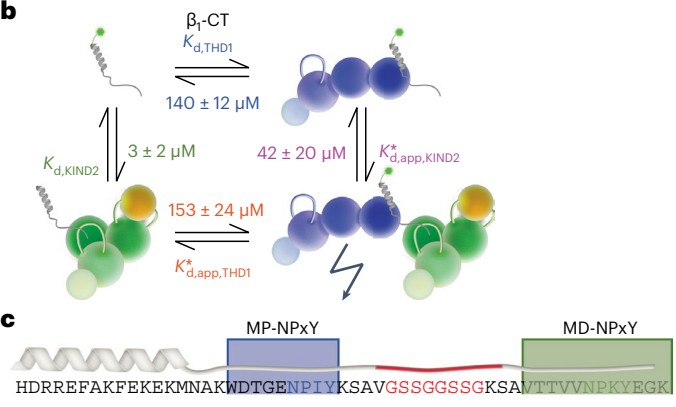

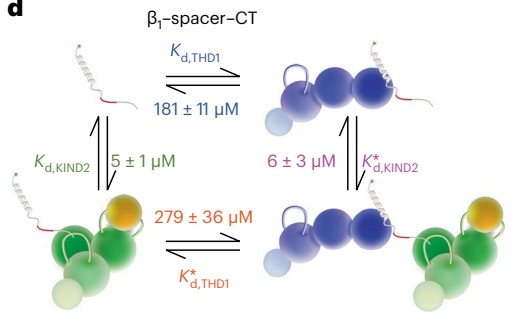

**Fig. 4 | Talin- and kindlin-binding sites on the β$_1$-CT are allosterically coupled. a**, Secondary chemical shifts (Δδ$^{13}$Cα–Δδ$^{13}$Cβ) of $^2$H,$^{13}$C,$^{15}$N-labeled β$_1$-CT in the absence (black, patterned) and the presence of partially deuterated ΔKIND2 (green), TLN1-F3 (blue) or ΔKIND2 and TLN1-F3 together (purple). Amino acid numbering is shown below plots. **b**, MST-derived binding affinities of THD1 and/or KIND2 for the β$_1$-CT at thermodynamic equilibrium. The measured affinities are inconsistent with a simple ternary-complex model. **c**, Scheme of the β$_1$–spacer–CT mutant. **d**, MST-derived binding affinities of THD1 and/or KIND2 for β$_1$–spacer–CT at thermodynamic equilibrium. The observed affinities are consistent with a simple ternary-complex model.

PIP3-supplemented nanodiscs (Fig. 3c,d). These data indicate that disruption of the talin–kindlin interaction leads to bidirectional competition of THD1 and KIND2 for β-CTs.

The low population of the ternary talin–β-CT–kindlin complexes, which further decreases in the absence of direct talin–kindlin interaction, impedes functional analysis of the ternary complex. To increase talin–kindlin binding events and thereby talin–β-CT–kindlin complex formation, we fused the C terminus of TLN1-F3 via a flexible GS linker to the N terminus of ΔKIND2 to produce the TLN1-F3–ΔKIND2 fusion protein and determined affinities for wild-type and mutant 488–β₁-CTs. The affinities of TLN1-F3–ΔKIND2 for talin-binding-impaired β₁-CT$^{Y783A}$ and kindlin-binding-impaired β₁-CT$^{Y795A}$ were in good agreement with titrations of only KIND2 or only THD1 with β₁-CT (Fig. 3e,f, Extended Data Fig. 1f,g and Extended Data Table 1). Furthermore, measurements revealed a positive, upward shift upon TLN1-F3–ΔKIND2 binding to β₁-CT$^{Y783A}$ (positive ΔF$_{norm}$ values; Fig. 3e) and a negative, downward shift upon TLN1-F3–ΔKIND2 binding to β₁-CT$^{Y795A}$ (negative ΔF$_{norm}$ values; Fig. 3f) in normalized MST traces, which we also observed in titrations of KIND2 and THD1 with β₁-CT (Extended Data Fig. 1e,g). The opposing MST trends allowed us to distinguish the contributions of the talin–integrin and the kindlin–integrin binding site in titration experiments with the TLN1-F3–ΔKIND2 fusion protein with wild-type (WT) β₁-CT (Fig. 3g). The experiment revealed that the titration of TLN1-F3–ΔKIND2 with β₁-CT or β₁-CT$^{Y783A}$ produced the same kindlin-mediated upward shift in MST data and similar affinities ($K_{d,\beta1\text{-}CT\text{-}WT}$ = 2.9 ± 0.4 μM; $K_{d,\beta1\text{-}CT\text{-}Y783A}$ = 4.8 ± 1.1 μM; Fig. 3g). Of note, TLN1-F3-mediated ΔKIND2 competition remains invisible in titration experiments due to equimolar talin and kindlin concentrations in the TLN1-F3–ΔKIND2 fusion protein and the lower affinity of talin for β₁-CT than the affinity of kindlin for β₁-CT. In sharp contrast to the unaffected kindlin-mediated upward shift, the talin-mediated downward shift became strikingly steeper in the titration experiments, producing an 18-fold increase in affinity for β₁-CT compared to β₁-CT$^{Y795A}$ ($K_{d,\beta1\text{-}CT\text{-}WT}$ = 25 ± 3 μM; $K_{d,\beta1\text{-}CT\text{-}Y795A}$ = 470 ± 140 μM; Fig. 3g). These data strongly indicate that kindlin is capable of increasing talin affinity for β₁-CT when they both bind to the β₁-CT.

To determine whether the association with kindlin induces allosteric activation of talin, followed by increased talin affinity for the β₁-CT during ternary-complex formation, we titrated the TLN1-F3–ΔKIND2$^{Y13A}$ fusion protein with β₁-CT (Fig. 3h). We observed identical binding curves as in titration experiments with the wild-type TLN1-F3–ΔKIND2 fusion protein binding to β₁-CT (Fig. 3g,h). This result was confirmed in MST affinity measurements, which showed that THD1 affinity for β-CTs remained unchanged in the presence of β-CT-binding-deficient KIND2$^{Q614A,W615A}$ or KIND2-F0 (Fig. 3i,j), indicating that the association of TLN1 and KIND2 increases the population of the ternary talin–β-CT–kindlin complex but not THD1 affinity for β-CTs in the ternary complex.

### Ternary-complex formation involves allostery in the β-CT

Because direct talin–kindlin interaction is not involved in the kindlin-mediated increase in talin affinity, we tested whether kindlin binding to the β-CT induces conformational changes that, in turn, increase β-CT affinity for talin. To test this hypothesis, we produced $^2$H,$^{13}$C,$^{15}$N-labeled β₁-CT and partially deuterated TLN1-F3 and ΔKIND2

to record three-dimensional HNCACB spectra of β₁-CT in the presence and the absence of 1:0.8 TLN1-F3 and/or ΔKIND2. From these spectra, we derived secondary $^{13}$Cα and $^{13}$Cβ chemical shifts (Δδ) that are indicative of secondary structure. The calculated Δδ$^{13}$Cα−Δδ$^{13}$Cβ values (Fig. 4a) are around 0 for disordered regions and are positive for α-helices and negative for β-sheets, with a maximum value of ~6 ppm indicating 100% secondary structure population[23]. In line with published structures[4,11,24], $^{13}$C secondary chemical shifts of β₁-CT assigned to membrane-proximal α-helical region were positive and regions around the KIND2-binding site indicate β-strand conformation, reflected by negative values. However, values of around |1| indicate only partial folding of the β₁-CT in solution (Fig. 4a). Whereas addition of TLN1-F3 or ΔKIND2 alone induced minor changes, addition of both TLN1-F3 and ΔKIND2 induced $^{13}$C secondary chemical shifts pointing to increased α-helical conformation in the region between H758 and K774 and more extended β-type conformation in the region between W775 and V791. Compared to its unbound conformation, β₁-CT is more structured in the presence of TLN1-F3 and ΔKIND2, suggesting that the ternary complex with talin and kindlin changes or selects a specific β₁-CT conformation to enhance talin binding.

To test the importance of allosteric coupling of the talin- and kindlin-binding sites (Fig. 4b), we separated the two binding sites by duplicating the intervening sequence with a ten-amino acid spacer (termed β₁–spacer–CT; Fig. 4c) and found that neither THD1 changed KIND2 affinity nor KIND2 changed THD1 affinity for β₁–spacer–CT (Fig. 4d and Extended Data Fig. 4a,b) and that the calculated dissociation constants complied with the ternary-complex model:

$$\frac{1}{181 \pm 11\,\mu M} \times \frac{1}{6 \pm 3\,\mu M} \times (279 \pm 36\,\mu M) \times (5 \pm 1\,\mu M) = 1.3 \pm 0.7$$

The independent binding of talin and kindlin to β₁–spacer–CT increases ternary-complex formation, however, without inducing the structural conformational change in the integrin CT that is required for the function of the ternary talin–β-CT–kindlin complex.

### Talin–kindlin cooperation is essential for cell adhesion

To test whether talin–kindlin interaction-mediated regulation of the population of ternary talin–β-integrin–kindlin complexes affects integrin function in cells, we retrovirally transduced mouse kidney fibroblasts lacking expression of TLN1, TLN2, KIND1 and KIND2 (ref. 1) (quadruple knockout (qKO)) to express N-terminally mCherry-tagged wild-type KIND2 or KIND2$^{Y13A}$ and C-terminally YPet-tagged wild-type TLN1 or TLN1$^{K402E}$ and subsequently sorted cell populations with similar TLN1–YPet and mCherry–KIND2 protein levels (qKO-TLN1$^{WT}$KIND2$^{WT}$, qKO-TLN1$^{K402E}$KIND2$^{WT}$, qKO-TLN1$^{WT}$KIND2$^{Y13A}$ and qKO-TLN1$^{K402E}$KIND2$^{Y13A}$; Extended Data Fig. 5a–d) using flow cytometry. Whereas the total β₁-integrin surface levels were alike on qKO-TLN1$^{WT}$KIND2$^{WT}$ and qKO-TLN1$^{K402E}$KIND2$^{WT}$ cells (Extended Data Fig. 5d), β₁-integrin-activation-associated epitope 9EG7 levels were reduced on qKO-TLN1$^{K402E}$KIND2$^{WT}$ cells before and after Mn$^{2+}$ treatment

**Fig. 5 | Interaction between TLN1 and KIND2 is required for integrin functions. a**, Binding of the β₁-integrin-activation-reporting 9EG7 antibody in the presence of magnesium (white bars) or manganese (patterned bars) normalized to total β₁-integrin levels. Dots represent experiments on different days (qKO, $n$ = 5; dKO, $n$ = 3; one-sample or two-sample $t$-test). **b**, Total internal reflection fluorescence (TIRF) microscopy images used for focal adhesion analysis of the indicated, serum-starved cells spread for 40 min on FN. Nuclei were stained with 4,6-diamidino-2-phenylindole (DAPI) (blue). Scale bar, 10 μm. **c**, Average focal adhesion count and area per cell ($n$ = 4 independent experiments; >8 cells per condition; one-way ANOVA of repeated measurements with Tukey's post hoc test) quantified from the mCherry–KIND2 signal. **d,e**, qKO-TLN1$^{WT}$KIND2$^{WT}$ (black) and qKO-TLN1$^{K402E}$KIND2$^{WT}$ (blue) cells (**d**) and dKO-TLN1$^{WT}$ (gray) and dKO-TLN1$^{K402E}$ (red) cells (**e**) attached to a concanavalin

A-coated cantilever brought in contact with FNIII7–10 for the indicated contact times and separated with a loading rate of 5 μm s$^{-1}$. Dots represent separation forces of single fibroblasts, and lines indicate the median value. $P$ values were calculated using two-tailed Mann–Whitney test and represent comparisons between qKO-TLN1$^{WT}$KIND2$^{WT}$ and qKO-TLN1$^{K402E}$KIND2$^{WT}$ cells or dKO-TLN1$^{WT}$ and dKO-TLN1$^{K402E}$ cells at the indicated contact times. **f,g**, Quantification of cell area from the TLN1–YPet signal 16 h after seeding serum-starved cells on FN, VN or laminin-111 (**f**) (laminin-111 (LN111), $n$ = 3 independent experiments, >50 cells per condition) or FN-coated hydrogels with 4-, 12- or 50-kPa rigidity (**g**) ($n$ = 3 independent experiments, >50 cells per condition analyzed, one-way ANOVA of repeated measurements with Tukey's post hoc test). For details on statistical methods and data representation, see Methods.

(Fig. 5a). In line with the reduced $\beta_1$-integrin activity, qKO-TLN1$^{K402E}$-KIND2$^{WT}$ cells produced fewer but on average larger adhesion sites, quantified by the mCherry–KIND2 signal (Fig. 5b,c), and showed slower adhesion strengthening, quantified by atomic force microscopy-based single-cell force spectroscopy (SCFS)[25,26] (Fig. 5d,e), and impaired spreading on fibronectin (FN), laminin-111 and vitronectin (VN; Fig. 5f).

Whereas qKO-TLN1$^{WT}$KIND2$^{WT}$ cells responded with increasing cell size to increasing rigidity of FN-coated hydrogels, qKO-TLN1$^{K402E}$-KIND2$^{WT}$ cells retained their size on FN-coated hydrogels of 4-kPa, 12-kPa and 50-kPa rigidity (Fig. 5g). Importantly, qKO-TLN1$^{WT}$KIND2$^{Y13A}$, qKO-TLN1$^{K402E}$KIND2$^{WT}$ and qKO-TLN1$^{K402E}$KIND2$^{Y13A}$ cells displayed a similar phenotype. These findings were confirmed with an independent

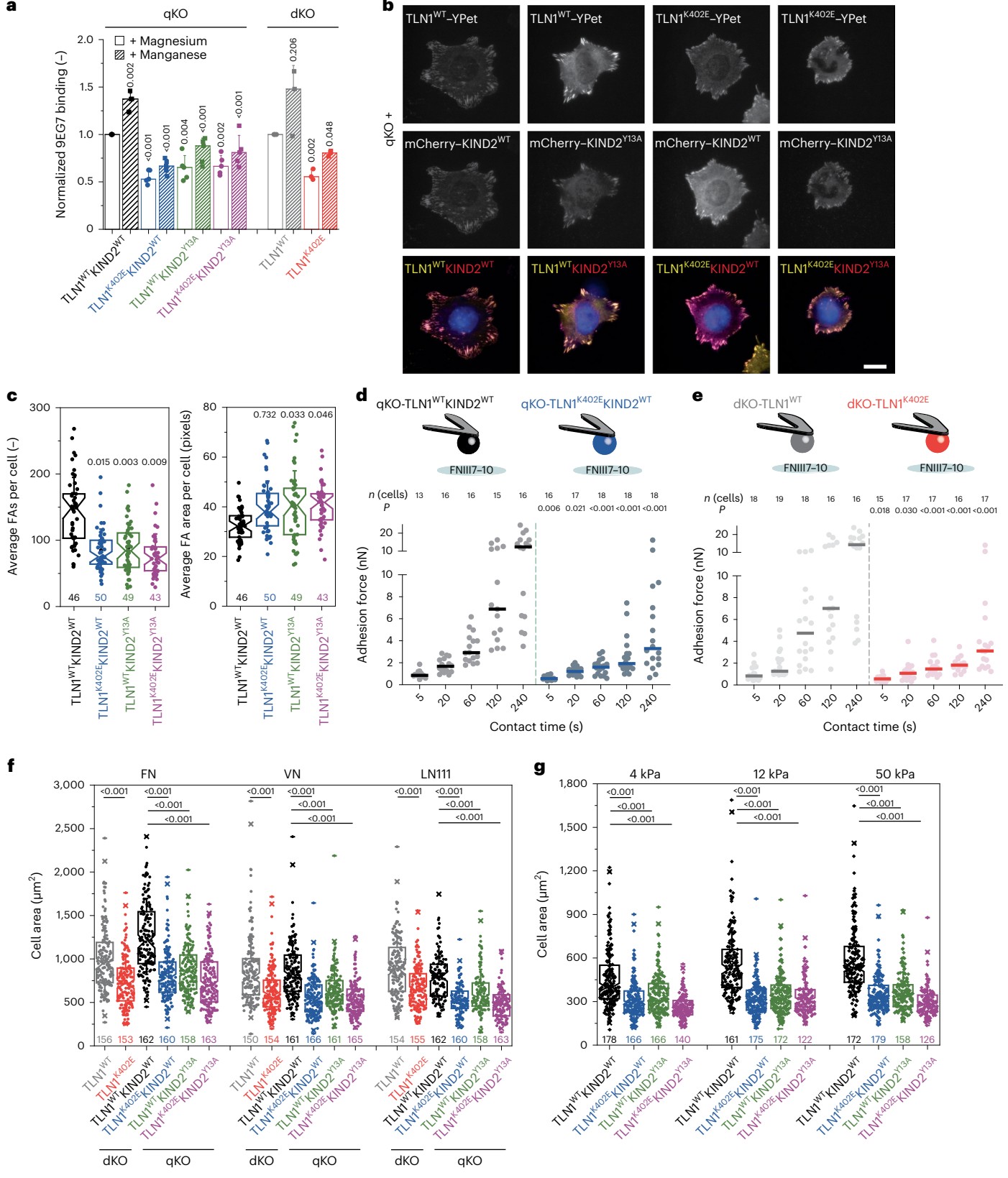

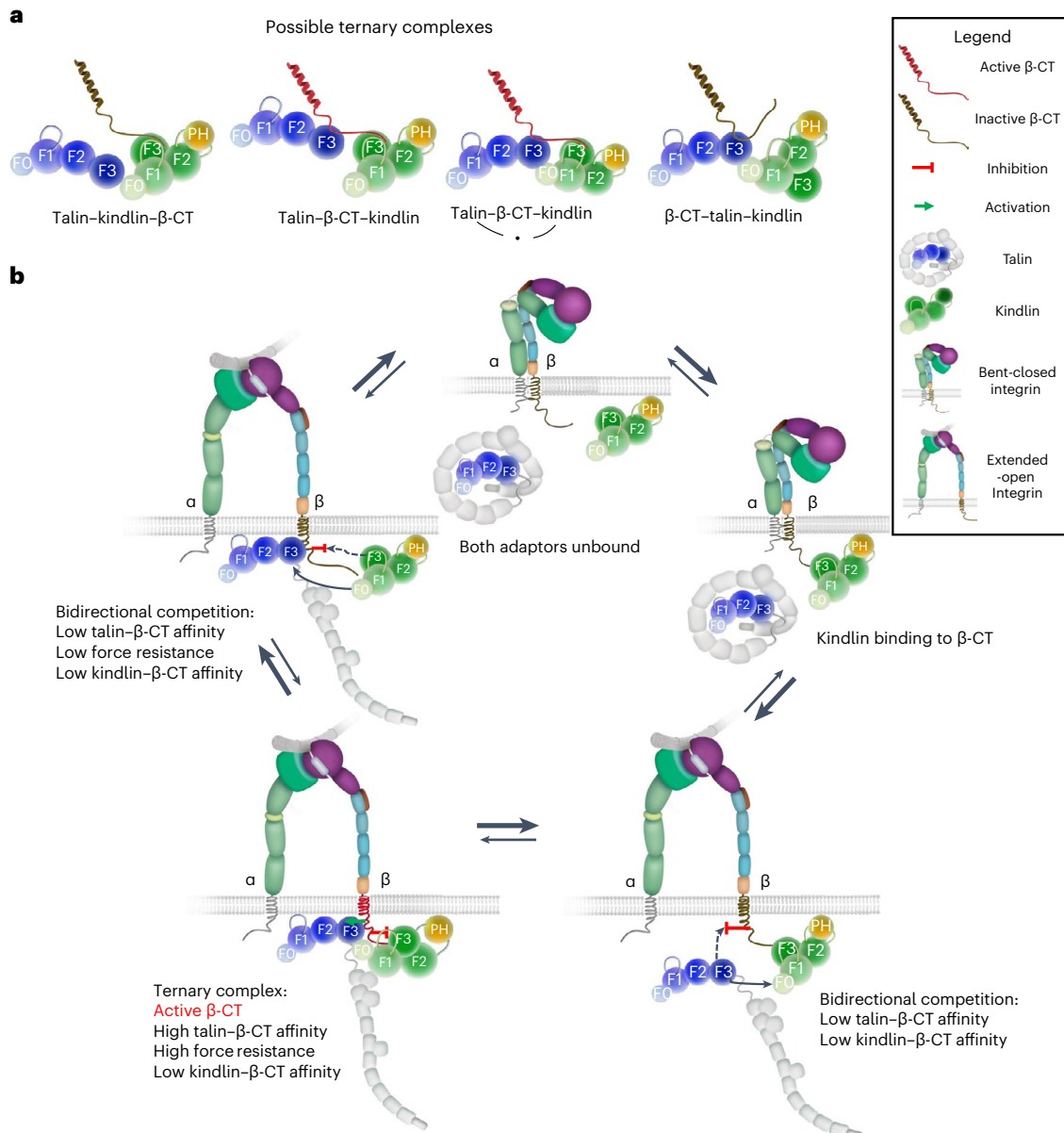

**Fig. 6 | Model of ternary talin–β-CT–kindlin complex assembly and disassembly. a**, Ternary complexes forming between talin, β-CT and kindlin in the presence or the absence of talin–kindlin binding. Because talin and kindlin compete for the inactive β-CT conformation, direct talin–kindlin interaction may help to overcome low talin affinity for the kindlin-occupied β-CT during ternary talin–β-CT–kindlin complex assembly. **b**, Talin–kindlin cooperativity involves a switch from an inactive to an active β-CT conformation. Kindlin interacts with inactive (bent-closed) and active (extended-open) integrins. Binding of kindlin to the extended-open integrin activates the β-CT, leading to high talin affinity and high resistance to actomyosin pulling forces. High talin affinity in turn decreases kindlin affinity for the β-CT, resulting in kindlin dissociation from the β-CT, β-CT inactivation, a decrease in talin affinity and the integrin bent-closed state. Talin–kindlin interaction at the β-CT maintains a low population of ternary talin–β-CT–kindlin complexes by helping to overcome low talin affinity for the kindlin-occupied β-CT until the active β-CT conformation is induced and the ternary complex is assembled. The low population of ternary complexes in focal adhesions is able to transmit high forces, while the remaining integrins, occupied by only talin or kindlin, transmit low forces.

cell line[3] expressing KIND2 from the endogenous *Fermt2* locus, lacking expression of TLN1 and TLN2, and the defect was rescued with YPet-tagged wild-type TLN1 or TLN1[K402E] (double knockout (dKO)-TLN1; Extended Data Fig. 5a,c–f).

To test the function of an integrin tail with decoupled talin- and kindlin-binding sites, we transduced $\beta_1$-integrin complementary DNA encoding wild-type $\beta_1$ or $\beta_1$–spacer into $\beta_1$-knockout fibroblasts (Extended Data Fig. 5g). Whereas flow cytometry revealed similar surface levels of total $\beta_1$-integrin, cells expressing the $\beta_1$–spacer displayed reduced levels of the integrin-activation-reporting epitope 9EG7 on their surface compared to cells expressing wild-type $\beta_1$ (Extended Data

Fig. 5h). Furthermore, FN-seeded $\beta_1$–spacer-expressing cells exhibited profound adhesion and spreading defects (Extended Data Fig. 5i,j), which altogether indicates that formation of functional talin–$\beta_1$-CT– kindlin complexes depends on the close proximity of talin–kindlin.

## Discussion

Our NMR data indicate that $\beta_1$-CT can assemble a ternary talin–$\beta_1$– kindlin complex. The ternary complex, which was shown to form also with the $\beta_3$-CT and the $\beta_2$-CT[14,27], emerges as a principal molecular setting that ensures cooperativity between talin and kindlin, required to induce and maintain the active conformation of integrins. We expected

that the two adaptor proteins augment each other's binding to the β-CT rather than compete or bind independently (noncompetitively) of each other[10]. However, affinity measurements at equilibrium in solution and in the presence of charged lipid membranes point to a unidirectional competition mechanism, in which talin outcompetes kindlin from $\beta_1$-CT or $\beta_3$-CT, while kindlin does not affect talin binding. This observation agrees with single-particle tracking microscopy of live cells assigning kindlin a shorter immobilization time than talin in focal adhesions of FN-seeded fibroblasts[28] and predicts three important consequences. First, the unidirectional competition leading to a drastic decrease in kindlin affinity for the β-CT indicates that the ternary talin–β-CT–kindlin complex is transient and rare. Second, unidirectional competition results from ternary talin–β-CT–kindlin complex formation. Third, assembly of the ternary talin–β-CT–kindlin complex is inconsistent with microscopic reversibility, suggesting that complex conformational change(s) are involved in the asymmetric binding behavior of talin and kindlin.

In search of such conformational changes, we first looked for a direct interaction between talin and kindlin that may influence their affinities for β-CTs. Although a direct interaction between talin and kindlin has not been reported thus far[14], we detected, by NMR and cross-linking mass spectrometry, an interaction between the C-terminal α-helix of the TLN1-F3 domain and the N-terminal F0 domain of KIND2. Mutational disruption of the interaction decreased the population of ternary talin–β-CT–kindlin complexes by shifting the unidirectional competition toward bidirectional competition between talin and kindlin, in which kindlin decreases the affinity of talin and talin decreases the affinity of kindlin for β-CTs, although the latter less potently when compared with the competition studies in which the talin–kindlin interaction is intact. These findings indicate that, upon assembly of the ternary talin–β-CT–kindlin complex, kindlin induces a major increase in affinity of talin for the $\beta_1$-CT and talin profoundly decreases kindlin affinity for the $\beta_1$-CT. The increase in talin affinity for the $\beta_1$-CT induced by kindlin could indeed be confirmed with a talin–kindlin construct connected by a flexible peptide linker.

Because β-CTs deficient for talin or kindlin binding excluded a role of direct talin–kindlin binding for inducing conformational changes in talin or kindlin that account for their asymmetric affinity behavior for the β-CT, we searched for a conformational communication of the talin- and kindlin-binding regions in the $\beta_1$-CT that changes a low talin-affinity β-CT (termed 'inactive' β-CT) to a high talin-affinity β-CT (termed 'active' β-CT) conformation (Fig. 6a). We found secondary structure changes in the $\beta_1$-CT region by NMR and could abolish allosteric coupling by introducing a spacer between the talin- and kindlin-binding sites that also abrogates asymmetric binding and curbs adhesion and spreading in cells. Interestingly, the population of ternary complexes is small, which is in line with single-molecule force measurements of RGD-bound $\alpha_v\beta_3$- and $\alpha_5\beta_1$-integrins in cells[29] that identified less than 10% of integrins in focal adhesions as transmitting high forces (exceeding 11 pN) and the remaining integrins, probably occupied only by either talin or kindlin, as transmitting low forces (below 11 pN).

THD1 is a FERM domain, and its crystal structure revealed both an atypical, linear and a canonical, cloverleaf-like conformation[5,30]. The cloverleaf-like conformation displays an interdomain interaction between D125 and E126 of TLN1-F1 and K401 and K402 of the TLN1-F3 domain, the latter of which is directly adjacent to the talin–kindlin interaction site identified in our study (Extended Data Fig. 6). It is conceivable that the linear and cloverleaf conformations of THD1 are in equilibrium, and a preference of KIND2 binding for the linear TLN1 conformation might increase membrane affinity via accessibility of additional membrane contacts by the TLN1-F0 and TLN1-F1 domains. This hypothesis is supported by our FC-RDA experiments, which showed that unidirectional talin-mediated kindlin competition is more pronounced from $\beta_1$-TM–CT- and $\beta_3$-TM–CT-containing nanodiscs than from β-CTs in solution. Hence, the interaction between talin and

kindlin may also promote talin activation by increasing talin affinity for the plasma membrane, whereas interactions occurring simultaneously between talin, β-CT and kindlin increase talin affinity for the β-CT.

Our data identify a complex mechanism underlying talin and kindlin cooperativity (Fig. 6b). Based on experimental evidence, we envision that talin–kindlin cooperativity commences with kindlin, which encounters the integrin before talin[31]. Because talin and kindlin compete for the inactive β-CT conformation, direct talin–kindlin interaction may help to overcome low talin affinity for the kindlin-occupied β-CT and keep talin close to the β-CT until the active β-CT conformation is induced. Binding of talin and kindlin induces the 'active' β-CT conformation, which increases talin affinity for the β-CT, reinforces talin–β-CT binding and enhances resistance to actomyosin pulling forces. The fascinating feature of the asymmetric binding behavior is that it is inherently autonomous: elevated talin affinity for β-CT results in a decrease in kindlin affinity for the active β-CT conformation, leading to kindlin unbinding, a decrease in talin affinity and eventually talin dissociation, integrin inactivation and initiation of a new cycle of ternary talin–β-CT–kindlin assembly.

## Online content

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

## Methods

### Cloning

Deletions, mutations and short insertions in complementary DNA were introduced by PCR using PfuUltra II (600670, Agilent Technologies) or Q5 Hot Start polymerase (M0493, New England Biolabs) according to the manufacturer's protocol, followed by DpnI (R0176, New England Biolabs) digestion of template DNA (2 h, 37 °C). After cleanup (QIAquick PCR Purification Kit, 28104, Qiagen), the PCR product was phosphorylated using T4 PNK (M0201, New England Biolabs), followed by another cleanup, ligated (Fast-Link DNA Ligation Kit, Epicenter) for 30 min at room temperature and transformed into *Escherichia coli* OmniMAX (Promega) or NEB 5-alpha competent *E. coli* (New England Biolabs).

Long insertions were introduced using the NEBuilder HiFi DNA Assembly Cloning Kit (New England Biolabs, E5520S) according to the manufacturer's protocol. The assembled constructs were transformed into *E. coli* OmniMAX or NEB 5-alpha competent *E. coli*.

DNA encoding the $\beta_{1A}$-CT was synthesized by Eurofins Genomics.

### Protein expression and purification

**Kindlin-2.** KIND2 and ΔKIND2 were expressed and purified as described earlier[1]. Briefly, kindlin constructs were cloned into pCoofy17 (ref. 32), which adds an N-terminal His$_{10}$–SUMO tag and expressed in soluble form in *E. coli* Rosetta cells at 18 °C overnight. After purification by immobilized metal chelate affinity chromatography (IMAC) in high-salt TBS buffer (20 mM Tris, pH 7.5, 500 mM NaCl, 1 mM Tris(2-carboxyethyl) phosphine (TCEP)), the SUMO tag was removed by SenP2 protease (obtained from the Max Planck Institute of biochemistry (MPIB) Core Facility) digest overnight, and the protein was further purified by SEC using TBS (20 mM Tris, pH 7.5, 200 mM NaCl, 1 mM TCEP) containing 5% glycerol as the running and storage buffer.

**Talin.** DNA for the THD and F3 domains of TLN1 (1–405 and 305–405) and TLN2 (1–408 and 311–408) was cloned into pCoofy17, which adds an N-terminal His$_{10}$–SUMO tag. After induction with isopropyl β-ᴅ-1-thiogalactopyranoside (IPTG) at an OD$_{600}$ of 0.6, proteins were expressed in *E. coli* (DE3) Rosetta 2 cells grown in TB, LB or M9 medium at 18 °C for 18–24 h. Cells were collected by centrifugation (6,000*g*, 20 min, 4 °C), resuspended in lysis buffer (20 mM Tris, pH 7.5, 500 mM NaCl, 1 mM TCEP, lysozyme, benzonase), disrupted by sonication (1-s pulses, 10 min, 90% amplitude) or high-pressure homogenization (Avestin Emulsiflex C3, ATA Scientific) and centrifuged (58,000*g*, 20 min, 4 °C), and the supernatant containing soluble proteins was filter sterilized (0.22 μm) and subjected to IMAC (HisTrap FF, GE Healthcare), followed by elution with a stepwise gradient. Fractions containing the protein (typical elution with 250–500 mM imidazole) were pooled, concentrated and rebuffered against TBS (20 mM Tris, pH 7.5, 200 mM NaCl) to remove imidazole before adding 40–80 μl SenP2 protease and incubating at 4 °C overnight. The cleaved His–SUMO tag, uncleaved protein and the SenP2 protease were removed by incubating the sample at 4 °C for 30 min while rotating with Ni-NTA beads. The sample was filtered (0.22 μm) and purified by SEC using a Superdex 75 10/300 GL, a Superdex 200 10/300 GL or a HiLoad Superdex 200 PG column (GE Healthcare), depending on molecular weight and purification scale.

**α-TM–CTs and β-TM–CTs.** Integrin TM–CTs were expressed and purified as described earlier with slight variations[33]. The β1-TM–CT (726–798), the β$_3$-TM–CT (715–787), the mutant β$_1$-TM–CT (W775A/ Y783A, T778A/T789A/Y795A) and α$_5$-TM (993–1,025) constructs were produced in *E. coli* BL21 (DE3) Rosetta 2 cells, leading to the formation of inclusion bodies. The cells were resuspended in lysis buffer (50 mM HEPES, 150 mM NaCl, 0.1% Triton X-100, pH 7.5, 1 mM TCEP, 1 mM PMSF, lysozyme, benzonase), disrupted by sonication (1-s pulses, 10 min, 90% amplitude) and centrifuged (58,000*g*, 20 min, 4 °C), and supernatant containing soluble proteins was discarded. The sediment was washed once with 50 mM HEPES, 150 mM NaCl, 0.1% Triton X-100, pH 7.5, followed by another centrifugation (58,000*g*, 20 min, 4 °C), and resuspended in 50 mM HEPES, 150 mM NaCl, 3% EMPIGEN (β-TM–CT) or 3% *n*-dodecyl-β-ᴅ-maltopyranoside (α$_5$-TM), 1 mM TCEP, pH 7.5 to solubilize the TM–CT constructs from inclusion bodies. The sample was incubated overnight on a rotating wheel at 4 °C, except for mutant β$_1$-TM–CTs, which were incubated for 2 d at 37 °C. After centrifugation (58,000*g*, 20 min, 4 °C), the supernatant was filter sterilized and purified by IMAC (HisTrap FF, GE Healthcare) and cation-exchange chromatography (HiTrap SP HP, GE Healthcare) at pH 5.8.

**Integrin-β cytoplasmic tails.** The integrin-β$_1$-CT (758–798) was cloned into pCoofy17, which adds an N-terminal His$_{10}$-SUMO tag. After induction with IPTG at an OD$_{600}$ of 0.6, proteins were expressed in *E. coli* (DE3) Rosetta 2 cells in M9 medium at 18 °C for 36 h. Cells were collected by centrifugation (6,000*g*, 20 min, 4 °C), resuspended in lysis buffer (20 mM Tris, pH 7.5, 500 mM NaCl, 1 mM TCEP, lysozyme, benzonase), disrupted by sonication (1-s pulses, 10 min, 90% amplitude) and centrifuged (58,000*g*, 20 min, 4 °C); the supernatant containing soluble proteins was filter sterilized (0.22 μm) and subjected to IMAC (HisTrap FF, GE Healthcare), followed by elution with a stepwise gradient. Fractions containing the protein (typical elution with 250–500 mM imidazole) were pooled, diluted with high-salt TBS buffer and concentrated by another IMAC (HisTrap FF, GE Healthcare) with a single elution step using high-salt TBS buffer with 500 mM imidazole. The resulting elution fraction was diluted tenfold with 20 mM Tris-HCl, pH 7.0, before adding 80 μl SenP2 protease and incubating at 4 °C overnight. The cleaved His–SUMO tag, uncleaved protein and the SenP2 protease were removed by incubating the sample at 4 °C for 30 min while rotating with Ni-NTA beads. The sample was filtered (0.22 μm), and the pH value was adjusted to about 6 using 1 M MES, pH 5.8 and confirmed with pH paper. Next, the protein was loaded on a cation-exchange chromatography column (HiTrap SP, 5 ml, GE Healthcare), washed with 25 mM MES, pH 5.8, 40 mM sodium chloride buffer and eluted with TBS (20 mM Tris, pH 7.5, 200 mM sodium chloride, 1 mM TCEP). Purity and labeling efficiency with $^{13}$C and $^{15}$N isotopes were determined by mass spectrometry and were >98%.

After the final chromatography step, the purity, integrity and identity of recombinant KIND2 and talin proteins were controlled by SDS–PAGE, high-resolution mass spectrometry and dynamic light scattering (DLS). Integrin TM–CT peptides were controlled by SDS–PAGE and high-resolution mass spectrometry. Peptides were controlled by high-resolution mass spectrometry.

### Fluorescent labeling and biotinylation of proteins

Before labeling, proteins were transferred into a buffer suitable for the intended labeling reaction using desalting columns. Integrin CTs were labeled with ATTO 488 *N*-hydroxysuccinimide (NHS) at their N terminus during chemical synthesis (by the MPIB core service facility), integrin TM–CTs were labeled with biotin-maleimide, whereas kindlin and talin were labeled with ATTO 565 NHS or Alexa 647 NHS or maleimide. For maleimide labeling, the pH value was adjusted to 7.0–7.5 using 1 M Tris, pH 7.5, and cysteines were reduced by adding TCEP to a final concentration of 2 mM before the reaction. Next, thiol-reactive dye was added at a molar excess of 10–20× and incubated at room temperature for 2 h in the dark or at 4 °C overnight. For amino-reactive dyes, the protein was first transferred to NHS labeling buffer (PBS with 10 mM NaHCO$_3$, pH 9.0) and then mixed with dye at a molar excess of 2.5×. The labeling reaction was carried out for 1 h at room temperature or overnight on ice in the dark. Excessive dye was removed with desalting columns.

### Assembly of nanodiscs

Lipids were dissolved in chloroform (1,2-dimyristoyl-*sn*-glycero-3-phosphocholine (DMPC)), 20:9:1 chloroform–methanol–water (PIP2) or 80:40:1 chloroform–methanol–HCl (PIP3) at a final concentration of 50 mg ml$^{-1}$ (DMPC) or 10 mg ml$^{-1}$ PIP2 or PIP3, which yielded a final

concentration of 10% PIP2 and 90% DMPC or 10% PIP3 and 90% DMPC to obtain molar ratios of 1:9 PIP2–DMPC and 1:9 PIP3–DMPC, respectively. The desired volume of lipid solution was transferred to a clean glass tube using a Hamilton glass syringe, dried under an $N_2$ stream, followed by desiccation overnight, and dissolved in cholate buffer (20 mM Tris, pH 7.5, 100 mM sodium chloride, 100 mM sodium cholate) to yield a 50 mM lipid stock solution.

For nanodisc assembly, integrin TM–CT, MSP2N2 scaffold protein (obtained from the MPIB Biochemistry Core Facility) and lipid stock solution were mixed 1:1:330 to obtain on average one integrin TM–CT per nanodisc. Before adding lipids to the mixture, cholate buffer was added to obtain a final cholate concentration between 10 and 20 mM and to avoid precipitation of lipids. The samples were then dialyzed at least three times against 1 l of nanodisc buffer (20 mM HEPES, pH 7.5, 150 mM sodium chloride, 0.5 mM EDTA) at room temperature, filter sterilized and purified with a Superdex 200 Increase 10/300 GL (GE Healthcare) or an SEC 650 (Bio-Rad) column to separate assembled nanodiscs from non-assembled components. The elution fractions were analyzed by SDS–PAGE with silver staining and flow cytometry for talin and/or kindlin binding.

## Microscale thermophoresis measurements

All MST measurements were performed as published[34] on a Monolith NT.115 red–blue machine (NanoTemper) using premium coated capillaries to reduce nonspecific interaction of proteins with the glass surface. Both interaction partners (ligand and receptor) were transferred into MST buffer (20 mM Tris, pH 7.5, 200 mM sodium chloride, 1 mM TCEP, 0.05% Tween-20) to avoid artifacts derived from buffer mismatches. ATTO 488-labeled integrin-β-CTs (50–200 nM, synthesized by the MPIB Core Facility) were used as ligands. Measurements were carried out at 10–20% LED power and 20% and 40% MST power. Data were analyzed using MO.Affinity Analysis Software (NanoTemper) as shown in Extended Data Fig. 1e. MST figures display data from individual titrations that were pooled (circles) and fitted with a global one-site-binding curve (lines). Affinity data in the figure insets (bar charts), in Extended Data Table 1 and in the text are mean ± s.d. of the replicates, which are given as *n*.

## Flow cytometry-based reporter-displacement assay

Fluorescently labeled 565–THD1 or 647–KIND2 were dissolved at a final concentration of 100 nM and 50 nM, respectively, in 200 μl 20 mM Tris, pH 7.5, 200 mM sodium chloride containing 0.25 μl Dynabeads M-280 Streptavidin (11205D, Invitrogen). To this buffer, competitors, that is, unlabeled THD1 and KIND2, were titrated before adding 10 μl of biotinylated integrin α-TM–CT or β-TM–CT in nanodiscs and incubating the solution for 10 min at room temperature while rotating. The samples were then injected in an LSRFortessa X-20 flow cytometer (BD Biosciences), and data were analyzed with FlowJo 10 (FlowJo) and OriginPro 2019b (OriginLab).

For data analysis, beads were gated in FlowJo and used as the negative control without protein and nanodiscs. MFIs of the samples were determined and exported to OriginPro 2019b. Here, data were fitted to a dose–response curve, setting top and bottom asymptotes to the values measured for the positive and negative controls, respectively. The positive control was measured in the presence of nanodiscs but in the absence of competitor, while, in the negative control, competitor and nanodiscs were absent. We report $IC_{50}$ values of these fits of individual experiments with different nanodisc preparations, which were recorded on different days as mean ± s.d.

## Dynamic light scattering

Before conducting a DLS measurement, protein samples were centrifuged for 15 min at 21,000*g* and 4 °C. DLS measurements were performed in triplicate on a DynaPro NanoStar instrument (Wyatt) at 20 °C with laser power set to auto-attenuation, an acquisition time of 5 s and

15 acquisitions. The hydrodynamic radius of the particles in the sample was calculated with Dynamics software (Wyatt).

## Circular dichroism spectroscopy

CD spectra were acquired using a quartz cuvette with a path length of 1 cm and a sample volume of 300 μl in a Jasco J-715 spectropolarimeter. Before the measurements, a buffer reference and protein samples (concentration, 0.1 mg ml⁻¹) were prepared with PBS buffer. First, a CD spectrum of the buffer sample was acquired from 190 nm to 250 nm at 25 °C. Afterward, buffer was removed from the cuvette, and the protein sample was added and measured using the same parameters. The buffer reference spectrum was subtracted from the protein spectrum to eliminate effects caused by the buffer.

## Thermal stability

Thermal stability was measured on a Prometheus NT.48 (NanoTemper) using a temperature gradient from 20 to 95 °C with an increase in temperature of 1 °C min⁻¹ while measuring internal tryptophan fluorescence at $\lambda = 330$ nm.

## NMR spectroscopy

For triple-resonance experiments with the $\beta_1$-CT, M9 medium was supplemented with 0.5 mg ml⁻¹ [¹⁵N]ammonium chloride or with 2 mg ml⁻¹ [¹³C]glucose. All experiments were performed in NMR buffer consisting of 20 mM Tris, 200 mM NaCl, 1 mM DTT, pH 7.5 and 5–10% $D_2O$ for the NMR lock signal. For deuterated $\beta_1$-CT samples, cells were adapted to deuterated M9 minimal medium in steps of 0%, 50% and 80% $D_2O$ supplemented with 0.5 mg ml⁻¹ [¹⁵N]ammonium chloride and 2 mg ml⁻¹ [¹³C]glucose, whereas ΔK2- and T1-F3-expressing cells were grown in 67% $D_2O$ M9 minimal medium. All datasets were acquired from Bruker Avance III spectrometers at a proton frequency of 600–800 MHz equipped with triple-resonance cryoprobes using TopSpin 3.2–3.5 software. Data were processed with TopSpin or NMRPipe and analyzed using CcpNmr Analysis software. Sample concentrations ranged from 70 μM to 700 μM, with all NMR experiments carried out at 298 K.

Protein backbone resonance assignments of $\beta_1$-CT were obtained using 3D HNCO, HNcaCO, HNCACB and CBCAcoNH experiments. Assignments for $\beta_1$-CT methyl resonances were performed using 3D (H)C(CCO)NH and H(CCCO)NH. Titrations were performed by forming binary or ternary complexes of either ¹⁵N- or ¹³C,¹⁵N-isotope-labeled integrin β1-CT mixed with unlabeled TLN1-F3 and ΔKIND2 at the indicated stoichiometries. For each titration point, ¹H–¹³C constant time HSQC or ¹H–¹⁵N HSQC spectra were recorded.

For NMR titration measurements, wild-type TLN1-F3 and the TLN1-F3^K402E mutant were expressed in M9 medium supplemented with 0.5 mg ml⁻¹[¹⁵N]ammonium chloride. Backbone assignments for wild-type TLN1-F3 were transferred from a previously published study in the Biological Magnetic Resonance Data Bank (BMRB 7061). Titrations were performed by forming binary or ternary complexes of ¹⁵N-labeled wild-type TLN1-F3 or TLN1-F3^K402E protein with unlabeled $\beta_1$-CT or ΔKIND2 at the indicated stoichiometry. For KIND2-F0, assignments were transferred from the Biological Magnetic Resonance Data Bank (BMRB 30659). About 47% of the backbone amide chemical shifts could be transferred by comparing ¹H–¹⁵N correlation spectra. Other signals exhibited some chemical shift differences or could not be unambiguously identified (Supplementary Table 1). Titrations involving wild-type KIND2-F0 or KIND2-F0^Y13 were performed at the indicated stoichiometries. For all measurements, ¹H–¹⁵N HSQC spectra were recorded at each point of the titration, the chemical shift changes of amide resonances in the fast-exchange regime were measured, and the reported weighted-average values of ¹⁵N and ¹H chemical shift changes are given by equation (3):

$$\Delta\delta_{H,N} = \sqrt{\left((\Delta\delta_H)^2 + \frac{1}{6}(\Delta\delta_N)^2\right)} \qquad (3)$$

## Cross-linking of proteins

The TLN1-F3–β1-CT fusion protein was mixed with KIND2 at a molar excess of 7.5-fold and concentrated to a final concentration of 60 μM TLN1-F3−β1-CT and 450 μM KIND2 in 120 μl TBS. The protein mixture was rebuffered against PBS using a 0.5-ml Zeba Spin column according to the manufacturer's protocol and subjected to chemical cross-linking using the GraFix method as described earlier[35]. In brief, 5–20% sucrose gradients were generated in SW 40 ultracentrifuge tubes using a Gradient Master station (model IP, Biocomp) with 0.5 mM DSS in the heavy solution. Concentrated proteins were added to the tubes on top of the gradients, and the setup was centrifuged in an SW 40 Ti swing bucket rotor at 40,000 r.p.m. for 16 h (Beckman Coulter). After centrifugation, the gradients were fractionated on the Gradient Master station coupled to a Bio-Rad fraction collector, and fractions were analyzed for complex-containing fractions by SDS–PAGE and Coomassie staining. Fractions of interest were analyzed by mass spectroscopy.

## Cross-linking mass spectrometry

Cross-linked protein pellets were incubated in digestion buffer (1:1; 1% SDC, 40 mM CAA, 10 mM TCEP, 50 mM Tris) for 20 min at 37 °C and then diluted with water (VWR) and finally digested at 37 °C overnight with 2 μg trypsin (Promega). The peptide mixture was acidified, desalted with Sep-Pak C18 1 cc vacuum cartridges (Waters), dried in a vacuum and dissolved in buffer A (0.1% formic acid, at a concentration of 400 ng μl$^{-1}$). The peptides (400 ng) were separated with the Thermo EASY-nLC 1200 System (Thermo Fisher Scientific; flow rate of 250 nl min$^{-1}$), equipped with a 30-cm analytical column (inner diameter, 75 μm; packed in house with ReproSil-Pur C18-AQ 1.9-μm beads, Dr. Maisch) coupled to the benchtop Orbitrap Q Exactive HF (Thermo Fisher Scientific) mass spectrometer, with an increasing gradient of buffer B (80% acetonitrile, 0.1% formic acid).

The raw data were processed with Proteome Discoverer (version 2.5.0.400) with XlinkX/PD nodes integrated[36]. DSS or BS3 was set as a cross-linker, cysteine carbamidomethylation was set as a fixed modification, and methionine oxidation and protein N-terminal acetylation were set as dynamic modifications. 'Trypsin/P' was specified as the protease, and up to two missed cleavages were allowed. Identifications were only accepted with a minimal score of 40 and a minimal delta score of 4. Filtering at a false discovery rate of 1% was calculated with the XlinkX Validator node with the setting 'simple'.

## Generation of the structural model

As structural data are unavailable for critical regions of the proteins used in our study, we generated the structural model manually. To this end, we used the crystal structure of the β1D-CT–TLN2 complex (PDB 3G9W) because the structures of neither β$_{1A}$-CT nor β$_{1A}$-CT in complex with TLN1 (β$_{1A}$-CT–TLN1) have been solved yet. Because the kindlin-binding site of β$_{1D}$-CT differs from that of β$_{1A}$-CT and is not resolved in the structure, we used the β$_{1A}$-CT−ΔKIND2 crystal structure (PDB 5XQ0) and aligned the resolved amino acids of β$_{1A}$-CT−ΔKIND2 and β$_{1D}$-CT–TLN2. Furthermore, structural information of the flexible N terminus of the KIND2-F0 domain is also not resolved in any of the published KIND2 crystal structures. Therefore, we aligned the NMR structures of the individual KIND2-F0 domain (PDB 6U4N) with the model. Subsequently, we mapped the intramolecular and intermolecular cross-links obtained in our cross-linking mass spectrometry experiments onto these assembled published structures (TLN2-F2F3–β$_{1D}$[11], ΔKIND2–β$_{1A}$[4] and KIND2-F0 (ref. 37)) using the XMAS plugin[38] for ChimeraX[39]. We colored cross-links of the relevant distance of 5–30 Å in yellow and all remaining cross-links in red and then iteratively adjusted the orientation of TLN1, KIND2 and β1-CT toward each other until the maximal number of cross-links were of the relevant distance.

## Cell culture

Cells were grown and maintained in DMEM medium supplemented with 10% FBS and 1% penicillin–streptomycin on 10-cm Petri dishes. At about 80% confluency, cells were washed with 5 ml PBS and detached with 1 ml 0.05% trypsin–EDTA in PBS at room temperature before adding 5 ml warm DMEM with 10% FBS. Cells were sedimented by centrifugation (5 min, 350$g$, room temperature), the supernatant was removed, and the cells were resuspended in 5 ml warm DMEM with 10% FBS. Cells were stained with Trypan blue to determine cell count and viability (EVE, NanoEnTek). Next, 200,000–400,000 cells were seeded in 10 ml DMEM with 10% FBS and 1% penicillin–streptomycin in 10-cm Petri dishes. All applied cell lines regularly tested negative for mycoplasma contamination.

The plasmid encoding murine TLN1–YPet in the retroviral vector pLPCXmod has been described previously[40]. The TLN1$^{K402E}$ mutation was introduced by site-specific mutagenesis as described in Cloning. The plasmid encoding mCherry–KIND2 is based on EGFP–KIND2 in the retroviral vector pRetroQ-AcGFP-C1 (Clontech) as published earlier[1]. The plasmid encoding human β$_1$-integrin in the retroviral vector pLZRS was described earlier[41].

Retroviruses for stable transduction were grown in HEK293T cells, and *Tln1*$^{-/-}$;*Tln2*$^{-/-}$-dKO, *Tln1*$^{-/-}$;*Tln2*$^{-/-}$;*Fermt1*$^{-/-}$;*Fermt2*$^{-/-}$-qKO and *Itgb1*$^{-/-}$ murine kidney fibroblasts were generated and transduced as published earlier[1,3,41]. After transduction, dKO cells rescued with wild-type TLN1–YPet and TLN1–YPet$^{K402E}$ were sorted for equal YPet MFI signal, qKO cells rescued with wild-type TLN1–YPet and TLN1–YPet$^{K402E}$ and wild-type mCherry–KIND2 and mCherry–KIND2$^{Y13A}$ were sorted for equal YPet and mCherry MFI signals, and rescued *Itgb1*-knockout cells were sorted for equal PE MFI signal after staining for total β$_1$-integrin surface levels with a PE-labeled anti-β$_1$-integrin antibody using the FACSAria III Cell Sorter (BD Biosciences). Talin and kindlin expression levels of wild-type TLN1–YPet and TLN1–YPet$^{K402E}$ dKO cells and wild-type TLN1–YPet and TLN1–YPet$^{K402E}$ qKO cells were further compared to those of *Tln1*$^{fl/fl}$;*Tln2*$^{-/-}$;*Fermt1*$^{fl/fl}$;*Fermt2*$^{fl/fl}$ fibroblasts by western blot following our previous publication[1].

## Flow cytometry

Around 400,000 cells were seeded into a well of a six-well cell culture plate the day before performing flow cytometry. The cells were detached from the culture plates with 500 μl trypsin and EDTA in PBS, trypsin was neutralized with 500 μl DMEM medium supplemented with 10% FBS, and samples were transferred into 4 ml DMEM supplemented with 10% FBS and split into different FACS tubes or 96-well plates. The medium was removed by centrifugation (5 min, 350$g$, 4 °C), and cells were washed twice with cold PBS and incubated with primary antibodies (9EG7 monoclonal antibody, which binds extended β$_1$-integrin (1:100), or total β$_1$-integrin (1:200), diluted in adhesion buffer (PBS, 3% BSA, 4.5 g l$^{-1}$ glucose, 1 mM calcium chloride, 1 mM magnesium chloride)) for 30 min on ice. After washing with cold PBS, secondary antibodies (anti-rat 647 and streptavidin-720) were diluted 1:500 in adhesion buffer and added to cells for 30 min on ice. The cells were washed again with cold PBS and suspended in PBS supplemented with 3% BSA. For integrin profiling, cells were incubated after washing with PBS at a 1:50 dilution of PE-labeled anti-β$_1$-integrin, anti-β$_3$-integrin, anti-α$_5$-integrin, anti-α$_V$-integrin antibodies or the corresponding isotype controls in PBS supplemented with 3% BSA for 15 min at room temperature. All measurements were performed with an LSRFortessa X-20 flow cytometer (BD Biosciences). Data were analyzed with FlowJo 10 and OriginPro 2019b. Binding of the 9EG7 antibody was normalized to total β$_1$-integrin levels (9EG7 signal divided by total β$_1$ signal). For integrin profiling, the MFI for each integrin staining was first corrected for its corresponding isotype control, and then data from TLN1–YPet$^{K402E}$ dKO and qKO cells were normalized to wild-type TLN1–YPet dKO and qKO cells. Each data point shown originates from an experiment on an individual day. Bar charts show mean ± s.d.

## Focal adhesion analysis

Eight-well glass slides (tissue culture treated, 0030 742.036, Eppendorf) were coated with 5 µg ml$^{-1}$ FN (341635, Merck) in PBS for at least 30 min at 37 °C, followed by blocking with 3% BSA in PBS for at least 30 min at 37 °C. After washing the wells with PBS and DMEM, 5,000–10,000 cells serum-starved for 4 h in DMEM were seeded and incubated for 40 min at 37 °C. The medium was removed, and the cells were fixed with 4% paraformaldehyde (PFA) in PBS for 10 min at room temperature, followed by DAPI staining (1:10,000 in PBS) for 5 min at room temperature and three washing steps with PBS. The cells were imaged on a custom-made TIRF microscope (VisiTIRF, Visitron Systems) based on an Observer Z1 microscopy stand (Carl Zeiss). TIRF illumination was performed with a ×100 TIRF objective (Plan-Apochromat ×100/1.46 Oil, Carl Zeiss). Focal adhesion properties were analyzed based on TIRF images recorded in the 561-nm laser channel for the mCherry–KIND2 signal using the Focal Adhesion Analysis Server, setting the threshold to 2 and the minimal adhesion size to five pixels[42].

## Spreading

Eight-well slides (µ-Slides, 80826, ibiTreat, ibidi) or six-well plates with hydrogels of different stiffnesses (Cell Guidance Systems) were coated with 5 µg ml$^{-1}$ FN (341635, Merck) in PBS, 2.5 µg ml$^{-1}$ VN (07180, Stemcell Technologies) or 10 µg ml$^{-1}$ laminin-111 in PBS (L2020, Sigma-Aldrich) for at least 30 min at 37 °C, followed by blocking with 3% BSA in PBS for at least 30 min at 37 °C. After washing the wells with PBS and DMEM, 5,000–10,000 cells serum-starved for 4 h in DMEM were seeded and incubated for different times. When working with eight-well slides, the medium was afterward removed, and the cells were fixed with 4% PFA in PBS for 10 min at room temperature, followed by DAPI staining (1:10,000 in PBS) for 5 min at room temperature and three washing steps with PBS. When working with six-well plates and hydrogels, cells were incubated for 24 h, washed with PBS and immediately imaged to avoid artifacts from overfixation. Cells were imaged on an EVOS FL Auto 2 microscope (Invitrogen). The area of at least 50 individual cells was determined with Fiji ImageJ using the fluorescent signal of the reconstituted TLN1–YPet constructs.

## Single-cell force spectroscopy

For cantilever functionalization, cantilevers (NP-0, Bruker) were first plasma cleaned (PDC-32G, Harrick Plasma) and then incubated overnight in PBS containing concanavalin A (2 mg ml$^{-1}$, Sigma-Aldrich) at 4 °C[43]. For substrate coating, 200-µm-thick four-segmented polydimethylsilane masks were fused to glass surfaces of Petri dishes (WPI)[44]. Polydimethylsilane-framed glass surfaces were incubated overnight with an FN fragment (FNIII7–10$^{RGD}$, 50 µg ml$^{-1}$ in PBS) at 4 °C. A NanoWizard II AFM equipped with a CellHesion module (both from JPK Instruments) mounted on an inverted fluorescence microscope (Observer Z1, Zeiss) was used for SCFS. The temperature was controlled at 37 °C by a PetriDishHeater (JPK Instruments). Tipless V-shaped silicon nitride cantilevers (200 µm long, NP-0) having nominal spring constants of 0.06 N m$^{-1}$ were used. Each cantilever was calibrated before measurement by determining its sensitivity and spring constant using the thermal noise analysis of the AFM.

Fibroblasts were grown to a confluency of ~80%, washed with PBS, detached with trypsin for 2 min, suspended in SCFS medium (bicarbonate-free DMEM supplemented with 20 mM HEPES) containing 1% (vol/vol) FCS and centrifuged, and the sedimented cells were resuspended in serum-free SCFS medium. Fibroblasts were allowed to recover from trypsin detachment for at least 30 min[45]. Afterward, suspended fibroblasts were pipetted onto substrate-coated Petri dishes, and the functionalized cantilever was lowered onto a single fibroblast with a speed of 10 µm s$^{-1}$ until a force of 5 nN was recorded. After 5 s of contact, the cantilever was retracted at 10 µm s$^{-1}$ until the cell was completely detached from the substrate, and the cantilever-bound fibroblast was incubated for 3–5 min on the cantilever to ensure firm

**Table 1 | Reagents used in the study**

| Reagent | Supplier |
| --- | --- |
| **Antibodies** | |
| Anti-KIND2 (1:1,000) | Merck Millipore, MAB2617 |
| Anti-talin head HRP conjugate (1:100) | Santa Cruz Biotechnology, sc-365875 HRP |
| Anti-talin (1:1,000) | Sigma, T3287 |
| Anti-β$_1$-integrin (total level, biotinylated) (1:200) | eBioscience, 13-0291-80 |
| Anti-β$_1$integrin (9EG7, extended conformation) (1:100) | PharMingen, 550531 |
| Anti-β$_1$-integrin PE (1:200) | BioLegend, 102207 |
| Anti-β$_3$-integrin PE (1:50) | eBioscience, 12-0611 |
| Anti-α$_V$-integrin PE (1:50) | BD, 551187 |
| Anti-α$_5$integrin PE (1:50) | PharMingen, 557447 |
| Anti-GAPDH (1:1,000) | Calbiochem, CB1001 |
| Goat anti-mouse IgG HRP conjugate (1:10,000) | Bio-Rad, 1721011 |
| Anti-rat 647 (1:500) | Invitrogen, A21247 |
| Streptavidin eFluor 780 (1:500) | eBioscience, 47-4317-82 |
| Rat IgG1 PE isotype control (1:50) | PharMingen, 554685 |
| Rat IgG2 PE isotype control (1:50) | PharMingen, 555844 |
| Hamster IgG PE isotype control (1:50) | eBioscience, 1091682 |
| **Enzymes and proteins** | |
| Concanavalin A | Sigma-Aldrich |
| DNase I/benzonase | MPIB Biochemistry Core Facility |
| SenP2 | MPIB Biochemistry Core Facility |
| MSP2N2 | Cube Biotech, 26176 and MPIB Biochemistry Core Facility |
| Restriction enzymes | NEB |
| PfuUltra II Fusion HS DNA Polymerase | Agilent, 600670 |
| Q5 DNA Polymerase | NEB, M0493 |
| PNK | NEB, M0201 |
| Fast-Link DNA Ligation Kit | Lucigen, LK6201H |
| Lysozyme | Sigma-Aldrich, L6876 |
| FN | Merck, 341635 |
| Vitronectin | Stemcell Technologies, 07180 |
| Laminin | Sigma-Aldrich, L2020 |
| Trypsin+EDTA | Gibco, 15400-054 |
| Integrin CT, ATTO 488 labeled | MPIB Biochemistry Core Facility, peptide synthesis |
| **Fluorescent dyes** | |
| Alexa 488 NHS ester | Thermo Fisher, A20000 |
| ATTO 488 NHS ester | ATTO-TEC, AD 488-31 |
| ATTO 488 maleimide | ATTO-TEC, AD 488-41 |
| ATTO 565 NHS ester | ATTO-TEC, AD 565-31 |
| Alexa 647 maleimide | Thermo Fisher, A20347 |
| Alexa 647 NHS ester | Thermo Fisher, A20006 |
| **Bacterial strains** | |
| *E. coli* Rosetta Bl21 (DE3) (F$^-$ *ompT hsd*SB(rB$^-$ mB$^-$) *gal dcm* (DE3) pRARE (Cam$^R$)) | Merck Millipore, 70954-3 |
| *E. coli* One Shot OmniMAX 2 T1$^R$ | Thermo Fisher, C854003 |

**Table 1 (continued) | Reagents used in the study**

| Reagent | Supplier |
| --- | --- |
| NEB 5-alpha Competent *E. coli* | New England Biolabs, C2987 |
| **Chromatography** | |
| DextraSEC PRO2 desalting columns | AppliChem, A8710,0050 |
| Zeba Spin desalting columns | Thermo Fisher, 89882 and 89891 |
| ENrich SEC 650 | Bio-Rad, 780-1650 |
| HiLoad 16/600 Superdex 200 PG | GE Healthcare, 28-9893-35 |
| Superdex 200 Increase 10/300 GL | GE Healthcare, 28-9909-44 |
| Superdex 75 Increase 10/300 GL | GE Healthcare, 29-1487-21 |
| HisTrap HP, 5 ml | GE Healthcare, 17-5248-02 |
| HiTrap SP HP, 1 ml | GE Healthcare, 17115101 |
| Ni-NTA agarose beads | Qiagen, 30210 |
| **Miscellaneous** | |
| MST capillaries, premium coated (hydrophilic) | NanoTemper, MO-K005 |
| Amicon Ultra-4, 10 kDa | Merck Millipore, UFC801024 |
| Amicon Ultra-15, 10 kDa | Merck Millipore, UFC901024 |
| Amicon Ultra-15, 30 kDa | Merck Millipore, UFC903024 |
| Amicon Ultra, 0.5 ml, 10 kDa | Merck Millipore, UFC501096 |
| Ultrafree-MC GV, PVDF, 0.22 μm | Merck Millipore, UFC30GV00 |
| Millex-GV, 0.22 μm, PVDF | Merck Millipore, SLGV033RS |
| Millex-HA, 0.45 μm, MCE | Merck Millipore, SLHA033SS |
| Stericup and Steritop, 0.22 μm, GP | Merck Millipore, SCGPT05RE |
| Slide-A-Lyzer MINI Dialysis Devices, 10-kDa MWCO, 0.5 ml | Thermo Fisher, 88401 |
| Dynabeads M-280 Streptavidin | Invitrogen, 11205D |
| **Kits** | |
| QIAprep Spin Miniprep Kit | Qiagen, 27106 |
| QIAquick PCR Purification Kit | Qiagen, 28106 |
| NEBuilder HiFi DNA Assembly Cloning Kit (Gibson assembly) | NEB, E5520S |
| **Lipids** | |
| *n*-Dodecyl-β-D-maltopyranoside | Anatrace, D310-25GM |
| DMPC lipid | Corden Pharma, LP-R4-B58 |
| EMPIGEN BB, 30% solution | Merck, US1324690-100ML |
| PI(4,5)P2 (1,2-dioctanoyl-*sn*-glycero-3-phospho-(1'-myo-inositol-4',5'-bisphosphate)) | Avanti Polar Lipids, 850185P |
| PI(3,4,5)P3 (1,2-dioleoyl-*sn*-glycero-3-phospho-(1'-myo-inositol-3',4',5'-trisphosphate) (ammonium salt)) | Avanti Polar Lipids, 850156P |
| **Chemicals** | |
| Acetic acid | Sigma-Aldrich, 33209 |
| Acetone | Sigma-Aldrich, 32201 |
| AgNO$_3$ | Riedel-de Haën, 31630 |
| Ampicillin | Sigma-Aldrich, A9518 |
| Biotin-maleimide | Sigma-Aldrich, B1267 |
| Brilliant Blue R | Sigma-Aldrich, B0149 |
| Chloramphenicol | Sigma-Aldrich, C0378 |
| Chloroform | Merck, 1.02445 |
| DMEM+GlutaMAX | Gibco, 31966-021 |
| EDTA | Merck, 1.08418 |

**Table 1 (continued) | Reagents used in the study**

| Reagent | Supplier |
| --- | --- |
| Ethanol | Sigma-Aldrich, 32221 |
| FBS | Gibco, 10270-106 |
| Formaldehyde, 37% | Merck Millipore, 1.04003.1000 |
| Glycerol, 86% | Roth, 4043.1 |
| Glycine | Sigma-Aldrich, 33226 |
| HCl, 37% | VWR Chemicals, 20252.335 |
| HEPES | Biomol, 05288.100 |
| Imidazole | Merck Millipore, 1.04716.0250 |
| IPTG | Roth, 2316.5 |
| Isopropanol | Sigma-Aldrich, 33539 |
| Kanamycin | Sigma-Aldrich, K1876 |
| K$_2$HPO$_4$·3H$_2$O | Roth, 6878.1 |
| MES | Merck Millipore, 1.06126.0025 |
| Methanol | Sigma-Aldrich, 32213 |
| MgCl$_2$·6H$_2$O | Merck Millipore, 1.05833.1000 |
| MnCl$_2$ | Merck, 1.05934.0100 |
| NaCl | Roth, 3957.2 |
| Na$_2$CO$_3$ | Merck Millipore, 1.06392.1000 |
| NaHCO$_3$ | Merck Millipore, 6329.1000 |
| NaH$_2$PO$_4$·H$_2$O | Merck Millipore, 1.06346.1000 |
| NaOH | VWR Chemicals, 28245.298 |
| Na$_2$S$_2$O$_3$·5H$_2$O | Merck Millipore, 0077695 |
| NiSO$_4$·6H$_2$O | Sigma-Aldrich, 31483 |
| PBS | Sigma-Aldrich, P4417 |
| Penicillin+streptomycin | Gibco, 15140122 |
| PMSF | Sigma-Aldrich, P7626 |
| ROTIPHORESE Gel 30 | Roth, 3029.1 |
| SDS pellets | Roth, CN30.3 |
| Sodium azide | Merck Millipore, 1.06688.0100 |
| Sodium cholate hydrate | Sigma-Aldrich, C6445 |
| Sulfuric acid | Sigma-Aldrich, 258105 |
| TCEP | Roth, HN95.2 |
| Trichloroacetic acid | Merck Millipore, 1.00807.1000 |
| Tricine | Sigma-Aldrich, T0377 |
| Triton X-100 | Roth, 3051 |
| Trizma base (Tris) | Sigma-Aldrich, T1503 |
| Tryptone/peptone | Roth, 8952.2 |
| Tween-20, 10% solution | Pierce, 28320 |
| Yeast extract | Roth, 2363.3 |
| ZnCl$_2$ | Merck Millipore, 8816.0250 |

binding to the cantilever. The morphological state of the fibroblast was monitored by optical microscopy throughout adhesion experiments. Round, cantilever-bound fibroblasts approached the substrate at 5 μm s$^{-1}$ until a contact force of 1 nN was recorded. Throughout the contact times of 5, 20, 60, 120 or 240 s, the height of the cantilever was maintained constant and subsequently retracted at 5 μm s$^{-1}$ for 100 μm until the fibroblast was fully separated from the substrate. Before another experimental cycle was initiated, the cantilever-bound fibroblast was allowed to recover for a time period equal to the contact

time. A single fibroblast was used to probe the adhesion force for all contact times once or until morphological changes (that is, spreading) were observed. The sequence of contact times and area on the substrate were varied. Adhesion forces were determined from force–distance curves as the maximum downward deflection of the cantilever after baseline correction using JPK software (JPK Instruments).

## Plate-and-wash assay

Untreated 96-well plates were coated with 100 μl of 10 μg ml$^{-1}$ FN or a 1:10 dilution of poly-L-lysine (Sigma) in 20 mM Tris, pH 8.8, overnight at 4 °C. Next, 200 μl of 3% BSA in PBS was added to the wells for at least 30 min at 37 °C. The plates were washed with 200 μl PBS and 200 μl DMEM medium before seeding 40,000 serum-starved cells in 100 μl DMEM. After incubating for 30 min at 37 °C, the whole 96-well plate was washed by immersing in PBS three times, followed by fixation in 4% PFA for 10 min at room temperature. The cells were stained with 75 μl crystal violet solution for 30 min at room temperature and washed three times by immersing with dH$_2$O. The wells were incubated with 100 μl 2% SDS while shaking at room temperature, until all crystal violet was dissolved. Absorption was measured at λ = 595 nm in a SpectraMax ABS plate reader (Molecular Devices). The data for every cell line were first blanked to those of control wells coated with only BSA and then corrected for the plated cell count obtained from poly-L-lysine-coated wells. Finally, data were normalized to those of the wild-type β$_1$-expressing cell line.

## Statistics and data representation

Two samples were compared to each other using a two-sample two-tailed Student's $t$-test (Figs. 1e,f, 3l,j and 5a and Extended Data Figs. 1g,h,l,j,m, 3k–n and 4a,b) or a two-tailed Mann–Whitney test (Fig. 5d,e). For comparisons with more than two samples, one-way repeated-measures ANOVA with Tukey's post hoc test was used (Fig. 5c,f,g and Extended Data Fig. 5e,f,I,j). Normalized data were tested with a one-sample two-tailed Student's $t$-test for a difference from 1 for comparison with the reference sample (Fig. 5a and Extended Data Fig. 5d,h,i). $P$ values from statistical tests are written in the figures over the corresponding bar chart or box plot in black, whereas $n$ is written in the same color as the chart or plot. Unless stated otherwise, bar charts represent average values and error bars represent standard deviations of individual experiments or analyzed single cells. In box plots, the boxes show the upper and lower quartiles and the median and the error bars show standard deviations of individual experiments.

## Reagents

Reagents used in the study are detailed in Table 1.

## Reporting summary

Further information on research design is available in the Nature Portfolio Reporting Summary linked to this article.

## Data availability

Additional data and materials can be obtained from the corresponding author upon request. Source data are provided with this paper.

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

## Acknowledgements

We thank S. Bach for technical assistance, S. Uebel for help with MST measurements, β-CT synthesis and discussion of data, S. Suppmann, L. Urich and C. Strasser for protein expression, F. Brod and B. Steigenberger for help with cross-linking experiments, S. Asami and G. Gemmecker for NMR support and S. Grün for gel electrophoresis. This work was funded by the European Research Council (grant agreement no. 810104, point to R.F.), the DFG (SFB-863 to R.F. and SFB1035 and GRK1721 to M.S.), the Swiss National Science Foundation (grant no. 31003A_182587/1 to N.S.) and the Max Planck Society.

## Author contributions

J.A. and R.F., conceptualization and writing original draft; J.A., M.A. and N.S., formal analysis, investigation, methodology and visualization; J.A., M.A., N.S and M.S., writing (review and editing); M.S. and R.F., funding acquisition, resources and supervision.

## Funding

## Competing interests

The authors declare no competing interests.

## Additional information

**Extended data** is available for this paper at https://doi.org/10.1038/s41594-023-01139-9.

**Correspondence and requests for materials** should be addressed to Reinhard Fässler.

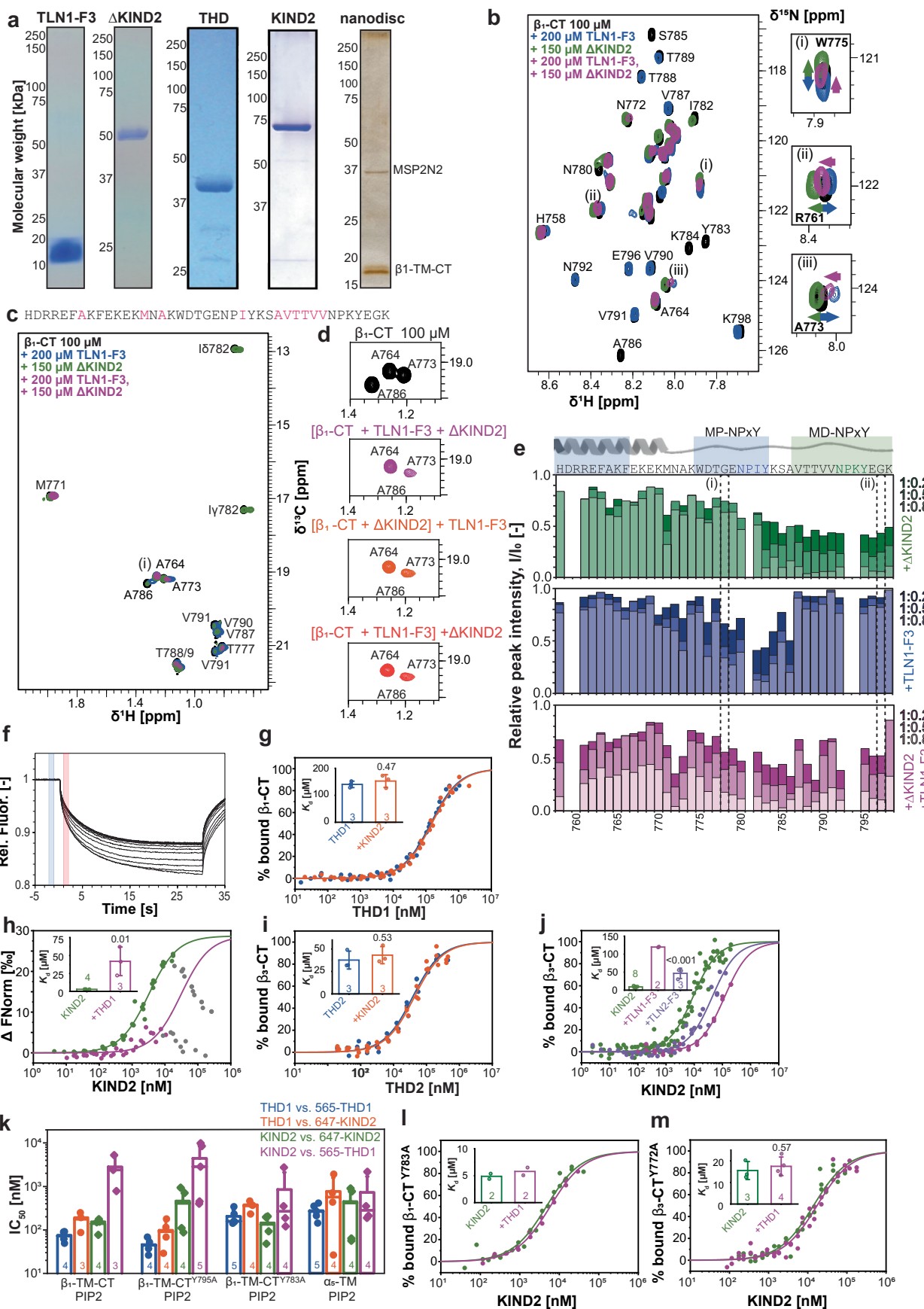

**Extended Data Fig. 1 | See next page for caption.**

**Extended Data Fig. 1 | Analysis of ternary interactions of TLN1-F3, ΔKIND2 and β-CT. a**) Exemplary SDS-PAGEs of TLN1-F3, ΔKIND2, THD1, KIND2 and β₁-TM-CT containing nanodisc preparations. **b**) Overlay of $^1$H-$^{15}$N HSQC NMR spectra of $^{15}$N-labeled β₁-CT before (black) and after addition of TLN1-F3 (green), ΔKIND2 (blue), or TLN1-F3 and ΔKIND2 (purple). Expansions of the overlaid spectra (i-iii) highlight the differences recorded in presence of TLN1-F3 or ΔKIND2 only, or TLN1-F3 and ΔKIND2 together. Arrows indicate changes of chemical shifts induced by addition of TLN1-F3 (green), ΔKIND2 (blue), or both proteins (purple). **c**) Overlay of methyl region of $^1$H-$^{13}$C correlation NMR spectra of uniformly $^{13}$C,$^{15}$N-labeled β₁-CT before (black) and after addition of TLN1-F3 (blue), ΔKIND2 (green), or TLN1-F3 and ΔKIND2 together (purple). Methyl-containing amino acids observed in this experiment are highlighted in red in the β₁-CT sequence (top). (**i**) highlights spectra shown in Fig. 1c and panel **e. d**) TLN1-F3 (200 μM; red) or ΔKIND2 (150 μM; orange) were preincubated with $^1$H-$^{13}$C-labeled integrin β₁-CT for 30 min at room temperature, then 150 μM ΔKIND2 was added to β₁-CT/TLN1-F3 or 200 μM TLN1-F3 to β₁-CT/ΔKIND2 samples and incubated for 10 min before NMR spectra were recorded. The methyl regions of the spectra are compared to those of β₁-CT alone (black) and β₁-CT incubated with 200 μM TLN1-F3 and 150 μM ΔKIND2 together (purple). Stepwise and simultaneous incubation of TLN1-F3, ΔKIND2 and β₁-CT produced identical spectra indicating that ternary complexes form at equilibrium. **e**) Intensity ratio of peaks of the β₁-CT in presence and absence of increasing concentrations of ΔKIND2 (top, green), TLN1-F3 (middle, blue), or both (bottom, magenta). Reported talin (blue) and kindlin (green) binding sites are indicated above the plots (MP: membrane proximal, MD: membrane-distal) and are reflected by line-broadening upon protein addition. The isolated peaks shown

in Fig. 1b are indicated by (i) and (ii), respectively. Amino acid numbering is shown below plots. **f**) Exemplary MST traces of THD1 titrated to ATTO488-labeled β₃-CT. To minimize heat effects, fluorescence changes were analyzed 1.5 s before (blue line) and after (red line) turning on the infrared laser. **g**) Titration of THD1 to ATTO488-labeled β₁-CT in the presence (orange) or absence (blue) of 30 μM KIND2 (corresponding to 6-fold excess of $K_d$). **h**) Titration of KIND2 to ATTO488-labeled β₁-CT in the presence (purple) or absence (green) of 300-400 μM THD1 (corresponding to 2-3-fold excess of $K_d$). Only concentrations up to about 10 μM KIND2 were considered and hence, non-linear regressions were extrapolated. Non-considered data points are shown in grey. **i**) Titration of THD2 to ATTO488-labeled β₃-CT in the presence (orange) or absence (blue) of 30 μM KIND2 (corresponding to 3-fold excess of $K_d$). **j**) MST measurements of KIND2 affinity for ATTO488-labeled β₃-CT in the absence (green) and presence of 250-450 μM TLN1-F3 (purple) or TLN2-F3 domain (dark purple; both corresponding to 3-5-fold excess of $K_d$). **k**) FC-RDA generated IC₅₀ values of unlabeled THD1 competing binding of 565-THD1 (blue) and 647-KIND2 (orange), and unlabeled KIND2 competing binding of 647-KIND2 (green) and 565- THD1 (purple) to β₁-TM-CT, β₁-TM-CT$^{Y795A}$ (KIND2-binding impaired), β₁-TM-CT$^{Y783A}$ (THD1-binding impaired), and α₅-TM lacking α₅-CT (α₅-TM) embedded in 10% PIP2/90% PC-containing nanodiscs. Bars display mean ± SD (standard deviation) with a line for the median. **l**) MST affinity measurements of KIND2 for ATTO488-labeled talin-binding-impaired β₁-CT$^{Y783A}$ in absence (green) or presence (purple) of 400-600 μM THD1. **m**) MST affinity measurements of KIND2 for ATTO488-labeled talin-binding-impaired β₃-CT$^{Y772A}$ in absence (green) or presence (purple) of 500 μM THD1.

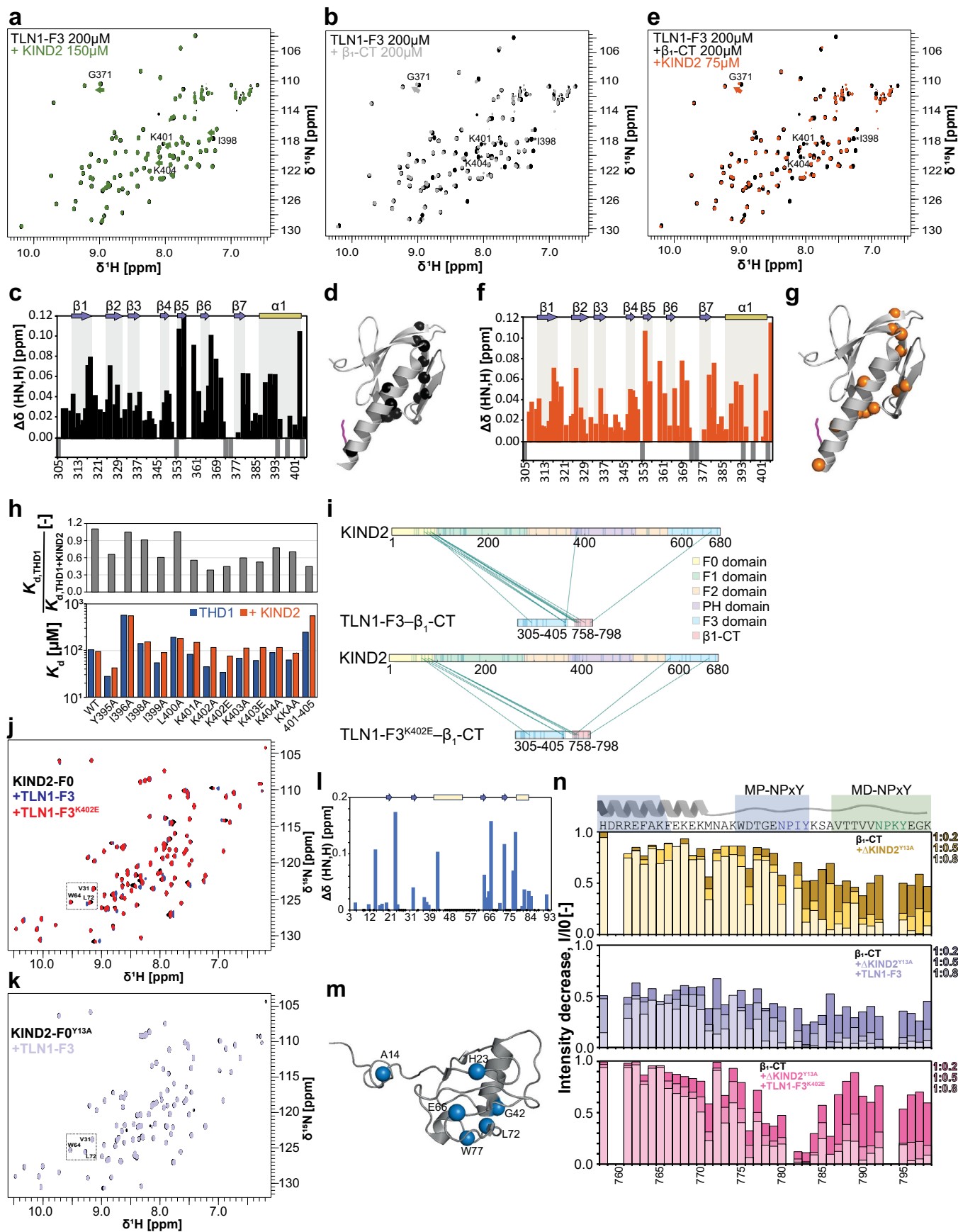

Extended Data Fig. 2 | See next page for caption.

**Extended Data Fig. 2 | The α1-helix in TLN1-F3 binds to KIND2-F0. a,b**) $^1$H-$^{15}$N HSQC NMR spectra of $^{15}$N-labeled TLN1-F3 (black) in the presence of KIND2 (**a**, green) or integrin $\beta_1$-CT peptide (**b**, grey). **c**) CSP plot of $^{15}$N-labeled TLN1-F3 upon addition of 200 µM integrin $\beta_1$-CT. Amino acid numbering is shown below the plot. **d**) CSPs recorded in $^{15}$N-labeled TLN1-F3 were mapped on crystal structure of TLN2-F3 (dark grey). **e**) $^1$H-$^{15}$N HSQC NMR spectra of $^{15}$N-labeled TLN1-F3 in the presence of integrin $\beta_1$-CT (black) and KIND2 (orange). **f**) CSP plot of $^{15}$N-labeled TLN1-F3 upon addition of 200 µM $\beta_1$-CT together with 75 µM KIND2. Amino acid numbering is shown below the plot. **g**) CSPs recorded in uniformly $^{15}$N-labeled TLN1-F3 upon addition of $\beta_1$-CT as well as KIND2 (indicated by orange spheres) mapped on the crystal structure of TLN2-F3. **h**) Top: Affinity reduction of THD1 mutants for $\beta_3$-CT calculated by dividing the affinity of THD1 in absence ($K_{d,THD1}$) and presence of KIND2 ($K_{d,THD1+KIND2}$) measured by MST. Bottom: Bar chart of MST-derived affinities of indicated THD1 mutants for ATTO488-labeled $\beta_3$-CT (n = 2) in absence (blue) and presence (orange) of 30-40 µM KIND2 (corresponding to 3-4-fold excess of $K_d$). **i**) Interprotein cross-links identified by mass spectrometry following disuccinimidyl suberate cross-linking of 60 µM TLN1-F3–$\beta_1$-CT (top) and TLN1-F3$^{K402E}$–$\beta_1$-CT (bottom) fusion protein with 450 µM KIND2 using GraFix. **j, k**) $^1$H-$^{15}$N HSQC NMR spectra of j) 100 µM $^{15}$N-labeled KIND2-F0 (black) in the presence of 100 µM TLN1-F3 (blue) or 100 µM TLN1-F3$^{K402E}$ (red) and of **k**) 100 µM $^{15}$N-labeled KIND2-F0$^{Y13A}$ (black) in the presence of 100 µM TLN1-F3 (light blue). Boxes indicate the region shown in Fig. 2h,i. **l**) CSP plot of $^{15}$N-labeled KIND2-F0 upon addition of 100 µM TLN1-F3 titration. Amino-acid numbering is shown below the plot. **m**) CSPs recorded in $^{15}$N-labeled KIND2-F0 were mapped on NMR structure of KIND2-F0 (blue). **n**) Peak intensity ratios in HSQC spectra of $^{15}$N-$\beta_1$-CT in presence of increasing concentrations of ΔKIND2$^{Y13A}$ (yellow), TLN1-F3 and ΔKIND2$^{Y13A}$ (purple), and TLN1-F3$^{K402E}$ and ΔKIND2$^{Y13A}$ (pink). Talin (blue) and kindlin (green) binding sites in $\beta_1$-CT are indicated above the plots (MP: membrane proximal, MD: membrane-distal). Stoichiometries of titrations are shown at the right side of plots and amino acid numbers below plots.

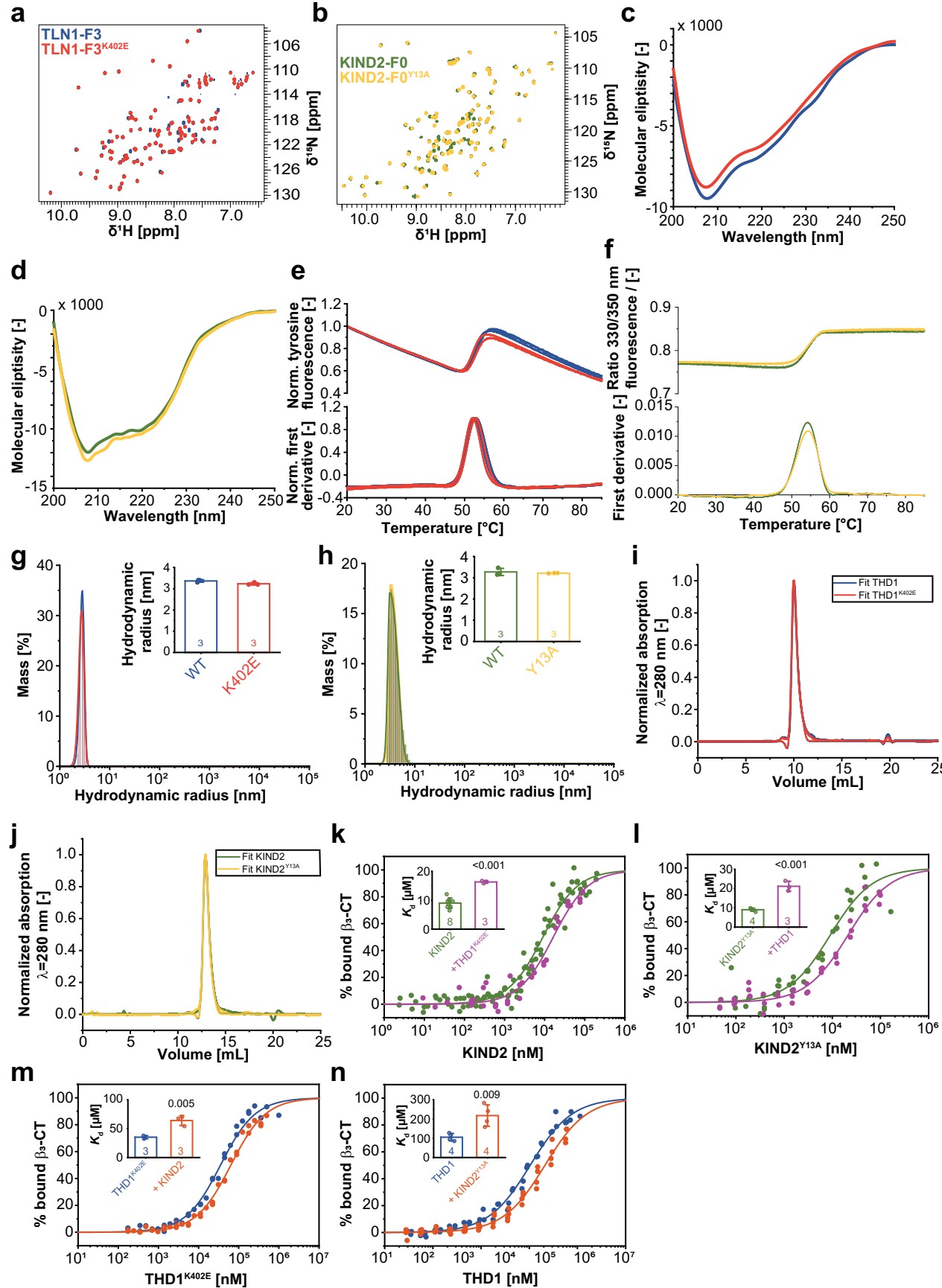

**Extended Data Fig. 3 | See next page for caption.**

**Extended Data Fig. 3 | Characterization of talin-1$^{K402E}$ and kindlin-2$^{Y13A}$ mutants. a,b)** Similar $^1$H-$^{15}$N HSQC NMR spectra of (**a**) $^{15}$N-labeled TLN1-F3 (blue) and TLN1-F3$^{K402E}$ (red) and (**b**) $^{15}$N-labeled KIND2-F0 (green) and KIND2-F0$^{Y13A}$ (yellow) confirm that the mutations do not perturb the overall fold of the proteins. **c,d)** CD spectra of (**c**) THD1 (blue) and THD1$^{K402E}$ (red) and (**d**) KIND2 (green) and KIND2$^{Y13A}$ (yellow) revealed identical secondary structures. **e,f)** Similar thermal stability of (**e**) THD1 (blue) and THD1$^{K402E}$ (red) and (**f**) KIND2 (green) and KIND2$^{Y13A}$ (yellow) determined by differential scanning fluorimetry. **g,h)** Representative dynamic light scattering measurement of (**g**) THD1 (blue) and THD1$^{K402E}$ (red) and (**h**) KIND2 (green) and KIND2$^{Y13A}$ (yellow), which excludes aggregate formation and hydrodynamic radius changes due to altered tertiary structure. Hydrodynamic radius obtained for THD1 was $3.4 \pm 0.1$ nm, for THD1$^{K402E}$ $3.2 \pm 0.1$ nm, and for KIND2 and KIND2$^{Y13A}$ $3.2 \pm 0.1$ nm (bar chart). **i,j)** SEC of (**i**) THD1 (blue) and THD1$^{K402E}$ (red) on Superdex75 and (**j**) KIND2 (green)

and KIND2$^{Y13A}$ (yellow) on a Superdex200 excludes changes in hydrodynamic radius. Data are fitted to Gram-Charlier peak function resulting in identical maximum at 10.2 mL and a half-peak width of 0.34 mL for THD1 and THD1$^{K402E}$ and identical maximum at 13.1 mL and a half-peak width of 0.35 mL for KIND2 and KIND2$^{Y13A}$. **k)** MST affinity measurements of KIND2 for ATTO488-labeled $\beta_3$-CT in the absence (green) and presence (purple) of 120 µM THD1$^{K402E}$ (corresponding to 3-4-fold excess of $K_d$), and **l)** of KIND2$^{Y13A}$ for ATTO488-labeled $\beta_3$-CT in absence (green) and presence (purple) of 300-400 µM THD1 (corresponding to 3-4-fold excess of $K_d$; n = 3). **m)** MST affinity measurements of THD1$^{K402E}$ for ATTO488-labeled $\beta_3$-CT in the absence (blue) and presence (orange) of 30-40 µM KIND2 (corresponding to 3-4-fold excess of $K_d$), and n) of THD1 for ATTO488-labeled $\beta_3$-CT in absence (blue) and presence (orange) of 30-40 µM KIND2$^{Y13A}$ (corresponding to 3-4- fold excess of $K_d$).

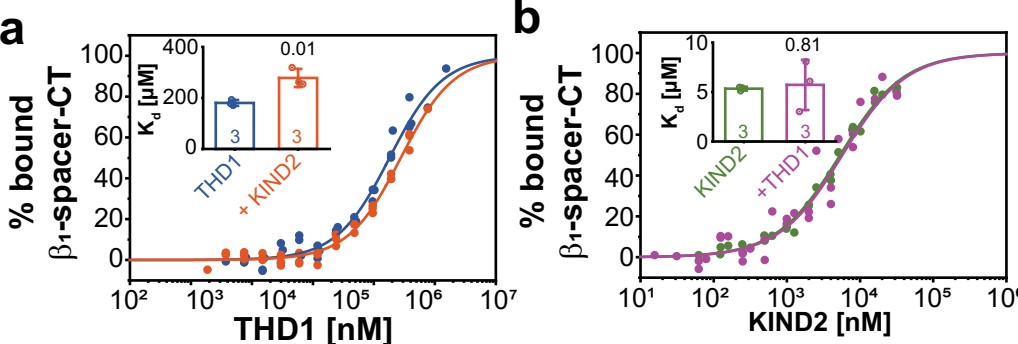

**Extended Data Fig. 4 | The talin and kindlin binding sites on β₁-CT are allosterically coupled. a)** MST affinity measurements of THD1 for ATTO488-labeled β₁-spacer-CT in absence (blue) and presence of 30-35 μM KIND2 (orange, corresponding to 6-7-fold excess of $K_d$), and **b)** of KIND2 for ATTO488-labeled β₁-spacer-CT in absence (green) and presence of 300-400 μM THD1 (purple, corresponding to 2-3-fold excess of $K_d$).

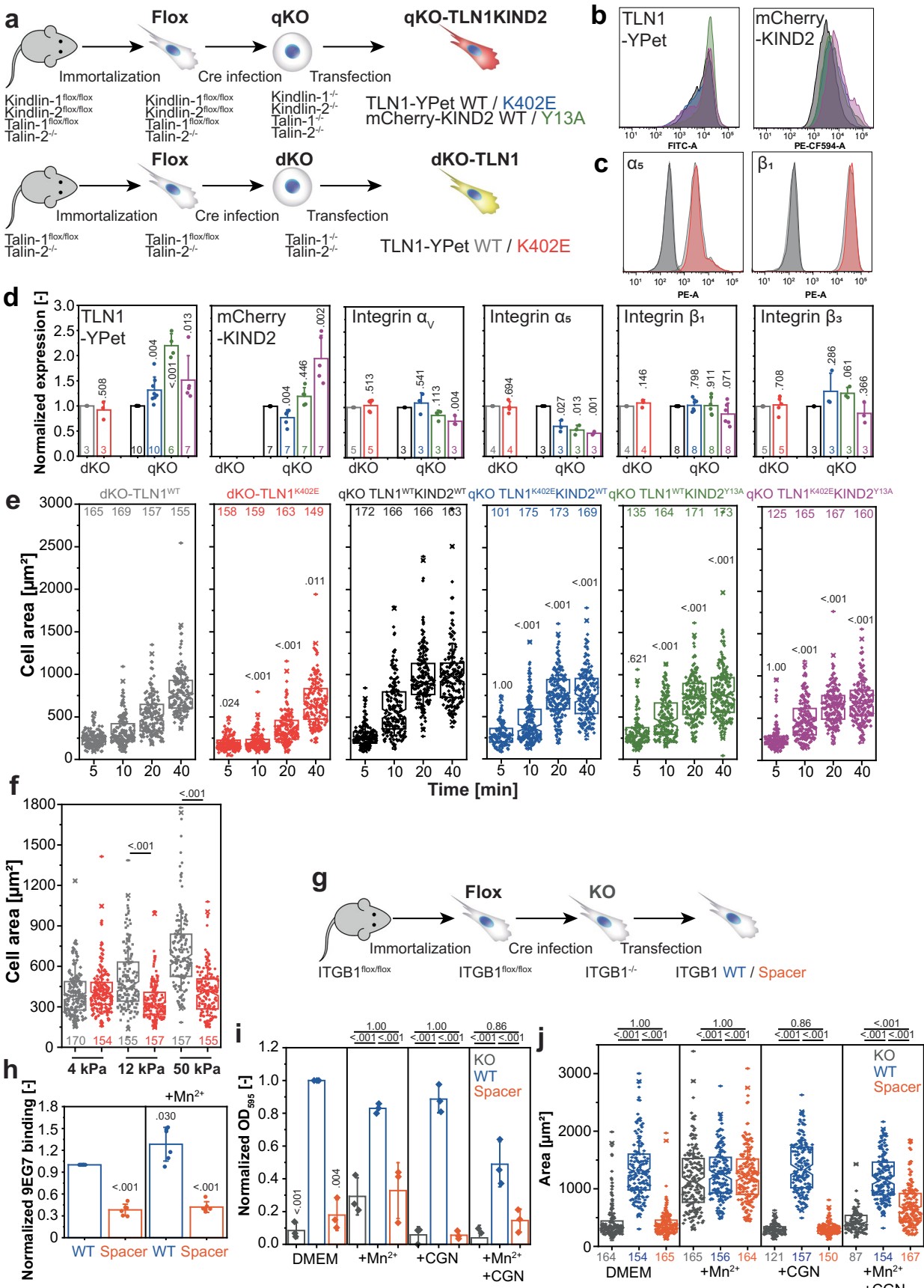

**Extended Data Fig. 5 | See next page for caption.**

**Extended Data Fig. 5 | Generation and characterization of TLN1$^{K402E}$-YPet, mCherry-KIND2$^{Y13A}$ and β$_1$-spacer expressing cells. a)** Cartoon depicting cell line generation. Upper image: mouse fibroblasts deficient of *Talin-1*, *Talin-2*, *Fermt1* and *Fermt2* (quadruple knockout; qKO) were reconstituted with mCherry-KIND2 or mCherry-KIND2$^{Y13A}$ together with either YPet-TLN1 (qKO-TLN1$^{WT}$KIND2$^{WT}$, black; qKO-TLN1$^{WT}$KIND2$^{Y13A}$, green) or YPet-TLN1$^{K402E}$ (qKO-TLN1$^{K402E}$KIND2$^{WT}$, blue; qKO-TLN1$^{K402E}$KIND2$^{Y13A}$, purple). Lower image: mouse fibroblasts deficient of *Talin-1* and *Talin-2* (double knockout, dKO) and expressing endogenous KIND2 were reconstituted with either YPet-TLN1 (dKO-TLN1$^{WT}$; grey) or YPet-TLN1$^{K402E}$ (dKO-TLN1$^{K402E}$; red). **b,c)** TLN1-YPet and mCherry-KIND2 levels in qKO cells (**b**) and surface integrin α$_5$ and β$_1$ in dKO cells (**c**) measured by flow cytometry. **d)** Quantification of TLN1-YPet, mCherry-KIND2, and integrin surface expression of indicated cells measured by flow cytometry. Data of qKO-TLN1$^{K402E}$KIND2$^{WT}$ (blue), qKO-TLN1$^{WT}$KIND2$^{Y13A}$ (green), qKO-TLN1$^{K402E}$KIND2$^{Y13A}$ (purple), and dKO-TLN1$^{K402E}$ cells (red) was normalized to WT (black/grey) and analyzed for differences using a one sample t-test. **e)** Spreading area of indicated cells seeded on FN for indicated times. ≥50 cells (numbers indicated on top of each graph) were analyzed on three independent days. Statistics by one-way ANOVA for repeated measurements with Tukey post hoc test compared to dKO-TLN1$^{WT}$ or qKO-TLN1$^{WT}$KIND2$^{WT}$ cells at identical timepoints). **f)** Spreading area of dKO-TLN1 cells seeded on FN-coated hydrogels with indicated Young's modulus for 24 h. At least 50 cells were analyzed on three independent days. **g)** Scheme outlining the generation of mouse fibroblast cells deficient for β$_1$ integrin (β$_1$-KO, grey) reconstituted with human β$_1$$^{WT}$ (blue) or β$_1$-spacer (orange, Fig. 4c). **h)** Flow cytometry of β$_1$$^{WT}$ and β$_1$-spacer cells for 9EG7 levels normalized to total β$_1$ integrin surface levels (n = 6). **i)** Plate-and-wash adhesion assays of serum-starved β$_1$ knockout (KO), β$_1$$^{WT}$ and β$_1$-spacer cells 30 min after seeding on FN in absence and presence of MnCl$_2$ and/or cilengitide (CGN). Data normalized to values from β$_1$$^{WT}$ cells cultured in absence of MnCl$_2$ and CGN (n = 3). **j)** Spreading area of serum-starved β$_1$ knockout (KO), β$_1$$^{WT}$ and β$_1$-spacer cells 1 h after seeding on FN in absence and presence of MnCl$_2$ and/or CGN. Cells were stained with DAPI and CellMask orange and areas of ≥50 cells were analyzed on three different days.

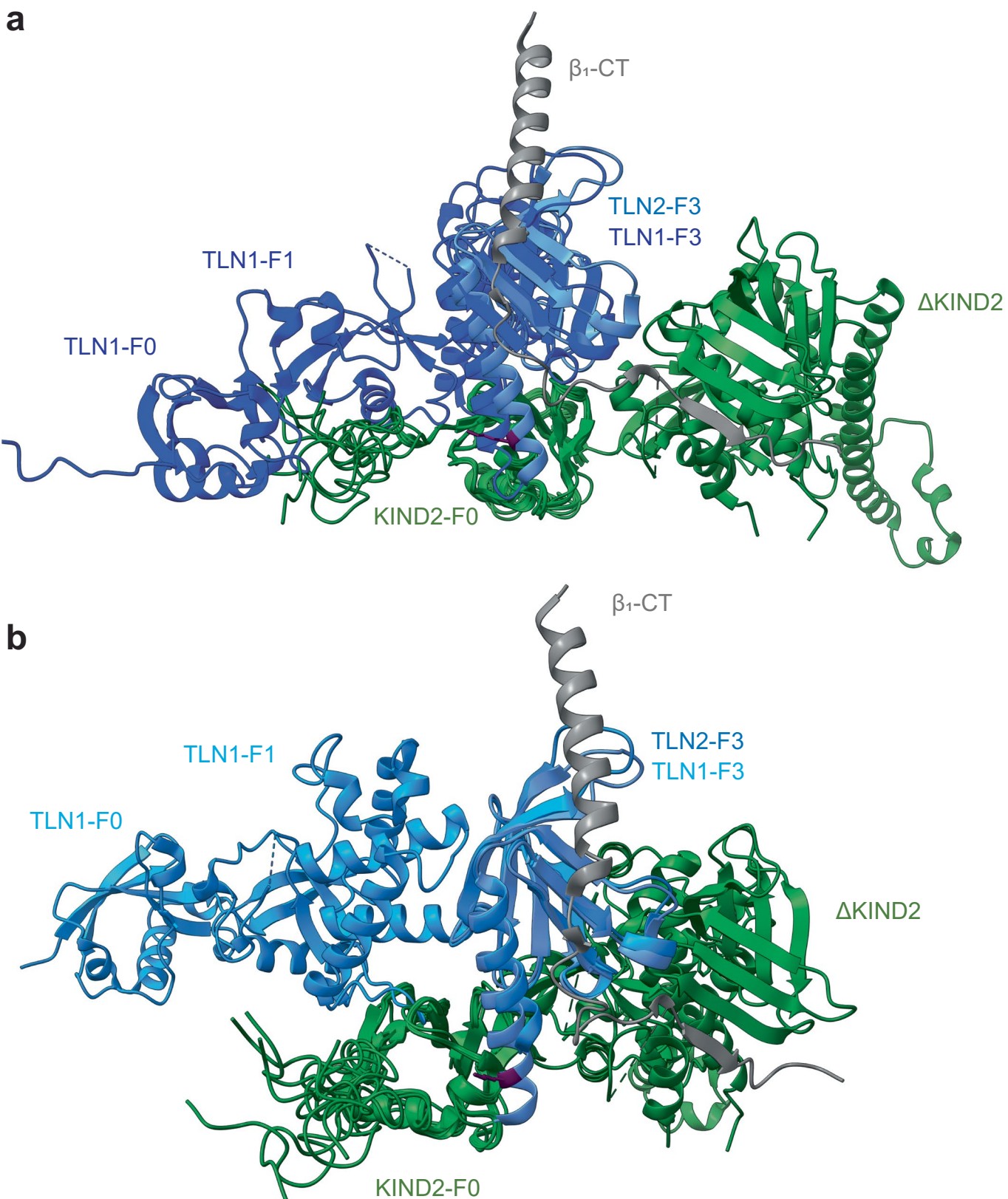

**Extended Data Fig. 6 | Overlay of THD1 structure corresponding to.**
**a**) the canonical cloverleaf (dark blue; PDB: 6VGU) and **b**) an atypical linear conformation (light blue; 3IVF) with the model of TLN-F3-$\beta_1$-CT-ΔKIND2 complex based on chemical crosslinks identified by mass spectrometry (Fig. 2e) using the reported structures of TLN2-F2F3 (blue)-$\beta_{1D}$-CT (grey; PDB: 39 GW), ΔKIND2

(green)-$\beta_{1A}$-CT (grey; PDB: 5XQ0) and KIND2-F0 (green; PDB: 2LGX). Note that the flexible N-terminus of KIND2-F0 occupies the space between the TLN1-F1 and TLN1-F3 domain in the cloverleaf fold (**a**) whereas the distance between the TLN1-F1 and TLN1-F3 domain suffice to accommodate the KIND2-F0 domain in the TLN1 linear fold (**b**).

**Extended Data Table 1 | Mean affinities with standard deviation measured in this study**

| Ligand | Receptor | Affinity (nM) | Reported value (nM) |
|---|---|---|---|
| THD1 | $\beta_1$-CT | 140,000 ± 12,000 | |
| TLN1-F3 | $\beta_1$-CT | | 491,000 ± 10,000[a] |
| THD2 | $\beta_1$-CT | 25,000 | |
| TLN2-F3 | $\beta_1$-CT | | 652,000 ± 20,000[a] |
| THD1 | $\beta_1$-CT$^{Y783A}$ | n.d. | |
| TLN1-F3 | $\beta_1$-CT$^{Y783A}$ | | ~5,000,000[a] |
| THD1 | $\beta_3$-CT | 108,000 ± 12,000 | |
| TLN1-F3 | $\beta_3$-CT | 106,000 | 273,000 ± 6,400[a] |
| THD2 | $\beta_3$-CT | 37,600 ± 10,000 | |
| TLN2-F3 | $\beta_3$-CT | 98,000 ± 7,000 | 438,000 ± 15,000[a] |
| THD1 | $\beta_3$-CT$^{Y772A}$ | 511,000 | |
| TLN1-F3 | $\beta_3$-CT$^{Y772A}$ | | 1,317,000 ± 36,000[a] |
| THD1$^{Y395A}$ | $\beta_3$-CT | 27,700 ± 4,500 | |
| THD1$^{I396A}$ | $\beta_3$-CT | 578,700 ± 55,400 | |
| THD1$^{I398A}$ | $\beta_3$-CT | 141,400 ± 18,500 | |
| THD1$^{I399A}$ | $\beta_3$-CT | 55,200 ± 5,300 | |
| THD1$^{L400A}$ | $\beta_3$-CT | 190,200 ± 14,700 | |
| THD1$^{K401A}$ | $\beta_3$-CT | 82,500 | |
| THD1$^{K402A}$ | $\beta_3$-CT | 45,000 ± 1,400 | |
| THD1$^{K402E}$ | $\beta_3$-CT | 34,000 ± 2,000 | |
| THD1$^{K403A}$ | $\beta_3$-CT | 68,100 ± 400 | |
| THD1$^{K403E}$ | $\beta_3$-CT | 60,900 ± 9,900 | |
| THD1$^{K404A}$ | $\beta_3$-CT | 90,100 ± 3,800 | |
| THD1$^{KK402/403AA}$ | $\beta_3$-CT | 62,100 ± 6,900 | |
| THD1 1-400 | $\beta_3$-CT | 249,800 ± 63,100 | |
| KIND2 | $\beta_1$-CT | 2,900 ± 200[b] | 17,000 ± 1,000[c] 13,400 ± 6,570[c] |
| KIND2 | $\beta_1$-CT$^{Y783A}$ | 4,900 ± 700[b] | |
| KIND2 | $\beta_3$-CT | 9,000 ± 1,800 | |
| KIND2 | $\beta_3$-CT$^{Y772A}$ | 15,500 ± 4,200 | |
| KIND2$^{QW/AA}$ | $\beta_1$-CT | n.d. | |
| KIND2$^{QW/AA}$ | $\beta_3$-CT | n.d. | |
| ΔKIND2 | $\beta_3$-CT | 6,800 ± 2,200 | |
| KIND2$^{Y13A}$ | $\beta_3$-CT | 9,400 ± 400 | |
| THD1 | $\beta_3$-TM-CT in 10% PIP2 | 66 ± 42[d] | |
| THD1 | $\beta_1$-TM-CT in 10% PIP2 | 73 ± 21[d] | |
| THD1 | $\beta_1$-TM-CT$^{Y795A}$ in 10% PIP2 | 98 ± 117[d] | |
| THD1 | $\beta_1$-TM-CT$^{Y783A}$ in 10% PIP2 | 307 ± 262[d] | |
| THD1 | $\alpha_5$-TM in 10% PIP2 | 274 ± 106[d] | 250[e] (781[f], 980[g]) |
| THD1 | $\beta_3$-TM-CT in 10% PIP3 | 83 ± 39[d] | |
| THD1 | $\beta_1$-TM-CT in 10% PIP3 | 92 ± 54[d] | |
| THD1 | $\beta_1$-TM-CT$^{Y795A}$ in 10% PIP3 | 127 ± 37[d] | |
| THD1 | $\beta_1$-TM-CT$^{Y783A}$ in 10% PIP3 | 427 ± 474[d] | |
| THD1 | $\alpha_5$-TM in 10% PIP3 | 411 ± 388[d] | |
| THD1$^{K402E}$ | $\beta_3$-TM-CT in 10% PIP2 | 70 ± 26[d] | |
| THD1$^{K402E}$ | $\beta_1$-TM-CT in 10% PIP2 | 131 ± 65[d] | |
| THD1$^{K402E}$ | $\alpha_5$-TM in 10% PIP2 | 246 ± 28[d] | |
| THD1$^{K402E}$ | $\beta_3$-TM-CT in 10% PIP3 | 51 ± 20[d] | |
| THD1$^{K402E}$ | $\beta_1$-TM-CT in 10% PIP3 | 54 ± 8[d] | |
| THD1$^{K402E}$ | $\alpha_5$-TM in 10% PIP3 | 211 ± 109[d] | |
| KIND2 | $\beta_3$-TM-CT in 10% PIP2 | 95 ± 68[d] | |
| KIND2 | $\beta_1$-TM-CT in 10% PIP2 | 120 ± 59[d] | |
| KIND2 | $\beta_1$-TM-CT$^{Y795A}$ in 10% PIP2 | 370 ± 422[d] | |
| KIND2 | $\beta_1$-TM-CT$^{Y783A}$ in 10% PIP2 | 117 ± 78[d] | |
| KIND2 | $\alpha_5$-TM in 10% PIP2 | 444 ± 447[d] | |
| KIND2 | $\beta_3$-TM-CT in 10% PIP3 | 79 ± 11[d] | |
| KIND2 | $\beta_1$-TM-CT in 10% PIP3 | 102 ± 81[d] | |
| KIND2 | $\beta_1$-TM-CT$^{Y795A}$ in 10% PIP3 | 162 ± 55[d] | |
| KIND2 | $\beta_1$-TM-CT$^{Y783A}$ in 10% PIP3 | 255 ± 141[d] | |
| KIND2 | $\alpha_5$-TM in 10% PIP3 | 471 ± 343[d] | ~ 400[h] |
| KIND2$^{Y13A}$ | $\beta_3$-TM-CT in 10% PIP2 | 219 ± 98[d] | |
| KIND2$^{Y13A}$ | $\beta_1$-TM-CT in 10% PIP2 | 98 ± 24[d] | |
| KIND2$^{Y13A}$ | $\alpha_5$-TM in 10% PIP2 | 162 ± 64[d] | |
| KIND2$^{Y13A}$ | $\beta_3$-TM-CT in 10% PIP3 | 173 ± 56[d] | |
| KIND2$^{Y13A}$ | $\beta_1$-TM-CT in 10% PIP3 | 78 ± 35[d] | |
| KIND2$^{Y13A}$ | $\alpha_5$-TM in 10% PIP3 | 128 ± 65[d] | |
| TLN1-F3–ΔKIND2 | $\beta_1$-CT$^{Y783A}$ | 4,300 ± 700 | |
| TLN1-F3–ΔKIND2 | $\beta_1$-CT$^{Y795A}$ | 470,000 ± 140,000 | |
| TLN1-F3–ΔKIND2 | $\beta_1$-CT: ΔKIND2 contribution | 2,900 ± 400 | |
| | $\beta_1$-CT: TLN1-F3 contribution | 25,300 ± 2,800 | |
| TLN1-F3–ΔKIND2$^{Y13A}$ | $\beta_1$-CT: ΔKIND2$^{Y13A}$ contribution | 2900 ± 400 | |
| | $\beta_1$-CT: TLN1-F3 contribution | 15,700 ± 1,100 | |
| THD1 | $\beta_1$-spacer-CT | 181,000 ± 11,000 | |
| KIND2 | $\beta_1$-spacer-CT | 5,300 ± 200[b] | |

[a]: Value obtained by NMR[19] [b]: Extrapolated, as MST traces of highly concentrated samples were inconsistent and excluded from analysis. [c]: Value obtained by ITC[4,11] [d]: IC$_{50}$ value obtained by FC-RDA [e]: Value obtained by FRET between TLN1-F2F3 domains and 10% PIP2-containing nanodiscs[4] [f]: Value obtained by titrating large unilamellar vesicles containing 10% PIP2 to THD1 using ITC[22,35] [g]: Value obtained for THD1 binding to 10% PIP2 containing monolayers by SPR[22] [h]: Value obtained for kindlin-3-PH domain binding to 10% PIP3 monolayers by SPR[35]

# Reporting Summary

## Statistics

For all statistical analyses, confirm that the following items are present in the figure legend, table legend, main text, or Methods section.

| n/a | Confirmed | |
|---|---|---|
| ☐ | ☒ | The exact sample size (*n*) for each experimental group/condition, given as a discrete number and unit of measurement |
| ☐ | ☒ | A statement on whether measurements were taken from distinct samples or whether the same sample was measured repeatedly |
| ☐ | ☒ | The statistical test(s) used AND whether they are one- or two-sided<br>*Only common tests should be described solely by name; describe more complex techniques in the Methods section.* |
| ☐ | ☒ | A description of all covariates tested |
| ☐ | ☒ | A description of any assumptions or corrections, such as tests of normality and adjustment for multiple comparisons |
| ☐ | ☒ | A full description of the statistical parameters including central tendency (e.g. means) or other basic estimates (e.g. regression coefficient) AND variation (e.g. standard deviation) or associated estimates of uncertainty (e.g. confidence intervals) |
| ☐ | ☒ | For null hypothesis testing, the test statistic (e.g. *F*, *t*, *r*) with confidence intervals, effect sizes, degrees of freedom and *P* value noted<br>*Give P values as exact values whenever suitable.* |
| ☒ | ☐ | For Bayesian analysis, information on the choice of priors and Markov chain Monte Carlo settings |
| ☒ | ☐ | For hierarchical and complex designs, identification of the appropriate level for tests and full reporting of outcomes |
| ☒ | ☐ | Estimates of effect sizes (e.g. Cohen's *d*, Pearson's *r*), indicating how they were calculated |

*Our web collection on statistics for biologists contains articles on many of the points above.*

## Software and code

Policy information about availability of computer code

| Data collection | NMR: Topspin 3.2-3.5 (Bruker)<br>MST: NT Control v2.1.33 (Nanotemper);<br>Flow Cytometry: FACSDiva Software Version 9.0 (BD);<br>CL-MS: Foundation 3.1SP7, Xcalibur 4.3 (Thermo Fisher Scientific);<br>SCFS: NanoWizard control software version 4.3.55 (JPK);<br>Microscopy: Auto 2 (Invitrogen), VisiView 4.0 (Visitron Systems GmbH);<br>Plate reader: SoftMax Pro 7.1 (Molecular Devices);<br>Thermal stability: PR.Therm Control Version 2.1.2 (Nanotemper);<br>Dynamic light scattering: Dynamics 8.0.0.89 (Wyatt);<br>CD spectroscopy: Spectra Manager Version 2.12.00 (Jasco); |
|---|---|
| Data analysis | NMR: Topspin 3.2-3.5 (Bruker), NMR Pipe (NIST IBBR), CcpNmr Analysis 2.5.2 (Skinner et al. J Biomol NMR 66 (2016) 111), TALOS-N (Yang Shen, and Ad Bax, J. Biomol. NMR, 56, 227-241(2013));<br>MST: MO Affinity Analysis v2.3 (Nanotemper);<br>Flow Cytometry: FlowJo 10.6.1, non-linear regression and statistical tests: OriginPro 2019b (OriginLab);<br>CL-MS: Proteome Discoverer (Thermo Fisher Scientific, version 2.5.0.400) with the XlinkX/PD nodes integrated (Klykov et al., 2018), Visualization: XMAS 1.1.1 plug-in (Scheltema lab) in ChimeraX 1.3 (UCSF)<br>SCFS: Data Processing version 4.3.55 (JPK). Statistical tests were performed using Prism (GraphPad Software - Version 8.4.3 (471));<br>Microscopy: ImageJ 1.49 (Wayne Rasband, NIH); focal adhesion analysis server (https://faas.bme.unc.edu/, Shawn Gomez lab), statistical tests: OriginPro 2019b (OriginLab);<br>Plate reader: SoftMax Pro 7.1 (Molecular Devices), statistical tests: OriginPro 2019b (OriginLab); |

Thermal stability: PR.Therm Control Version 2.1.2 (Nanotemper);
Dynamic light scattering: Dynamics 8.0.0.89 (Wyatt)
CD spectroscopy: Spectra Manager Version 2.12.00 (Jasco);

For manuscripts utilizing custom algorithms or software that are central to the research but not yet described in published literature, software must be made available to editors and reviewers. We strongly encourage code deposition in a community repository (e.g. GitHub). See the Nature Portfolio guidelines for submitting code & software for further information.

## Data

Policy information about availability of data

All manuscripts must include a data availability statement. This statement should provide the following information, where applicable:
- Accession codes, unique identifiers, or web links for publicly available datasets
- A description of any restrictions on data availability
- For clinical datasets or third party data, please ensure that the statement adheres to our policy

No data with mandated deposition was generated in this study. The data is available upon request.

## Human research participants

Policy information about studies involving human research participants and Sex and Gender in Research.

| Reporting on sex and gender | N/A |
| --- | --- |
| Population characteristics | N/A |
| Recruitment | N/A |
| Ethics oversight | N/A |

Note that full information on the approval of the study protocol must also be provided in the manuscript.

# Field-specific reporting

Please select the one below that is the best fit for your research. If you are not sure, read the appropriate sections before making your selection.

☒ Life sciences     ☐ Behavioural & social sciences     ☐ Ecological, evolutionary & environmental sciences

For a reference copy of the document with all sections, see nature.com/documents/nr-reporting-summary-flat.pdf

# Life sciences study design

All studies must disclose on these points even when the disclosure is negative.

| Sample size | No statistical method was used to predetermine the sample size. Key experiments were repeated at least three times to allow for a statistical analysis. |
| --- | --- |
| Data exclusions | Only data of insufficient quality was excluded. The quality criteria were: MST: According to manufacturer's recommendation, data points that vary by more than 20% fluorescence intensity compared to the average intensity of the other data points or that show aberrant MST traces, which might be a sign for aggregation, were excluded. FC-RDA: Titrations with insufficient signal-to-noise were not considered. DLS: Scans were filtered with 0.01 baseline value. |
| Replication | All cell culture experiments were performed at least thrice on three individual days. Key cell culture experiments were confirmed with a second cell line. Experiments with nanodiscs were repeated at least thrice with different nanodisc preparations. All affinity measurements were performed with at least two different protein batches. Key affinity measurements were performed by different operators. All replication experiments were successful, except for individual nanodisc preparations that could not be used for FC-RDA experiments due to insufficient loading of TM-CTs. |
| Randomization | The order of contact times in SCFS was randomized for every cell and condition. All other experiments were not randomized as the measurements were in equilibrium and timing was therefore not cruicial. |
| Blinding | Microscopic images for cell spreading were recorded automatically, whereas images for focal adhesion analysis were recorded by a different operator. Blinding in all other experiments was not relevant since all measurements were analyzed after data acquisition and were therefore not susceptible to experimenter bias. |

# Reporting for specific materials, systems and methods

We require information from authors about some types of materials, experimental systems and methods used in many studies. Here, indicate whether each material, system or method listed is relevant to your study. If you are not sure if a list item applies to your research, read the appropriate section before selecting a response.

## Materials & experimental systems

| n/a | Involved in the study |
|-----|----------------------|
| ☐ | ☒ Antibodies |
| ☐ | ☒ Eukaryotic cell lines |
| ☒ | ☐ Palaeontology and archaeology |
| ☒ | ☐ Animals and other organisms |
| ☒ | ☐ Clinical data |
| ☒ | ☐ Dual use research of concern |

## Methods

| n/a | Involved in the study |
|-----|----------------------|
| ☒ | ☐ ChIP-seq |
| ☐ | ☒ Flow cytometry |
| ☒ | ☐ MRI-based neuroimaging |

## Antibodies

| | |
|---|---|
| Antibodies used | anti-β1 integrin (total level, biotinylated): eBioscience, 13-0291-80; anti-β1 integrin (9EG7, extended conformation): PharMingen, 550531; anti-β1 integrin PE: BioLegend, 102207; anti-β3 integrin PE: eBioscience, 12-0611; anti-αV integrin PE: BD, 551187; anti-α5 integrin PE: PharMingen, 557447; anti-GAPDH: Calbiochem, CB1001; Goat anti-mouse IgG HRP conjugate: BioRad, 1721011; anti-rat 647: Invitrogen, A21247; Streptavidin eFluor780: eBioscience, 47-4317-82; Rat IgG1 PE isotype control: PharMingen, 554685; Rat IgG2 PE isotype control: PharMingen, 555844; Hamster IgG PE isotype control: eBioscience, 1091682; anti-kindlin-2 (MAB2617): Merck Millipore; anti-talin-HRP: Santa Cruz, sc-365875; anti-talin: Sigma, T3287. |
| Validation | β1 integrin (clone eBioHMb1-1, 13-0291-80, hamster, eBioscience, Wu et al., Bone, 2008. FC); β1 integrin (clone 9EG7, 550531, rat, PharMingen, Lenter et al., PNAS, 1993. FC); β1 integrin (clone HMβ1-1, 102207, hamster, BioLegend, Noto K, et al., Int Immunol, 1995. FC).; β3 integrin (clone 2C9.G3, 12-0611, hamster, eBioscience, Treese et al., Cytometry Part A, 2008. FC); αV integrin (clone RMV-7, 551187, rat, BD, Bader et al., Cell, 1998. FC); α5 integrin (clone 5H10-27, 557447, rat, PharMingen, Kharbili et al, Oncotarget. 2017. FC); GAPDH (clone 6C5, CB1001, mouse, Calbiochem, Gagarin et al., J. Mol. Cell. Card., 2005. WB); kindlin-2 (MAB2617, mouse, Merck Millipore, Theodosiou et al, eLife 2016. WB); talin-1 (T3287, Sigma, Theodosiou et al, eLife 2016; WB). |

## Eukaryotic cell lines

Policy information about cell lines and Sex and Gender in Research

| | |
|---|---|
| Cell line source(s) | Mouse TLN1/TLN2/KIND1/KIND2 quadruple floxed fibroblast: generated in the lab<br>Mouse TLN1/TLN2 double floxed fibroblast: generated in the lab<br>Mouse ITGB1 knock-out fibroblasts: generated in the lab |
| Authentication | Mouse TLN1/TLN2/KIND1/KIND2 quadruple floxed fibroblasts were generated in our lab from transgenic mice and previously described in J Cell Biol, (2017) 216 (11): 3785.<br>Mouse TLN1/TLN2 double floxed fibroblasts were generated in our lab from transgenic mice and previously described in eLife (2016) 5:e10130.<br>Mouse ITGB1 knock-out fibroblasts were generated in our lab from transgenic mice and previously described in J Invest Dermatol (2013) 133, 2722. |
| Mycoplasma contamination | All cell lines were regularly tested negative for mycoplasma contamination. |
| Commonly misidentified lines<br>(See ICLAC register) | No commonly misidentified cell line was used in this study. |

## Flow Cytometry

### Plots

Confirm that:

☒ The axis labels state the marker and fluorochrome used (e.g. CD4-FITC).

☒ The axis scales are clearly visible. Include numbers along axes only for bottom left plot of group (a 'group' is an analysis of identical markers).

☒ All plots are contour plots with outliers or pseudocolor plots.

☒ A numerical value for number of cells or percentage (with statistics) is provided.

## Methodology

| | |
|---|---|
| Sample preparation | The adhesive cell lines were detached from the culture plates with 500 μL trypsin and EDTA in PBS, trypsin was neutralized with 500 μL DMEM supplemented with 10% FBS, transferred into 4 mL DMEM supplemented with 10% FBS and split in different FACS tubes for antibody staining after washing twice with PBS. |
| Instrument | BD LSRFortessa X-20 |
| Software | FACS data was recorded using: BD FACSDiva Software Version 9.0<br>FACS data was analyzed using: FlowJo 10.6.1 |
| Cell population abundance | Defined cell lines were used. |
| Gating strategy | Cells were gated on the major cell population using an "auto gate" in FlowJo, that included approx. 70-80% of the recorded 10000 - 20000 events. |

☒ Tick this box to confirm that a figure exemplifying the gating strategy is provided in the Supplementary Information.

