## [Peer Review File · Nature Structural & Molecular Biology]

Peer Review Information

Manuscript Title: Talin and kindlin deploy integrin tail allostery and direct binding to activate integrins

Corresponding author name(s): Dr Reinhard Fässler

Reviewer Comments & Decisions:

Decision Letter, initial version:

Message: 5th Jan 2023

Dear Dr. Fässler,

Thank you again for submitting your manuscript "Talin and kindlin combine integrin tail allostery and direct binding to activate integrins". I apologize for the delay in responding – staff shortages and the holiday season caused longer than usual processing times. Nevertheless, we now have comments (below) from the 2 reviewers who evaluated your paper. In light of those reports, we remain interested in your study and would like to see your response to the comments of the referees, in the form of a revised manuscript.

You will see that while both reviewers are generally positive, they bring up concerns which will need to be addressed before we can consider publication further. In line with reviewer #2 comments, we would expect further analysis carried out to strengthen the conclusions of binding experiments. As reviewers #1 and #2 agree, discussion of NMR data calls for clarification, together with more detailed description of structural model building. Additionally, we feel that the additional NMR titration experiments suggested by reviewer #2 would strengthen the conclusions of the manuscript.

Please be sure to address/respond to all concerns of the referees in full in a point-by-point response and highlight all changes in the revised manuscript text file. If you have comments that are intended for editors only, please include those in a separate cover letter.

We expect to see your revised manuscript within 6 weeks. If you cannot send it within this time, please contact us to discuss an extension; we would still consider your revision,

provided that no similar work has been accepted for publication at NSMB or published elsewhere.

Reporting Summary:

Please note that all key data shown in the main figures as cropped gels or blots should be presented in uncropped form, with molecular weight markers. These data can be aggregated into a single supplementary figure item. While these data can be displayed in a relatively informal style, they must refer back to the relevant figures. These data should be submitted with the final revision, as source data, prior to acceptance, but you may want to start putting it together at this point.

SOURCE DATA: we urge authors to provide, in tabular form, the data underlying the graphical representations used in figures. This is to further increase transparency in data reporting, as detailed in this editorial (<http://www.nature.com/nsmb/journal/v22/n10/full/nsmb.3110.html>). Spreadsheets can be submitted in excel format. Only one (1) file per figure is permitted; thus, for multi-paneled figures, the source data for each panel should be clearly labeled in the Excel file; alternately the data can be provided as multiple, clearly labeled sheets in an Excel file.

When submitting files, the title field should indicate which figure the source data pertains to. We encourage our authors to provide source data at the revision stage, so that they are part of the peer-review process.

Data availability: this journal strongly supports public availability of data. All data used in accepted papers should be available via a public data repository, or alternatively, as Supplementary Information. If data can only be shared on request, please explain why in your Data Availability Statement, and also in the correspondence with your editor. Please note that for some data types, deposition in a public repository is mandatory - more information on our data deposition policies and available repositories can be found below: <https://www.nature.com/nature-research/editorial-policies/reporting-standards#availability-of-data>

[Redacted]

Sincerely,

Katarzyna Ciazynska
(she/her)
Associate Editor
Nature Structural & Molecular Biology
<https://orcid.org/0000-0002-9899-2428>

Referee expertise:

Referee #1: integrins, structural biology

Referee #2: adhesion proteins, NMR

Reviewers' Comments:

Reviewer #1:

Remarks to the Author:

A: Summary of the key results

Aretz et al. used combination of biophysical analysis methods, protein engineering and cellular biology assays to show that talin and kindlin co-operate in binding to integrin cytoplasmic domain. They noticed that the presence of kindlin had minor influence on talin head binding to integrin CT, while talin head inhibited kindlin binding. The work resulted in identification of residues responsible for direct interaction between talin and kindlin. Authors also proposed allosteric coupling between talin and kindlin binding sites within integrin b1-CT.

B: Originality

The manuscript addresses key questions related to integrin activation and reveals details of the co-operation between talin and kindlin.

C: Data & Methodology

The work stems from thorough affinity analyses performed with MST, which allowed them to compare binding affinities and binding profiles in various combinations, enabling identification of ternary interactions between talin head, kindlin and integrin cytoplasmic domain. With help of NMR, a binding interface between talin and kindlin was identified and those were confirmed with extensive mutagenesis.

The work resulted in identification of residues responsible for direct interaction between talin and kindlin. Point mutations were introduced to block this interaction and the mutated forms were also introduced into qKO cells lacking both talins 1 and 2 and kindlins 1 and 2. Those cells were then transfected with different combinations of mutated talins and kindlins, confirming the importance of the proposed interaction site.

Authors also proposed allosteric coupling between talin and kindlin binding sites within integrin b1-CT. To decouple the binding events, they introduced integrin CT with a spacer between the talin and kindlin binding sites, resulting in decrease in competition. Such integrin was also introduced into b1-KO mice fibroblasts and was found to impair cell adhesion process and integrin activation.

D: Appropriate use of statistics and treatment of uncertainties

The number of data points and parallel experiments have been reported. Statistical tests

have been used to reveal differences between treatments. I didn't notice anything to complain regarding those.

E: Conclusions: robustness, validity, reliability

Overall, this study provides compelling evidence that talin and kindlin co-operate in integrin binding and activation. The data appears excellent in quality and wide array of up-to-date methods are used.

F: Suggested improvements: experiments, data for possible revision

1) Figure 2D shows that the peaks are disappearing and it is not absolutely clear for all the reader that this is because of broadening of the peaks. Perhaps rephrasing could be considered here?

2) Page 12: The statement "...direct KIND2-binding increases the population..." is somewhat unclear. Consider "Direct KIND2-binding" -> "direct fusion between"

3) Page 13 (lines 356-360): The change is negligible for K798.

4) Page: 14: "...interaction is important for integrin adhesion, whereas the allosteric coupling..." Rephrasing would be good, "whereas" has wrong tone here I think. Maybe replacing "whereas" by "and"?

5) According to Zhang et al. (<https://www.pnas.org/doi/10.1073/pnas.2014583117>), residue K402 is involved in interaction between talin F1 and F3. Therefore, it is possible that FERM-folded talin has lower affinity for kindlin than elongated "open" talin head or talin-F3 alone. This could be discussed, possibly also in the context of Figure 6.

6) The crosslinking data is reported for Talin-F3 ---dKIND2. Did you study crosslinks formed between full talin head and kindlin? Also, the molecular assembly predicted based on talin F3 alone should be evaluated using compact talin as a reference. Would the proposed assembly allow talin and kindlin to occupy integrin CT if talin would exist in compact fold as observed by Zhang et al.

<https://www.pnas.org/doi/10.1073/pnas.2014583117> ?

7) In Figure 3E, the downward shift in dFnorm is not shown, while such effect is observed for both talin and kindlin in Figure S1. Therefore, for transparency, Figure 3E should also include the dFnorm for concentrations in range of 10^5 nM and above to reveal possible downward shift observed. Is the direction of the shift associated with the distance between the fluorescent dye and the binding site within b-CT? Have you tried to perform the measurements with b-CT peptide labelled from another terminus? Alternatively, adding a spacer between peptide terminus and the fluorescent dye could be tried.

8) In Figure 5 panels F and G, it would be important to have a legend for the colours (presumably it is as in Fig. 5A).

9) In Figure 5B, it would be beneficial to show the fluorescence channels separately for better visual assignment.

10) In Figure S1I, please use term KIND2 instead of K2 for consistency.

11) Figure S3G/H: The X-axis scale for the hydrodynamic radius is somewhat odd. At least an insert covering only 0-10 could be included.

12) The MST measurement principle should be explained better. The authors refer to previous paper, which however does not provide any better explanation. Did you use fluorescence quenching or thermophoresis as a primary measure? Sufficient information should be provided to ensure reproducibility of the data.

13) This cross-linker target lysines. The cluster of lysines in the C-terminus of talin head may therefore become overrepresented in the analysis. Authors could discuss this potential limitation of the approach.

14) Discussion, last sentence "Upon dissociation of kindlin, the talin affinity for the β -CT also decreases eventually leading to the dissociation of talin, integrin inactivation and the

initiation of a new cycle of ternary talin• β -CT•kindlin assembly.” Perhaps authors could also consider a model, where kindlin is replaced with another adaptor protein and talin remains bound?

G: References: appropriate credit to previous work?

The key publications are cited and the manuscript provides credits for earlier studies on this topic.

H: Clarity and context: lucidity of abstract/summary, appropriateness of abstract, introduction and conclusions

I found the manuscript fluent and well written.

Reviewer #2:

Remarks to the Author:

Integrins are heterodimeric receptors composed of α and β subunits that contain a ligand binding ectodomain, a transmembrane domain and a cytoplasmic tail. Activation of these transmembrane receptors promotes signaling through a number of pathways, including those that control cell survival, proliferation, differentiation and migration. Talin and kindlin are adaptor proteins that play key roles in activation of integrins, as binding to the β -integrin cytoplasmic tail promotes a conformational change in the integrin from a ‘closed’ inactive to an ‘open’ active state.

The work herein employs NMR spectroscopy, biochemistry and cell biology approaches to investigate talin and kindlin binding interactions with the β -integrin tail. Based on their findings, the investigators propose that binding of talin and kindlin to the β -integrin tail induces a conformational change in the β -integrin tail that increases talin and decreases kindlin affinity. They also propose that weak yet direct talin/kindlin interactions mediate these interactions, as disruption of these interactions compromises integrin function.

While these studies provide new insights into the complex role of talin and kindlin in integrin activation, several concerns exist with the manuscript. First, the manuscript is difficult to read. A number of different constructs are used that differ between the experiments. The rationale for use and the ability to interpret data from different constructs is not clear in many cases. Concerns exist with potential shortcomings using binding data that do not reach saturation for these measurements. The binding data suggests negative cooperativity, that would be helpful to define in the context of Hill plots and coefficients. Some of the methods are described in sufficient detail whereas others are poorly described or not at all. For example, methods used for modeling are missing. Some of the NMR data is not correctly labeled and overlays difficult to interpret. Detailed concerns are listed below.

1. Based on NMR and MST experiments, the investigators state that binding of talin and kindlin to the β -integrin tail induces a conformational change in the β -integrin tail that increases talin and decreases kindlin affinity. However, apparent K_d 's are determined, as many of the binding plots do not reach saturation due to solubility limitations and weak affinity of the adaptor proteins in the absence of the membrane. This raises concerns about K_d determination. Moreover, in the case of cooperativity, the K_d values associated with the equations listed are not the true K_d 's, but rather ‘apparent’ since in a cooperative process there will be multiple true K_d s. It is a bit unclear why asymmetric allosteric

affinity modulation is used to describe the interactions under study and not the more commonly used term, negative cooperativity. Determining a Hill coefficient and putting this in context with other relevant systems would be helpful.

2. In the 2nd equation under section 'talin and kindlin affinities for β -integrin tails', shouldn't 102 be 120?

3. The authors use NMR to identify binding epitopes/residues of talin, kindlin and the integrin b-CT in binary complex and ternary complexes. In addition to chemical shift changes, linewidth differences are noted that are stated as likely due to changes in rotational correlation time upon complex formation. However, changes in line width due to other factors (dynamics, exchange processes, etc) should also be noted.

4. Chemical shift/broadening information is mapped onto the 3D structure in some cases (mostly for talin), but not others. It would be helpful to visualize in 3D the unique vs common residues upon binding to talin or kindlin onto integrin b1-CT as well as on kindlin. The investigators do not make clear why the NMR experiments on kindlin were conducted with KIND2 lacking a flexible loop in F1 and pH.

5. For Figure 1B, the rationale for showing the two selected residues is missing as it seems they are not the most affected residues upon binary and ternary complex formation. The nomenclature for the domains (e.g 565-THD1,etc) in Figure 1I should be defined.

6. Figure 1C and 2D show NMR chemical shift mapping at three molar ratios. The coloring used makes the plot look busy and difficult to follow. Also there is no average peak intensity line in Fig 1C, as noted in the last paragraph of page 5.

7. As talin interactions differ between b1-CT and b3-CT, some discussion is needed regarding whether these differences influence/alter the proposed model.

8. In Figure 2A, as the HSQC spectrum of Talin-F3 and Kindl2 (why is deltaKind2 not used?) does not show perturbation for residue 402. Hence, it is unclear how it is involved in binding. In fact the chemical shift perturbations are quite small and as such, difficult to ascertain whether a specific interaction occurs. This may be confirmed by a titration plot showing saturation binding? The chemical perturbations in Figure 2D are difficult to see given the different ratios and coloring. In Figure 2D, an HSQC of TLN1-F3-K402E is shown, yet a peak assigned to K402?

9. As suggested, binding of talin and kindlin promote allosteric changes in b1-CT which affects the affinity of talin and kindlin. However it is unclear what residue-specific allosteric changes occur, as little structural detail of the allosteric network is provided.

10. The investigators identify key residues that interfere with TLN1-F3 and Δ KIND2 binding (K402E and Y13A, respectively). On page 10, it is stated that "Since four lysine residues in the TLN1-F3 α 1-helix comprise the binding site for KIND2-F0 (Figure S2H), we expected either positively charged or aromatic residues on the KIND2-F0 counter surface as contact site (Figure 2F)". As lysines are positively charged, one might expect negatively charged residues to make interactions?

11. Some of the constructs in the Kd table S1 are not described in the text (THD1-Y395A, L400A, for example).

12. The lipid composition for the nanodisc studies differs between the text (PI, PC) in the results and methods sections. Use of THD1 vs THD2 is not explained (as well as in other sections as well). Talin and Kindlin show very weak affinities to β -CT in the absence of the membrane, which raises questions regarding binding studies in the solution state and how these interactions are influenced by PIP2/PIP3-mediated membrane association. Some discussion is warranted here.

13. The use of tailless $\alpha 5$ (as a likely control) is not explained.

14. The investigators state, 'for KIND2-F0 partial assignments were transferred from Biological Magnetic Resonance databank (entry 30659)'. It is unclear what partial assignment means and how this impacts NMR data interpretation.

15. The investigators should describe how the model was generated as well as the associated protein-protein interactions (Figures 2E and F) in more detail, as there is little discussion of the binary and ternary modeling in Figure 2E and F (with no methods described for the modeling). How did the authors use crosslinking data to build a model of the ternary complex? Was chemical crosslinking data incorporated as distance constraints in molecular docking tools such as HADDOCK to generate a 3D model of ternary complex with physiologically relevant inter-protein orientations? Was NMR chemical shift perturbation data used to generate the model and if so, how? How does the NMR data correlate with the crosslinking data?

Author Rebuttal to Initial comments

Point-by-point rebuttal to the comments raised by Reviewer #1

Suggested improvements: experiments, data for possible revision

1) Figure 2D shows that the peaks are disappearing and it is not absolutely clear for all the reader that this is because of broadening of the peaks. Perhaps rephrasing could be considered here?

Thank you for this relevant point and suggestion. We rephrased the text to: "Almost all NMR signals in the TLN1-F3-WT spectra experience severe line broadening beyond detection in the presence of 2-10-fold molar excess of Δ KIND2. Under these conditions the significant increase of molecular weight of the TLN1-F3• Δ KIND2 complex (from 11.6 kDa to 65.1 kDa) and potential dynamics in the binding interface lead to substantial line-broadening beyond detection. However, the NMR signals in the TLN1-F3-K402E spectra remain largely unaffected upon Δ KIND2 titration, consistent with the significantly reduced binding of TLN1-F3-K402E to Δ KIND2 (Figure 2D)."

2) Page 12: The statement "...direct KIND2-binding increases the population..." is somewhat unclear. Consider "Direct KIND2-binding" -> "direct fusion between"

We agree that this should be better phrased and changed the text to: "To determine whether the association with kindlin induces an allosteric activation of talin, followed by an increased talin affinity for the β 1-CT during ternary complex formation, we titrated the TLN1-F3 \leftrightarrow Δ KIND2-Y13A fusion protein (to increase the probability of the kindlin-talin association) to β 1-CT (Figure 3H). We observed identical binding curves as in titration experiments with the wildtype TLN1-F3 \leftrightarrow Δ KIND2 fusion protein binding to β 1-CT (Figure 3G). This result was confirmed in MST affinity measurements, which showed that the THD1 affinity for β -CTs remained unchanged in the presence of the β -CT-binding-deficient KIND2-Q614A-W615A or KIND2-F0 (Figure 3I,J) indicating that the association of TLN1 and KIND2 increases the population of the ternary talin• β -CT•kindlin complex, but not the THD1 affinity for β -CTs in the ternary complex."

3) Page 13 (lines 356-360): The change is negligible for K798.

We agree and changed the text to: "Whereas addition of TLN1-F3 or Δ KIND2 alone induced minor changes, addition of both, TLN1-F3 and Δ KIND2 induced ^{13}C secondary chemical shifts pointing to increased α -helical conformation in the region between H758 and K774 and more extended β -type conformation in the region between W775 and V791."

4) Page: 14: "...interaction is important for integrin adhesion, whereas the allosteric coupling..." Rephrasing would be good, "whereas" has wrong tone here I think. Maybe replacing "whereas" by "and"?

Thank you for pointing this out. We changed the text to: "The cellular experiments demonstrate that the population of ternary talin• β -integrin•kindlin complexes facilitated by the direct talin-kindlin interaction is important for integrin-mediated adhesion and the allosteric coupling mediated by the binding of talin and kindlin to the β -CT."

5) According to Zhang et al. (<https://doi.org/10.1073/pnas.2014583117>) residue K402 is involved in interaction between talin F1 and F3. Therefore, it is possible that FERM-folded talin has lower affinity for kindlin than elongated "open" talin head or talin-F3 alone. This could be discussed, possibly also in the context of Figure 6.

This is an excellent comment and is discussed (see response to comment-6).

6) The crosslinking data is reported for Talin-F3 ---dKIND2. Did you study crosslinks formed between full talin head and kindlin? Also, the molecular assembly predicted based on talin F3 alone should be evaluated using compact talin as a reference. Would the proposed assembly allow talin and kindlin to occupy integrin CT if talin would exist in compact fold as observed by Zhang et al. (<https://doi.org/10.1073/pnas.2014583117>)?

We performed crosslinking between full THD1, KIND2 and β 1-CT in solution. However, since the affinities between these proteins are very low, we had to apply high concentrations resulting in too many false positive crosslinks. To be able to analyze the ternary talin• β 1-CT•kindlin complex by XL-MS, we had to use fusion proteins and GraFix as described in our study. We prepared a THD1 \leftrightarrow β 1-CT fusion protein and crosslinked it with KIND2 in excess via GraFix. However, in these conditions many crosslinks were observed in the flexible loops of THD1 and KIND2 and between the KIND2-F0 domain and β 1-CT, contradicting published crystal structures. Therefore, we decided to use the minimal constructs TLN1-F3 \leftrightarrow β 1-CT and Δ KIND2 for our study, which resulted in reasonable amounts of crosslinks with almost no contradictions to published structures.

As requested by the reviewer, we analyzed the intramolecular THD1 crosslinks from our XL-MS experiments with THD1 \leftrightarrow β 1-CT fusion protein and KIND2 described above and indeed identified crosslinks that are consistent with the presence of both, linear and cloverleaf-like THD1 conformation. This observation suggests that the THD adopts an equilibrium between the cloverleaf-like and linear THD1 conformations, which was also reported by the IZARD lab for THD2 (10.1074/jbc.RA119.010789). It is thus not possible to conclude from our data whether KIND2 favors one of these two conformations. However, we discuss this possibility in our revised discussion.

Structural modeling (see response to comment 15 of reviewer 2) was performed with THD in both conformations and the kindlin interaction site indeed clashes with the F1 and F3 interaction site. We added a supplementary figure (Fig. S6) to support the revised discussion regarding the comments of reviewer 1 (comment 5, 6) and reviewer 2 (comment 12).

7) In Figure 3E, the downward shift in dF_{norm} is not shown, while such effect is observed for both talin and kindlin in Figure S1. Therefore, for transparency, Figure 3E should also include the dF_{norm} for concentrations in range of 10^4 nM and above to reveal possible downward shift observed. Is the direction of the shift associated with the distance between the fluorescent dye and the binding site within b-CT? Have you tried to perform the measurements with b-CT peptide labelled from another terminus? Alternatively, adding a spacer between peptide terminus and the fluorescent dye could be tried.

Thank you for this comment. First, we did not exclude data in Fig. 3E as we measured up to 75 μ M of TLN1-F3 \leftrightarrow Δ KIND2, which is about 15-fold over K_d for β 1-CT-Y783A and did not detect a bell-shaped binding curve. Titration of KIND2 to β 1-CT (Fig. S1H) showed the onset of a bell-

shaped binding curve already at 20 μM , and at 75 μM the difference between measured and expected ΔF_{norm} became clearly obvious. The downward shift of ΔF_{norm} values observed in the titration of TLN1-F3 \leftrightarrow Δ KIND2 to β 1-CT (Fig. 3G) is therefore not an artefact of a bell-shaped binding behavior of the KIND2 binding site.

Secondly, we performed MST experiments with a truncated, N-terminally labeled β 1-CT peptide harboring only the kindlin binding site (ATTO488-KSAVTTVVNPKYEGK, Figure 1 for the reviewer, orange). This peptide shows a similar affinity for KIND2 as the ATTO488-labelled full-length β 1-CT peptide (Fig. 1A for the reviewers). However, KIND2 binding quenches the fluorescent signal.

Thirdly, addition of a C-terminal GSSGGSSG linker to the truncated peptide, reduces KIND2 affinity by 2-3-fold (from 5.5 μM to 12.4 μM ; Fig. 1B for the reviewers), which is in line with a previous publication demonstrating that the C-terminal PDZ domain of β 1-CT is required for kindlin binding (Fitzpatrick et al. (2014) JBC 289(16):1183; DOI: 10.1074/jbc.M113.535369). Altogether, these data preclude C-terminal labeling of β 1-CT for affinity measurements.

Fourthly, an alternative approach to address the reviewer's comment was labelling lysine-784 of the β 1-CT peptide which is located C-terminal to the talin binding site to make use of the fluorescent quenching upon KIND2 binding to the truncated β 1-CT. However, substituting lysine-784 for e.g. alanine reduced talin affinity from 150 μM to 240 μM , which was also reported by Fitzpatrick et al. (JBC, 289:11193,2014; doi: 10.1074/jbc.M113.535369), and even worse, interferes with the talin and kindlin interplay (see also response to reviewer 2-comment 9), as the talin affinity decreases to about 630 μM in the presence of KIND2 (Fig. 1C for the reviewers). Hence, we decided against performing further experiment with this peptide.

Finally, we synthesized a full-length β 1-CT peptide containing an N-terminal GSSGGSSG linker between the β 1-CT and the fluorophore to separate the fluorophore from the talin and kindlin binding sites. However, the binding curves are identical to the β 1-CT peptides used in our study despite reaching a lower maximal response (Fig. 1D for reviewers). These bell-shaped binding curves observed of ATTO488- β 1-CT in presence of KIND2 might, e.g. originate in a second binding site with low affinity, multimerization of KIND2 at higher concentrations, hydrodynamic nonideality or other unknown physical phenomena and are therefore, inherent to this system (Scheuermann et al. (2016) Anal Biochem, doi: 10.1016/j.ab.2015.12.013). Overall, the affinities of KIND2 binding to the three different β 1-CT constructs (direct N-terminal label, truncated peptide, N-terminal GSSG-linker) are similar (Fig. 1E for reviewers) and in the same affinity range as obtained by ITC for KIND2• β 1-CT published by Li et al. (2017) PNAS (doi: 10.1073/pnas.1703064114) suggesting that extrapolating the K_d values for KIND2 binding to ATTO488-labeled β 1-CT is reasonably accurate.

Figure 1 for the reviewers: (A) MST measurements of KIND2 binding to a truncated, N-terminally labeled β 1-CT peptide (ATTO488-KSAVTTVVNPKYEGK, orange) containing the kindlin binding site only. Increasing concentrations of KIND2 lead to a quenching of the fluorescence signal, which was fitted to a one-site-binding model. (B) The truncated, N-terminally labeled β 1-CT peptide (ATTO488-KSAVTTVVNPKYEGK, orange) with an additional C-terminal GSSGSSG linker (ATTO488-KSAVTTVVNPKYEGKGSSGSSG, red) reduced affinity for KIND2 ($K_d = 5.5 \pm 1.9 \mu\text{M}$ vs. $K_d = 12.4 \pm 0.3 \mu\text{M}$). (C) Titration of THD1 to ATTO488-labeled β 1-CT-K784A in the presence (orange) or absence (blue) of $30 \mu\text{M}$ KIND2 (corresponding to 6-fold excess of K_d , $n=2$). (D) Titration of KIND2 to ATTO488-labeled β 1-CT with (purple) and without (green) an N-terminal GSSGSSG linker between fluorophore and β 1-CT peptide. Only concentrations up to about $10 \mu\text{M}$ KIND2 were considered for the non-linear regressions which resulted in K_d values of $K_d = 3.7 \pm 0.6 \mu\text{M}$ and $K_d = 2.9 \pm 0.2 \mu\text{M}$ for the ATTO488-labeled β 1-CT with (purple) and without (green) an N-terminal GSSGSSG linker, respectively. The non-considered data points are shown in grey. (E) Comparison between KIND2 titrations to ATTO488-labeled β 1-CT with (purple) and without (green) the N-terminal GSSGSSG linker and the truncated β 1-CT KIND2 binding peptide (ATTO488-KSAVTTVVNPKYEGK, orange).

8) In Figure 5 panels F and G, it would be important to have a legend for the colours (presumably it is as in Fig. 5A).

We show a more detailed x-axis labeling in the revised figure panels.

9) In Figure 5B, it would be beneficial to show the fluorescence channels separately for better visual assignment.

We show the individual channels for TLN1-YPet and mCherry-KIND2 in the revised Fig. 5B.

10) In Figure S11, please use term KIND2 instead of K2 for consistency.

Absolutely, the figure was revised accordingly.

11) Figure S3G/H: The X-axis scale for the hydrodynamic radius is somewhat odd. At least an insert covering only 0-10 could be included.

The x-axis scale was chosen to demonstrate that the protein sample is monodisperse and free of aggregates, which are characterized by broad peaks with hydrodynamic radii above 10 nm. To further demonstrate that WT and mutant proteins have the same hydrodynamic radii, we show the bar chart of the average hydrodynamic radii from three independent measurements, whereas the overlay of the DLS measurements is a representative measurement. We revised the figure caption accordingly to highlight this difference.

12) The MST measurement principle should be explained better. The authors refer to previous paper, which however does not provide any better explanation. Did you use fluorescence quenching or thermophoresis as a primary measure? Sufficient information should be provided to ensure reproducibility of the data.

We added the following text: “Next, we determined the talin and kindlin binding mode to β 1-CT (*scheme 1*) by microscale thermophoresis¹⁸ (MST), which quantifies temperature-induced changes in fluorescence that are induced by a temperature-related intensity change (TRIC) as well as thermophoresis of a fluorescently labelled probe. The extent of TRIC due to ligand binding and thermophoresis due to size, charge and solvation entropy differences were used to quantify binding affinities in titration experiments. To minimize heat effects, we measured the fluorescence changes only 1.5 s before and after turning on the infrared laser (Fig. S1F).

13) This cross-linker target lysines. The cluster of lysines in the C-terminus of talin head may therefore become overrepresented in the analysis. Authors could discuss this potential limitation of the approach.

We applied XL-MS to identify the talin-binding site on kindlin after we identified the kindlin-binding site on talin. The potential overrepresentation of crosslinks in the C-terminal TLN1 region turned out to be very helpful for identifying the corresponding binding site on kindlin. Furthermore, we carefully controlled the crosslinking experiment by using the THD1-K402E mutant as ‘negative’ control.

14) Discussion, last sentence “Upon dissociation of kindlin, the talin affinity for the β -CT also decreases eventually leading to the dissociation of talin, integrin inactivation and the initiation of a new cycle of ternary talin• β -CT•kindlin assembly.” Perhaps authors could also consider a model, where kindlin is replaced with another adaptor protein and talin remains bound?

We believe that this request, although very interesting, is beyond the scope of this study, because it becomes very complicated once additional FA proteins are used. We mention this at the end of the revised discussion.

Point-by-point rebuttal to the comments raised by Reviewer #2:

Remarks to the Author:

Integrins are heterodimeric receptors composed of α and β subunits that contain a ligand binding ectodomain, a transmembrane domain and a cytoplasmic tail. Activation of these transmembrane receptors promotes signaling through a number of pathways, including those that control cell survival, proliferation, differentiation and migration. Talin and kindlin are adaptor proteins that play key roles in activation of integrins, as binding to the β -integrin cytoplasmic tail promotes a conformational change in the integrin from a 'closed' inactive to an 'open' active state.

The work herein employs NMR spectroscopy, biochemistry and cell biology approaches to investigate talin and kindlin binding interactions with the β -integrin tail. Based on their findings, the investigators propose that binding of talin and kindlin to the β -integrin tail induces a conformational change in the β -integrin tail that increases talin and decreases kindlin affinity. They also propose that weak yet direct talin/kindlin interactions mediate these interactions, as disruption of these interactions compromises integrin function.

While these studies provide new insights into the complex role of talin and kindlin in integrin activation, several concerns exist with the manuscript. First, the manuscript is difficult to read. A number of different constructs are used that differ between the experiments. The rationale for use and the ability to interpret data from different constructs is not clear in many cases. Concerns exist with potential shortcomings using binding data that do not reach saturation for these measurements. The binding data suggests negative cooperativity, that would be helpful to define in the context of Hill plots and coefficients. Some of the methods are described in sufficient detail whereas other are poorly described or not at all. For example, methods used for modeling are missing. Some of the NMR data is not correctly labeled and overlays difficult to interpret. Detailed concerns are listed below.

Thank you for appreciating that our work contributes novel molecular insight into integrin activation. We have put a lot of efforts to change text and enhancing the clarity of the revised manuscript. The rationale for using and the interpretation of data from different constructs is addressed to comments raised below. It is important to note, however, that the MST experiments describing the unidirectional talin-mediated kindlin competition, have been performed with all constructs used in this study including the β 1-CT, β 3-CT, THD1, THD2, TLN1-F3, TLN2-F3, KIND2 and Δ KIND2. The experiments with these constructs consistently demonstrated the unidirectional talin-mediated kindlin competition, which strongly indicates that the results obtained with the different protein constructs are comparable with each other and consistent.

See the response to the remaining questions below.

1. Based on NMR and MST experiments, the investigators state that binding of talin and kindlin to the β -integrin tail induces a conformational change in the β -integrin tail that increases talin and decreases kindlin affinity. However, apparent K_d 's are determined, as many of the binding plots do not reach saturation due to solubility limitations and weak affinity of the adaptor proteins in the absence of the membrane. This raises concerns about K_d determination. Moreover, in the case of cooperativity, the K_d values associated with the

equations listed are not the true K_d 's, but rather 'apparent' since in a cooperative process there will be multiple true K_d s. It is a bit unclear why asymmetric allosteric affinity modulation is used to describe the interactions under study and not the more commonly used term, negative cooperativity. Determining a Hill coefficient and putting this in context with other relevant systems would be helpful.

We agree that we obtained apparent K_d values and that multiple true K_d values exist. As suggested by the reviewer, we determined a Hill coefficient for the titrations of KIND2 to β 3-CT in presence and absence of THD1 and THD2 shown in Fig. 1F. The Hill coefficient was 1.08 ± 0.06 for KIND2 alone, 0.89 ± 0.09 in presence of THD1 and 1.00 ± 0.08 in presence of THD2. We conclude that the asymmetric talin-mediated kindlin competition observed in our study does not point to negative cooperativity according to the Hill model but follows a more complicated model that we aim to analyze in detail in the future. We mention this in the new final paragraph of the revised discussion.

2. In the 2nd equation under section 'talin and kindlin affinities for β -integrin tails', shouldn't 102 be 120?

The values in the equation are the same as the values in Fig. 1G. They should indeed be 102 μ M.

3. The authors use NMR to identify binding epitopes/residues of talin, kindlin and the integrin β -CT in binary complex and ternary complexes. In addition to chemical shift changes, linewidth differences are noted that are stated as likely due to changes in rotational correlation time upon complex formation. However, changes in line width due to other factors (dynamics, exchange processes, etc) should also be noted.

We fully agree and apologize for not stating this more clearly. We changed the text to: "Of note, the line-width of an NMR signal is related to the molecular weight of the protein tumbling in solution. The line-broadening observed during complex formation thus will reflect the increase in the molecular weight of the complex. In addition, the stability of the complex, i.e the binding off-rate and local conformational dynamics due to interaction with the binding partners, can provide additional contributions to the line-width. The NMR signal of residues in β 1-CT undergoing the greatest line-broadening are those in the talin or kindlin binding sites, and the extent of the line-broadening reflects the molecular weight of the complex, suggesting that this is the main reason for the observed line-broadening, although we cannot exclude additional contributions from binding kinetics or conformational dynamics."

4. Chemical shift/broadening information is mapped onto the 3D structure in some cases (mostly for talin), but not others. It would be helpful to visualize in 3D the unique vs common residues upon binding to talin or kindlin onto integrin β 1-CT as well as on kindlin. The investigators do not make clear why the NMR experiments on kindlin were conducted with KIND2 lacking a flexible loop in F1 and pH.

We agree that mapping the chemical shifts/broadening information on the β 1-CT structure would be much more intuitive and easier to visualize. However, there is no β 1-CT structure available.

The Δ KIND2 lacking the flexible loop and the TLN1-F3 were used to minimize molecular weights as much as possible.

5. For Figure 1B, the rationale for showing the two selected residues is missing as it seems they are not the most affected residues upon binary and ternary complex formation. The nomenclature for the domains (e.g 565-THD1, etc.) in Figure 1I should be defined.

The two NMR signals illustrate the effects as representative examples to provide a clearer presentation, also considering the broad audience of NSMB. Given their proximity to the NPxY motifs, we used the two glycine signals; they report on local changes upon talin and kindlin binding, which was not the case for the NMR signals of the talin and kindlin binding sites as they are affected by additional line-broadening due to kinetics of the binding ($k_{\text{off}}+k_{\text{on}}$). Furthermore, the selected signals are in less crowded spectral regions, so their 1D traces taken from the 2D spectra can be readily analyzed.

The nomenclature in Fig. 1I is now mentioned in the revised Figure legends.

6. Figure 1C and 2D show NMR chemical shift mapping at three molar ratios. The coloring used makes the plot look busy and difficult to follow. Also there is no average peak intensity line in Fig 1C, as noted in the last paragraph of page 5.

We have removed the statement about the average peak intensity line in the text, and in fact do not show an average intensity line. To improve the visual representation in Figure 1C, we moved the plot showing the full data set to the Supplemental Figure 1 and only show one molar ratio in Figure 1C. For the spectra in Fig. 2D we feel that the overlays at these molar ratios highlight the line-differential broadening between the TLN1-F3 and TLN1-F3-K402E. We believe that the representation of the data using a consistent color coding used for specific proteins helps visualizing the effects.

7. As talin interactions differ between β 1-CT and β 3-CT, some discussion is needed regarding whether these differences influence/alter the proposed model.

As we observed the unidirectional competition of kindlin by talin for both β 1-CT and β 3-CT in solution and model membrane, we think that the proposed model is valid for both integrin isoforms. We highlight this point in the revised discussion:

“Our data indicate a complex mechanism underlying talin and kindlin cooperativity, and reveal important insight explaining how their cooperativity enables the strengthening of β 1 and β 3 integrin-cytoskeletal linkages required for the transmission of high actomyosin-generated forces to ligand-bound integrins of adherent cells, and at the same time remaining highly dynamic (Fig. 6B).”

“Interestingly, the low population of ternary complexes is in line with single molecule force measurements of RGD-bound α V β 3 and α 5 β 1 integrins in cells³⁴.”

8. In Figure 2A, as the HSQC spectrum of Talin-F3 and Kindl2 (why is deltaKind2 not used?) does not show perturbation for residue 402. Hence, it is unclear how it is involved in binding. In fact the chemical shift perturbations are quite small and as such, difficult to ascertain whether a specific interaction occurs. This may be confirmed by a titration plot showing saturation binding? The chemical perturbations in Figure 2D are difficult to see given the

different ratios and coloring. In Figure 2D, an HSQC of TLN1-F3-K402E is shown, yet a peak assigned to K402?

Thank you for the comments. Note, that the chemical shift changes in Fig 2A are monitored at sub-stoichiometric ratio of kindlin as the KIND2 solubility is limited. Assuming an affinity between talin and kindlin of $K_d = 300 \mu\text{M}$ (see Fig. 3 for the reviewers), the fraction of bound TLN1-F3 is about 22% under these conditions, thus the complex is not fully populated. The severe line-broadening in the presence of excess of kindlin (Fig. 2D) prevents analysis of chemical shift changes. In the latter titration, ΔKIND2 was used here, as it has higher solubility. Regarding the spectral changes observed for the amide proton of K402 in TLN1-F3 (Fig. 2A, D) we note that the lysine side chain that likely mediates electrostatic interactions is quite far away from the amide proton, which is monitored in NMR spectra shown. Moreover, chemical shift changes sense changes in the local electronic environment of the nuclear spin. It is thus possible that differential contributions due to direct contacts with the binding partner and changes in local conformation may cancel each other out and thus lead to small overall effects. In addition, the amide proton of K402 is part of a helical conformation and is not expected to directly mediate contacts between talin and kindlin – thus, large chemical shift changes are not expected. At excess of ΔKIND2 (Fig. 2D) substantial line-broadening is observed, consistent with complex formation.

The importance of K402 for the kindlin interaction is validated by our mutational analysis, where no line-broadening is observed for the K402E mutation (Fig. 2D top/bottom), demonstrating that the K402 side chain is important for the interaction.

For the initial NMR titration of isotope-labeled TLN1-F3 with KIND2, we used full length kindlin-2 protein, as the TLN1 binding site in KIND2 was not defined at this stage of the project. As requested by the reviewer, we performed a titration experiment (see Figure 2 for the reviewer) but were unable to reach saturation binding before the peaks disappear due to line broadening. Therefore, a detailed analysis of the K_d value between talin and kindlin is not possible by NMR.

Figure 2 for the reviewers: Overlay of ^1H - ^{15}N HSQC NMR spectra of 200 μM uniformly ^{15}N -labeled TLN1-F3 in absence (black) and presence of 75 μM (blue), 150 μM (green) and 285 μM (red) KIND2.

To estimate a K_d value and confirm the low affinity between talin and kindlin, we performed additional MST measurements. Since labeled THD1 as well as labelled KIND2 were not suitable for MST measurements, we titrated the integrin-binding-deficient KIND2-QW/AA to ATTO488-labeled $\beta 3$ -CT in the absence (yellow) and presence of THD1 (purple), TLN1-F3-K402A (blue) and TLN1-F3-K402E (brown) (see Figure 3 for the reviewers). Note that only the presence of THD1 induced MST signal upon KIND2-QW/AA titration to $\beta 3$ -CT. However, saturation binding could not be achieved as the maximal kindlin solubility was insufficient. The extrapolated K_d of KIND2 for THD1 is in the range of about 250 - 300 μM .

Figure 3 for the reviewers: A) MST affinity measurements of KIND2 (green) and integrin binding-deficient KIND2-614QW/AA615 for ATTO488-labeled $\beta 3$ -CT in absence (yellow) or presence (purple) of 400-450 μM THD1 ($n=2$). B) MST affinity measurements of integrin binding-deficient KIND2-614QW/AA615 for ATTO488-labeled $\beta 3$ -CT in absence (yellow) or presence of 400-450 μM THD1 (purple), 230 μM THD1-K402A (blue) and 250 μM THD1-K402E (brown; all $n=2$).

We thank the reviewer for making us aware of the incorrect labeling of K402. We corrected the assignment in the revised Fig. 2D.

9. As suggested, binding of talin and kindlin promote allosteric changes in β 1-CT which affects the affinity of talin and kindlin. However, it is unclear what residue-specific allosteric changes occur, as little structural detail of the allosteric network is provided.

This is an excellent question, that we began to address. Briefly, we used the results from the NMR line broadening experiment of our study (Fig. 1C) to perform an alanine scan of the β 1-CT in regions that were extraordinarily perturbed in ternary complex titrations to identify residues that might play a role in the allosteric communication between talin and kindlin. First, we measured the affinity of KIND2 towards these mutants. We did not identify significant changes in affinity for kindlin suggesting that the allosteric coupling is activated upon simultaneous binding of THD1 and KIND2.

Next, we measured the THD1 affinity for the β 1-CT mutants in the absence and presence of KIND2 and found that certain substitutions between the talin and kindlin NPxY motifs reduce talin affinity in the presence of kindlin (comparable to the results we obtained for the THD1-K402E or KIND2-Y13A mutations shown in Fig. 3 of our manuscript). The most interesting pair of mutations that we discovered in this mutational analysis was β 1-CT-S785A and -S785D. Whereas the apparent THD1 affinity for β 1-CT-S785A was unaffected by the presence of KIND2, the apparent THD1 affinity for β 1-CT-S785D significantly decreased by the presence of KIND2.

To test the *in vivo* consequences for such a change in talin affinity for β 1-CT-S785D, we generated a mouse model carrying the β 1-S785D substitution in the germline. Whereas mice carrying the β 1-S785A substitution are perfectly normal, mice carrying the β 1-S785D substitution lack kidneys. Our preliminary data indicate that the ureteric bud outgrowth is severely compromised in β 1-S785D mice. Importantly, we could confirm with a series of *in vivo* and *ex vivo* experiments that the kidney defect is caused by reduced talin affinity and not by integrin tail modification such as serine-785 phosphorylation. We envisage to complete the study within the next 12 months.

10. The investigators identify key residues that interfere with TLN1-F3 and Δ KIND2 binding (K402E and Y13A, respectively). On page 10, it is stated that “Since four lysine residues in the TLN1-F3 α 1-helix comprise the binding site for KIND2-F0 (Figure S2H), we expected either positively charged or aromatic residues on the KIND2-F0 counter surface as contact site (Figure 2F)”. As lysines are positively charged, one might expect negatively charged residues to make interactions?

Thank you for making us aware of the mistake. The text was changed accordingly.

11. Some of the constructs in the Kd table S1 are not described in the text (THD1-Y395A, L400A, for example).

We rechecked Table S1 thoroughly and could not find constructs that were used but not described in the manuscript. The constructs THD1-Y385A and L400A are in Table S1 and shown in Fig. S2H. They were part of the MST-based screen to identify the talin-kindlin interaction site.

12. The lipid composition for the nanodisc studies differs between the text (PI, PC) in the results and methods sections. Use of THD1 vs THD2 is not explained (as well as in other sections as well). Talin and Kindlin show very weak affinities to b-CT in the absence of the membrane, which raises questions regarding binding studies in the solution state and how these interactions are influenced by PIP2/PIP3-mediated membrane association. Some discussion is warranted here.

Talin is produced in two isoforms, talin-1 and talin-2. The latter is highly expressed in skeletal muscle and heart. We included the head domain of talin-2 (THD2) in MST measurements to demonstrate that the unidirectional competition is mediated by both, THD1 as well as THD2. We revised the Material and Methods section: "Lipids were dissolved in chloroform (DMPC), 20:9:1 chloroform:methanol:water (PIP2) or 80:40:1 chloroform:methanol:HCl (PIP3) to a final concentration of 50 mg mL⁻¹ (DMPC) or 10 mg mL⁻¹ PIP2 or PIP3, which yields a final concentration of 10% PIP2 and 90% DMPC or 10% PIP3 and 90% DMPC to obtain molar ratios of 1:9 PIP2:DMPC and 1:9 PIP3:DMPC, respectively."

Furthermore, we discuss that talin and kindlin bind much stronger to phospholipids than to the integrin tail. It is important to note, however, that our experiments with nanodiscs clearly demonstrate that membrane binding does not interfere with allosteric coupling via the β -CT peptide tail.

13. The use of tailless $\alpha 5$ (as a likely control) is not explained.

To explain it we revised the text in the result section to: "As the tailless $\alpha 5$ -TM interacts with neither talin nor kindlin, these findings indicate that the charged lipids contribute the largest binding energy for talin and kindlin whereas integrin tails make a minor contribution (see Table S1), which is in line with observations reported for THD1 and $\beta 3$ -CT²⁰."

14. The investigators state, 'for KIND2-F0 partial assignments were transferred from Biological Magnetic Resonance databank (entry 30659)'. It is unclear what partial assignment means and how this impacts NMR data interpretation.

The assignments for the NMR signals shown in Figure 2H and 2I were transferred and used accordingly. For most signals chemical shifts are in excellent agreement and only those were analyzed. We believe that this approach is adequate and does not impact the NMR data interpretation.

15. The investigators should describe how the model was generated as well as the associated protein-protein interactions (Figures 2E and F) in more detail, as there is little discussion of the binary and ternary modeling in Figure 2E and F (with no methods described for the modeling). How did the authors use crosslinking data to build a model of the ternary complex? Was chemical crosslinking data incorporated as distance constraints in molecular docking tools such as HADDOCK to generate a 3D model of ternary complex with physiologically relevant inter-protein orientations? Was NMR chemical shift perturbation data used to generate the model and if so, how? How does the NMR data correlate with the crosslinking data?

We added the following text to the Material and Methods part:

Generation of the structural model

Since structural data are unavailable for critical regions of the proteins used in our study, we generated the structural model manually. To this end, we used the crystal structure of the β 1D-CT•TLN2 complex (PDB: 3G9W) because the structures of neither β 1A-CT nor in complex with TLN1 (β 1A-CT•TLN1) have been solved yet. Since the kindlin binding site in β 1D-CT differs from β 1A-CT and is not resolved in the structure, we used the β 1A-CT• Δ KIND2 crystal structure (PDB: 5XQ0) and aligned the resolved amino acids of β 1A-CT• Δ KIND2 and β 1D-CT•TLN2. Furthermore, structural information of the flexible N-terminus of the KIND2-F0 domain is also not resolved in any of the published KIND2 crystal structures. Therefore, we aligned the NMR structures of the individual KIND2-F0 domain (PDB: 6U4N) with the model. Subsequently, we mapped the intra- and intermolecular crosslinks obtained in our XL-MS experiments on these assembled published structures (TLN2-F2F3• β 1D⁹, Δ KIND2• β 1A³ and KIND2-F0⁴¹) using the XMAS plugin⁴² for ChimeraX⁴³. We colored crosslinks of relevant distance of 5-30 Å in yellow and all remaining crosslinks in red and then iteratively adjusted the orientation of TLN1, KIND2 and β 1-CT towards each other until the maximal number of crosslinks were of relevant distance.

Decision Letter, first revision:

Message: Our ref: NSMB-A46550B

2nd Jun 2023

Dear Dr. Fässler,

Thank you for submitting your revised manuscript "Talin and kindlin combine integrin tail allostery and direct binding to activate integrins" (NSMB-A46550B). It has now been seen by the original referees and their comments are below. The reviewers find that the paper has improved in revision, and therefore we'll be happy in principle to publish it in Nature Structural & Molecular Biology, pending minor revisions to satisfy the referees' final requests and to comply with our editorial and formatting guidelines.

To facilitate our work at this stage, it is important that we have a copy of the main text as a word file. If you could please send along a word version of this file as soon as possible, we would greatly appreciate it; please make sure to copy the NSMB account (cc'ed above).

Sincerely,

Katarzyna Ciazynska
(she/her)
Associate Editor
Nature Structural & Molecular Biology
<https://orcid.org/0000-0002-9899-2428>

Reviewer #1 (Remarks to the Author):

The revised manuscript and additional information provided for the reviewers addressed the questions raised.

Reviewer #2 (Remarks to the Author):

In the resubmission of the manuscript entitled 'Talin and kindlin combine integrin tail allostery and direct binding to activate integrins', the majority of the comments were corrected as requested in the updated manuscript. However, some minor concerns re:

reviewer 2 should be addressed prior to publication.

Comment 4

- While the authors address the comment in the rebuttal, the explanation does not appear to be in the text.
- Authors state that the structure of β 1-CT is not available. A quick search on PDB shows a (1) ICAP1 structure in complex with integrin beta 1 cytoplasmic tail (PDB: 4DX9) and (2) Structural basis of kindlin-mediated integrin recognition and activation (PDB: 5XQ0) is available.
- The authors did not respond to the question regarding why mapping was not done for Kindlin.

Comment 6

- The authors state that "We have removed the statement about the average peak intensity line in the text, and in fact do not show an average intensity line". However, instead of adding the average intensity line they removed it. The authors should include it in both the text and figure.

Comment 8

- Fig 2 – The author's response for the lack of K402 CSP in panel 2A and 2D is rational considering mutagenesis of this residue affects binding interactions. However, K401 greatly broadens out while K402 shows no perturbation. This doesn't seem to correlate with their explanation that the K402 amide is involved in helical interactions as the main rationale for the lack of spectral perturbation.

Comment 11

- The constructs are described in supplementary data, yet not in the main text. The rationale for use of these constructs, and conclusion from the K_d should be described.

Comment 14

The authors respond to the concern regarding NMR assignment transfer as follows. "For most signals chemical shifts are in excellent agreement and only those were analyzed". It would be helpful to detail which assignments could not be reliably transferred, as they are no longer reporters for binding interactions.

Author Rebuttal, first revision:

Reviewer #1 (Remarks to the Author):

The revised manuscript and additional information provided for the reviewers addressed the questions raised.

Reviewer #2 (Remarks to the Author):

In the resubmission of the manuscript entitled 'Talin and kindlin combine integrin tail allostery and direct binding to activate integrins', the majority of the comments were corrected as requested in the updated manuscript. However, some minor concerns re: reviewer 2 should be addressed prior to publication.

Comment 4

- While the authors address the comment in the rebuttal, the explanation does not appear to be in the text.

We mention now in the text that proteins were minimized to reduce molecular weights: '.....we used NMR spectroscopy to characterize the interaction of isotope-labeled β 1-CT with the unlabeled F3 domain of TLN1 (TLN1-F3) and KIND2 lacking the flexible loop in F1 and the PH domain (Δ KIND2; Figures 1A; S1A) to reduce molecular weight of these constructs and enhance both solubility and NMR spectral quality.'

- Authors state that the structure of β 1-CT is not available. A quick search on PDB shows a (1) ICAP1 structure in complex with integrin beta 1 cytoplasmic tail (PDB: 4DX9) and (2) Structural basis of kindlin-mediated integrin recognition and activation (PDB: 5XQ0) is available.

There is no structure available of the complete β 1A-CT. The structures referred to by the reviewer comprise only small fragments of the β 1A-CT that were co-crystallized with ICAP and kindlin-2. In fact, we used this structural information with the appropriate reference to generate our structural model.

- The authors did not respond to the question regarding why mapping was not done for Kindlin. Mapping the spectral changes onto the full kindlin-2 structure is unfortunately not possible, as we could not determine chemical shift assignments for the full-length protein, as the high molecular weight and difficulties in sample quality precluded to obtain NMR spectra to enable chemical shift assignments. The list of KIND2-F0 backbone amide assignments (about 47%) that could be safely transferred from those published in the Biological Magnetic Resonance Data Bank (BMRB entry 30659) is now provided in suppl. Table 2. The CSPs for residues that could be assigned are mapped onto the structure in Fig. S2 now.

Comment 6

- The authors state that "We have removed the statement about the average peak intensity line in the text, and in fact do not show an average intensity line". However, instead of adding the average intensity line they removed it. The authors should include it in both the text and figure.

We added the average peak intensity line in the three main intensity plots in Figure 1C as requested. This is also mentioned in the text.

Comment 8

- Fig 2 – The author's response for the lack of K402 CSP in panel 2A and 2D is rational considering mutagenesis of this residue affects binding interactions. However, K401 greatly broadens out while K402 shows no perturbation. This doesn't seem to correlate with their

explanation that the K402 amide is involved in helical interactions as the main rationale for the lack of spectral perturbation.

The potential explanation of K402 being involved in a helical interaction is just a suggestion. However, we believe the data are fully consistent with both K401 and K402 being in the binding interface. It is important to note that different concentrations and stoichiometries are used for the NMR spectra in Fig. 2A (titration of TLN1-F3 with KIND2) and Fig. 2D (titration with Δ KIND2). Here, Fig. 2D shows NMR spectral effects for both the backbone amides of K401 and K402.

In Fig. 2A, there is clear line-broadening seen for the backbone amide of TLN1-F3-K401 at sub-stoichiometric concentrations of KIND2 protein. Interestingly, the K401 amide exhibits already some line-broadening in the absence of any ligand, indicative of conformational flexibility. In contrast, the lack of flexibility at μ s-ms timescales, which would argue that this residue may not be fully in a helical conformation, and the lack of notable spectral changes for the backbone amide of TLN1-F3-K402 under these substoichiometric conditions (Fig. 2A) may reflect its shielding in a helical conformation, as a possible explanation. Clearly high-resolution structural analysis eventually should resolve these details.

Most importantly, as shown in Fig. 2D, upon titration of TLN1-F3 with Δ KIND2 at 2-fold molar excess, we observed clear line-broadening for both the backbone amides of TLN1-F3-K401 and TLN1-F3-K402. This observation is fully consistent with both amides being located in the vicinity of the binding interface with KIND2. We apologize that the line-broadening may have been difficult to see in the superposition of the 2D NMR spectra. We have updated the presentation of the NMR spectra in Fig. 2D to make the line-broadening better visible.

Comment 11

- The constructs are described in supplementary data, yet not in the main text. The rationale for use of these constructs, and conclusion from the K_d should be described.

We substituted a large number of residues in THD1 to test which of them change the affinity for β -CT. Since we found that only K401, K402 and K403 lowered THD1 affinity for β -CT and not all the other residues we changed the text to 'MST measurements revealed that among all THD1 mutants tested (see Figure S2H), only lysine (K)401, K402 or K403, double substitutions of KK402/403 or deletions of amino acids 401-405 in THD1 lower the affinity for β 3-CTs in the presence of KIND2.'

Comment 14

The authors respond to the concern regarding NMR assignment transfer as follows. "For most signals chemical shifts are in excellent agreement and only those were analyzed". It would be helpful to detail which assignments could not be reliably transferred, as they are no longer reporters for binding interactions.

Backbone assignments for KIND2-F0 were transferred from a previously published study in the BMRB entry 30659. About 47% of the backbone amide chemical shifts could be transferred by comparison of $^1\text{H}, ^{15}\text{N}$ correlation spectra. Other signals could not be unambiguously identified due to signal overlap. The transferred chemical shift assignments of KIND2-F0 are now provided in suppl. Table 2.

We also have added a mapping of the chemical shift perturbations upon titration of KIND2 in Figure S3.

Final Decision Letter:**Message** 26th Sep 2023

:

Dear Dr. Fässler,

We are now happy to accept your revised paper "Talin and kindlin use integrin tail allostery and direct binding to activate integrins" for publication as an Article in Nature Structural & Molecular Biology.

As soon as your article is published, you can generate your shareable link by entering the DOI of your article here: http://authors.springernature.com/share. Corresponding authors will also receive an automated email with the shareable link

Your paper will be published online soon after we receive proof corrections and will appear in print in the next available issue. You can find out your date of online publication by contacting the production team shortly after sending your proof corrections. Content is published online weekly on Mondays and Thursdays, and the embargo is set at 16:00 London time (GMT)/11:00 am US Eastern time (EST) on the day of publication. Now is the

time to inform your Public Relations or Press Office about your paper, as they might be interested in promoting its publication. This will allow them time to prepare an accurate and satisfactory press release. Include your manuscript tracking number (NSMB-A46550C) and our journal name, which they will need when they contact our press office.

About one week before your paper is published online, we shall be distributing a press release to news organizations worldwide, which may very well include details of your work. We are happy for your institution or funding agency to prepare its own press release, but it must mention the embargo date and Nature Structural & Molecular Biology. If you or your Press Office have any enquiries in the meantime, please contact press@nature.com.

Please note that *Nature Structural & Molecular Biology* is a Transformative Journal (TJ). Authors may publish their research with us through the traditional subscription access route or make their paper immediately open access through payment of an article-processing charge (APC). Authors will not be required to make a final decision about access to their article until it has been accepted. [Find out more about Transformative Journals](https://www.springernature.com/gp/open-research/transformative-journals)

Authors may need to take specific actions to achieve [compliance](https://www.springernature.com/gp/open-research/funding/policy-compliance-faqs) with funder and institutional open access mandates. If your research is supported by a funder that requires immediate open access (e.g. according to [Plan S principles](https://www.springernature.com/gp/open-research/plan-s-compliance)) then you should select the gold OA route, and we will direct you to the compliant route where possible. For authors selecting the subscription publication route, the journal's standard licensing terms will need to be accepted, including [self-archiving policies](https://www.springernature.com/gp/open-research/policies/journal-policies). Those licensing terms will supersede any other terms

that the author or any third party may assert apply to any version of the manuscript.

Sincerely,

Katarzyna Ciazynska
(she/her)
Associate Editor
Nature Structural & Molecular Biology
<https://orcid.org/0000-0002-9899-2428>
